# Learning Neural Networks with Distribution Shift: Efficiently Certifiable Guarantees

**Gautam Chandrasekaran, Adam R. Klivans, Lin Lin Lee, Konstantinos Stavropoulos**
The University of Texas at Austin
`{gautamc,klivans,kstavrop}@cs.utexas.edu`
`llee3@utexas.edu`

## Abstract

We give the first provably efficient algorithms for learning neural networks with respect to distribution shift. We work in the Testable Learning with Distribution Shift framework (TDS learning) of Klivans et al. (2024a), where the learner receives labeled examples from a training distribution and unlabeled examples from a test distribution and must either output a hypothesis with low test error or reject if distribution shift is detected. No assumptions are made on the test distribution.

All prior work in TDS learning focuses on classification, while here we must handle the setting of nonconvex regression. Our results apply to real-valued networks with arbitrary Lipschitz activations and work whenever the training distribution has strictly sub-exponential tails. For training distributions that are bounded and hypercontractive, we give a fully polynomial-time algorithm for TDS learning one hidden-layer networks with sigmoid activations. We achieve this by importing classical kernel methods into the TDS framework using data-dependent feature maps and a type of kernel matrix that couples samples from both train and test distributions.

## 1 Introduction

Understanding when a model will generalize from a known training distribution to an unknown test distribution is a critical challenge in trustworthy machine learning and domain adaptation. Traditional approaches to this problem prove generalization bounds in terms of various notions of distance between train and test distributions (Ben-David et al., 2006; 2010; Mansour et al., 2009) but do not provide efficient algorithms. Recent work due to Klivans et al. (2024a) departs from this paradigm and defines the model of Testable Learning with Distribution Shift (TDS learning), where a learner may reject altogether if significant distribution shift is detected. When the learner accepts, however, it outputs a classifier and a proof that the classifier has nearly optimal test error.

A sequence of works has given the first set of efficient algorithms in the TDS learning model for well-studied function classes where no assumptions are taken on the test distribution (Klivans et al., 2024a;b; Chandrasekaran et al., 2024; Goel et al., 2024). These results, however, hold for classification and therefore do not apply to (nonconvex) regression problems and in particular to a long line of work giving provably efficient algorithms for learning simple classes of neural networks under natural distributional assumptions on the training marginal (Goel & Klivans, 2019; Diakonikolas et al., 2020a;c; 2022; Chen et al., 2022b; 2023; Wang et al., 2023; Gollakota et al., 2024a; Diakonikolas & Kane, 2024).

The main contribution of this work is the first set of efficient TDS learning algorithms for broad classes of (nonconvex) regression problems. Our results apply to neural networks with arbitrary Lipschitz activations of any constant depth. As one example, we obtain a fully polynomial-time algorithm for learning one hidden-layer neural networks with sigmoid activations with respect to any bounded and hypercontractive training distribution. For bounded training distributions, the running times of our algorithms match the best known running times for ordinary PAC or agnostic learning (without distribution shift). We emphasize that unlike all prior work in domain adaptation, we make no assumptions on the test distribution.

**Regression Setting.** We assume access to labeled examples from the training distribution and unlabeled examples from the marginal of the test distribution. We consider the squared loss $\mathcal{L}_{\mathcal{D}}(h) = \sqrt{\mathbb{E}_{(\boldsymbol{x},y)\sim\mathcal{D}}[(y - h(\boldsymbol{x}))^2]}$. The error benchmark is analogous to the benchmark for TDS learning in classification (Klivans et al., 2024a) and depends on two quantities: the optimum training error achievable by a classifier in the learnt class, $\text{opt} = \min_{f\in\mathcal{F}}[\mathcal{L}_{\mathcal{D}}(f)]$, and the best joint error achievable by a single classifier on both the training and test distributions, $\lambda = \min_{f'\in\mathcal{F}}[\mathcal{L}_{\mathcal{D}}(f') + \mathcal{L}_{\mathcal{D}'}(f')]$. Achieving an error of $\text{opt} + \lambda$ is the standard goal in domain adaptation (Ben-David et al., 2006; Blitzer et al., 2007; Mansour et al., 2009). We now formally define the TDS learning framework for regression:

**Definition 1.1** (Testable Regression with Distribution Shift)**.** For $\epsilon, \delta \in (0, 1)$ and a function class $\mathcal{F} \subseteq \{\mathbb{R}^d \to \mathbb{R}\}$, the learner receives iid labeled examples from some unknown training distribution $\mathcal{D}$ over $\mathbb{R}^d \times \mathbb{R}$ and iid unlabeled examples from the marginal $\mathcal{D}'_{\boldsymbol{x}}$ of another unknown test distribution $\mathcal{D}'$ over $\mathbb{R}^d \times \mathbb{R}$. The learner either rejects, or it accepts and outputs hypothesis $h : \mathbb{R}^d \to \mathbb{R}$ such that the following are true.

1. (Soundness) With probability at least $1 - \delta$, if the algorithm accepts, then the output $h$ satisfies $\mathcal{L}_{\mathcal{D}'}(h) \leq \min_{f\in\mathcal{F}}[\mathcal{L}_{\mathcal{D}}(f)] + \min_{f'\in\mathcal{F}}[\mathcal{L}_{\mathcal{D}}(f') + \mathcal{L}_{\mathcal{D}'}(f')] + \epsilon$.

2. (Completeness) If $\mathcal{D}_{\boldsymbol{x}} = \mathcal{D}'_{\boldsymbol{x}}$, then the algorithm accepts with probability at least $1 - \delta$.

## 1.1 Technical Statement of Results

Our results hold for classes of Lipschitz neural networks. In particular, we consider functions $f$ of the following form. Let $\sigma : \mathbb{R} \to \mathbb{R}$ be an activation function. Let $\mathbf{W} = (W^{(1)}, \ldots W^{(t)})$ with $W^{(i)} \in \mathbb{R}^{s_i \times s_{i-1}}$ be the tuple of weight matrices. Here, $s_0 = d$ is the input dimension and $s_t = 1$. Define recursively the function $f_i : \mathbb{R}^d \to \mathbb{R}^{s_i}$ as $f_i(\boldsymbol{x}) = W^{(i)} \cdot \sigma(f_{i-1}(\boldsymbol{x}))$ with $f_1(\boldsymbol{x}) = W^{(1)} \cdot \boldsymbol{x}$. The function $f : \mathbb{R}^d \to \mathbb{R}$ computed by the neural network $(\mathbf{W}, \sigma)$ is defined as $f(\boldsymbol{x}) := f_t(\boldsymbol{x})$. The depth of this network is $t$.

We now present our main results on TDS learning for neural networks.

| Function Class | Runtime (Bounded) | Runtime (Subgaussian) |
|---|---|---|
| One hidden-layer Sigmoid Net | $\text{poly}(d, M, 1/\epsilon)$ | $d^{\text{poly}(k \log(M/\epsilon))}$ |
| Single ReLU | $\text{poly}(d, M) \cdot 2^{O(1/\epsilon)}$ | $d^{\text{poly}(\log(M)/\epsilon)}$ |
| Sigmoid Nets | $\text{poly}(d, M) \cdot 2^{O\left((\log(1/\epsilon))^{t-1}\right)}$ | $d^{\text{poly}(k \log(M)(\log(1/\epsilon)^{t-1}))}$ |
| 1-Lipschitz Nets | $\text{poly}(d, M) \cdot 2^{\tilde{O}(k\sqrt{k}2^{t-1}/\epsilon)}$ | $d^{\text{poly}(k2^{t-1}\log(M)/\epsilon)}$ |

Table 1: In the above table, $k$ denotes the number of neurons in the first hidden layer. $M$ denotes a bound on the labels of the train and test distributions. One hidden-layer Sigmoid nets refers to depth 2 neural networks with sigmoid activation. The bounded distributions considered in the above table have support on the unit ball. We assume that all relevant parameters of the neural network are bounded by constants. For more detailed statements and proofs, see Definition A.12 and (1) Corollaries B.4 and B.6 and Theorems B.3 and B.5 for the bounded case, and (2) Theorems C.9 and C.10 for the Subgaussian case.

From the above table, we highlight that in the cases of bounded distributions with (1) one hidden-layer Sigmoid Nets, and (2) Single ReLU with $\epsilon < 1/\log d$, we obtain TDS algorithms that run in polynomial time in all parameters. Moreover, for the last row, regarding Lipschitz Nets, each neuron is allowed to have a different and unknown Lipschitz activation. Therefore, in particular, our results capture the class of single-index models (see, e.g., Kakade et al. (2011); Gollakota et al. (2024a)).

In the results of Table 1, we assume bounded labels for both the training and test distributions. This assumption can be relaxed to a bound on any moment whose degree is strictly higher than 2 (see Corollary D.2). In fact, such an assumption is necessary, as we show in Proposition D.1.

## 1.2 OUR TECHNIQUES

**TDS Learning via Kernel Methods.** The major technical contribution of this work is devoted to importing classical kernel methods into the TDS learning framework. A first attempt at testing distribution shift with respect to a fixed feature map would be to form two corresponding covariance matrices of the expanded features, one from samples drawn from the training distribution and the other from samples drawn from the test distribution, and test if these two matrices have similar eigen-decompositions. This approach only yields efficient algorithms for linear kernels, however, as here we are interested in spectral properties of covariance matrices in the feature space corresponding to low-degree polynomials, whose dimension is too large.

Instead we form a new data-dependent and concise reference feature map $\phi$, that depends on examples from both $\mathcal{D}_{\boldsymbol{x}}$ and $\mathcal{D}'_{\boldsymbol{x}}$. We show that this feature map approximately represents the ground truth, i.e., some function with both low training and test error (this is due to the representer theorem, see Proposition 3.7). To certify that error bounds transfer from $\mathcal{D}_{\boldsymbol{x}}$ to $\mathcal{D}'_{\boldsymbol{x}}$, we require *relative error* closeness between covariance matrix $\Phi' = \mathbb{E}_{\boldsymbol{x} \sim \mathcal{D}'_{\boldsymbol{x}}}[\phi(\boldsymbol{x})\phi(\boldsymbol{x})^\top]$ of the feature expansion $\phi$ over the test marginal with the corresponding matrix $\Phi = \mathbb{E}_{\boldsymbol{x} \sim \mathcal{D}_{\boldsymbol{x}}}[\phi(\boldsymbol{x})\phi(\boldsymbol{x})^\top]$ over the training marginal. We draw fresh sets of verification examples and show how the kernel trick can be used to efficiently achieve these approximations even though $\phi$ is a nonstandard feature map. For more technical details, see Section 3.1.

By instantiating the above results using a type of polynomial kernel, we can reduce the problem of TDS learning neural networks to the problem of obtaining an appropriate polynomial approximator. Our final *training* algorithm (as opposed to the testing phase) will essentially be kernelized polynomial regression.

**TDS Learning and Uniform Approximation.** Prior work in TDS learning has established connections between polynomial approximation theory and efficient algorithms in the TDS setting. In particular, the existence of low-degree sandwiching approximators for a concept class is known to imply dimension-efficient TDS learning algorithms for binary classification. The notion of sandwiching approximators for a function $f$ refers to a pair of low-degree polynomials $p_{\mathrm{up}}, p_{\mathrm{down}}$ with two main properties: (1) $p_{\mathrm{down}} \leq f \leq p_{\mathrm{up}}$ everywhere and (2) the expected absolute distance between $p_{\mathrm{up}}$ and $p_{\mathrm{down}}$ over some reference distribution is small. The first property is of particular importance in the TDS setting, since it holds everywhere and, therefore, it holds for any test distribution unconditionally.

Here we make the simple observation that the incomparable notion of uniform approximation suffices for TDS learning. A uniform approximator is a polynomial $p$ that approximates a function $f$ pointwise, meaning that $|p - f|$ is small in every point within a ball around the origin (there is no known direct relationship between sandwiching and uniform approximators). In our setting, uniform approximation is more convenient, due to the existence of powerful tools from polynomial approximation theory regarding Lipschitz and analytic functions.

Contrary to the sandwiching property, the uniform approximation property cannot hold everywhere if the approximated function class contains high-(or infinite-)degree functions. When the training distribution has strictly sub-exponential tails, however, the expected error of approximation outside the radius of approximation is negligible. Importantly, this property can be certified for the test distribution by using a moment-matching tester. See also Section 4.

## 1.3 RELATED WORK

**Learning with Distribution Shift.** The field of domain adaptation has been studying the distribution shift problem for almost two decades (Ben-David et al., 2006; Blitzer et al., 2007; Ben-David et al., 2010; Mansour et al., 2009; David et al., 2010; Mousavi Kalan et al., 2020; Redko et al., 2020; Kalavasis et al., 2024; Hanneke & Kpotufe, 2019; 2024; Awasthi et al., 2024), providing useful insights regarding the information-theoretic (im)possibilities for learning with distribution shift. The first efficient end-to-end algorithms for non-trivial concept classes with distribution shift were given for TDS learning in Klivans et al. (2024a;b); Chandrasekaran et al. (2024) and for PQ learning, originally defined by Goldwasser et al. (2020), in Goel et al. (2024). These works focus on binary classification for classes like halfspaces, halfspace intersections, and geometric concepts. In the regression setting, we need to handle unbounded loss functions, but we are also able to use Lipschitz

properties of real-valued networks to obtain results even for deeper architectures. For the special case of linear regression, efficient algorithms for learning with distribution shift are known to exist (see, e.g., Lei et al. (2021)), but our results capture much broader classes.

Another distinction between the existing works in TDS learning and our work, is that our results require significantly milder assumptions on the training distribution. In particular, while all prior works on TDS learning require both concentration and anti-concentration for the training marginal (Klivans et al., 2024a;b; Chandrasekaran et al., 2024), we only assume strictly subexponential concentration in every direction. This is possible because the function classes we consider are Lipschitz, which is not the case for binary classification.

**Testable Learning.** More broadly, TDS learning is related to the notion of testable learning (Rubinfeld & Vasilyan, 2023; Gollakota et al., 2023; 2024c; Diakonikolas et al., 2023; Gollakota et al., 2024b; Diakonikolas et al., 2024; Slot et al., 2024), originally defined by Rubinfeld & Vasilyan (2023) for standard agnostic learning, aiming to certify optimal performance for learning algorithms without relying directly on any distributional assumptions. The main difference between testable agnostic learning and TDS learning is that in TDS learning, we allow for distribution shift, while in testable agnostic learning the training and test distributions are the same. Because of this, TDS learning remains challenging even in the absence of label noise, in which case testable learning becomes trivial (Klivans et al., 2024a).

**Efficient Learning of Neural Networks.** Many works have focused on providing upper and lower bounds on the computational complexity of learning neural networks in the standard (distribution-shift-free) setting (Goel et al., 2017; Goel & Klivans, 2019; Goel et al., 2020a;b; Diakonikolas et al., 2020a;b;c; 2022; Chen et al., 2022a;b; 2023; Wang et al., 2023; Gollakota et al., 2024a; Diakonikolas & Kane, 2024; Li et al., 2020; Gao et al., 2019; Zhang et al., 2019; Vempala & Wilmes, 2019; Allen-Zhu et al., 2019; Bakshi et al., 2019; Manurangsi & Reichman, 2018; Ge et al., 2019; 2018; Du et al., 2018; Goel et al., 2018; Tian, 2017; Li & Yuan, 2017; Brutzkus & Globerson, 2017; Zhong et al., 2017; Zhang et al., 2016b; Janzamin et al., 2015). The majority of the upper bounds either require noiseless labels and shallow architectures or work only under Gaussian training marginals. Our results not only hold in the presence of distribution shift, but also capture deeper architectures, under any strictly subexponential training marginal and allow adversarial label noise.

The upper bounds that are closest to our work are those given by Goel et al. (2017). They consider ReLU as well as sigmoid networks, allow for adversarial label noise and assume that the training marginal is bounded but otherwise arbitrary. Our results in Section 3 extend all of the results in Goel et al. (2017) to the TDS setting, by assuming additionally that the training distribution is hypercontractive (see Definition 3.9). This additional assumption is important to ensure that our tests will pass when there is no distribution shift. For a more thorough technical comparison with Goel et al. (2017), see Section 3.

In Section 4, we provide upper bounds for TDS learning of Lipschitz networks even when the training marginal is an arbitrary strictly subexponential distribution. In particular, our results imply new bounds for standard agnostic learning of single ReLU neurons, where we achieve runtime $d^{\text{poly}(1/\epsilon)}$. The only known upper bounds work under the Gaussian marginal (Diakonikolas et al., 2020a), achieving similar runtime. In fact, in the statistical query framework (Kearns, 1998), it is known that $d^{\text{poly}(1/\epsilon)}$ runtime is necessary for agnostically learning the ReLU, even under the Gaussian distribution (Diakonikolas et al., 2020b; Goel et al., 2020b).

## 2 PRELIMINARIES

We use standard vector and matrix notation. We denote with $\mathbb{R}, \mathbb{N}$ the sets of real and natural numbers accordingly. We denote with $\mathcal{D}$ labeled distributions over $\mathbb{R}^d \times \mathbb{R}$ and with $\mathcal{D}_{\boldsymbol{x}}$ the marginal of $\mathcal{D}$ on the features in $\mathbb{R}^d$. For a set $S$ of points in $\mathbb{R}^d$, we define the empirical probabilities (resp. expectations) as $\mathbf{Pr}_{\boldsymbol{x} \sim S}[E(\boldsymbol{x})] = \frac{1}{|S|} \sum_{\boldsymbol{x} \in S} \mathbb{1}\{E(\boldsymbol{x})\}$ (resp. $\mathbb{E}_{\boldsymbol{x} \sim S}[f(\boldsymbol{x})] = \frac{1}{|S|} \sum_{\boldsymbol{x} \in S} f(\boldsymbol{x})$). We denote with $\bar{S}$ the labeled version of $S$ and we define the clipping function $\text{cl}_M : \mathbb{R} \to [-M, M]$, that maps a number $t \in \mathbb{R}$ either to itself if $t \in [-M, M]$, or to $M \cdot \text{sign}(t)$ otherwise.

**Loss function.** Throughout this work, we denote with $\mathcal{L}_{\mathcal{D}}(h)$ the squared loss of a hypothesis $h : \mathbb{R}^d \to \mathbb{R}$ with respect to a labeled distribution $\mathcal{D}$, i.e., $\mathcal{L}_{\mathcal{D}}(h) = \sqrt{\mathbb{E}_{(\boldsymbol{x}, y) \sim \mathcal{D}}[(y - h(\boldsymbol{x}))^2]}$. More-

over, for any function $f : \mathbb{R}^d \to \mathbb{R}$, we denote with $\|f\|_{\mathcal{D}}$ the quantity $\|f\|_{\mathcal{D}} = \sqrt{\mathbb{E}_{\boldsymbol{x} \sim \mathcal{D}_{\boldsymbol{x}}}[(f(\boldsymbol{x}))^2]}$. For a set of labeled examples $\bar{S}$, we denote with $\mathcal{L}_{\bar{S}}(h)$ the empirical loss on $\bar{S}$, i.e., $\mathcal{L}_{\bar{S}}(h) = \sqrt{\frac{1}{|S|} \sum_{(\boldsymbol{x},y) \in \bar{S}} (y - h(\boldsymbol{x}))^2}$ and similarly for $\|f\|_S$.

**Distributional Assumptions.** In order to obtain efficient algorithms, we will either assume that the training marginal $\mathcal{D}_{\boldsymbol{x}}$ is bounded and hypercontractive (Section 3) or that it has strictly subexponential tails in every direction (Section 4). We make no assumptions on the test marginal $\mathcal{D}'_{\boldsymbol{x}}$.

Regarding the labels, we assume some mild bound on the moments of the training and the test labels, e.g., (a) that $\mathbb{E}_{y \sim \mathcal{D}_y}[y^4], \mathbb{E}_{y \sim \mathcal{D}'_y}[y^4] \leq M$ or (b) that $y \in [-M, M]$ a.s. for both $\mathcal{D}$ and $\mathcal{D}'$. Although, ideally, we want to avoid any assumptions on the test distribution, as we show in Proposition D.1, a bound on some constant-degree moment of the test labels is necessary.

## 3 BOUNDED TRAINING MARGINALS

We begin with the scenario where the training distribution is known to be bounded. In this case, it is known that one-hidden-layer sigmoid networks can be agnostically learned (in the classical sense, without distribution shift) in fully polynomial time and single ReLU neurons can be learned up to error $O(\frac{1}{\log(d)})$ in polynomial time (Goel et al., 2017). These results are based on a kernel-based approach, combined with results from polynomial approximation theory. While polynomial approximations can reduce the nonconvex agnostic learning problem to a convex one through polynomial feature expansions, the kernel trick enables further pruning of the search space, which is important for obtaining polynomial-time algorithms. Our work demonstrates another useful implication of the kernel trick: it leads to efficient algorithms for testing distribution shift.

We will require the following standard notions:

**Definition 3.1** (Kernels (Mercer, 1909)). A function $\mathcal{K} : \mathbb{R}^d \times \mathbb{R}^d \to \mathbb{R}$ is a kernel. If for any set of $m$ points $\boldsymbol{x}_1, \ldots, \boldsymbol{x}_m$ in $\mathbb{R}^d$, the matrix $(\mathcal{K}(\boldsymbol{x}_i, \boldsymbol{x}_j))_{(i,j) \in [m]}$ is positive semidefinite, we say that the kernel $\mathcal{K}$ is positive definite. The kernel $\mathcal{K}$ is symmetric if for all $\boldsymbol{x}, \boldsymbol{x}' \in \mathbb{R}^d$, $\mathcal{K}(\boldsymbol{x}, \boldsymbol{x}') = \mathcal{K}(\boldsymbol{x}', \boldsymbol{x})$.

Any PSD kernel is associated with some Hilbert space $\mathbb{H}$ and some feature map from $\mathbb{R}^d$ to $\mathbb{H}$.

**Fact 3.2** (Reproducing Kernel Hilbert Space). *For any positive definite and symmetric (PDS) kernel $\mathcal{K}$, there is a Hilbert space $\mathbb{H}$, equipped with the inner product $\langle \cdot, \cdot \rangle : \mathbb{H} \times \mathbb{H} \to \mathbb{R}$ and a function $\psi : \mathbb{R}^d \to \mathbb{H}$ such that $\mathcal{K}(\boldsymbol{x}, \boldsymbol{x}') = \langle \psi(\boldsymbol{x}), \psi(\boldsymbol{x}') \rangle$ for all $\boldsymbol{x}, \boldsymbol{x}' \in \mathbb{R}^d$. We call $\mathbb{H}$ the reproducing kernel Hilbert space (RKHS) for $\mathcal{K}$ and $\psi$ the feature map for $\mathcal{K}$.*

There are three main properties of the kernel method. First, although the associated feature map $\psi$ may correspond to a vector in an infinite-dimensional space, the kernel $\mathcal{K}(\boldsymbol{x}, \boldsymbol{x}')$ may still be efficiently evaluated, due to its analytic expression in terms of $\boldsymbol{x}, \boldsymbol{x}'$. Second, the function class $\mathcal{F}_{\mathcal{K}} = \{\boldsymbol{x} \mapsto \langle \boldsymbol{v}, \psi(\boldsymbol{x}) \rangle : \boldsymbol{v} \in \mathbb{H}, \langle \boldsymbol{v}, \boldsymbol{v} \rangle \leq B\}$ has Rademacher complexity independent from the dimension of $\mathbb{H}$, as long as the maximum value of $\mathcal{K}(\boldsymbol{x}, \boldsymbol{x})$ for $\boldsymbol{x}$ in the domain is bounded (Thm. 6.12 in Mohri et al. (2018)). Third, the time complexity of finding the function in $\mathcal{F}_{\mathcal{K}}$ that best fits a dataset is actually polynomial to the size of the dataset, due to the representer theorem (Thm. 6.11 in Mohri et al. (2018)). Taken together, these properties constitute the basis of the kernel method, implying learners with runtime independent from the effective dimension of the learning problem.

In order to apply the kernel method to learn some function class $\mathcal{F}$, it suffices to show that the class $\mathcal{F}$ can be represented sufficiently well by the class $\mathcal{F}_{\mathcal{K}}$. We give the following definition.

**Definition 3.3** (Approximate Representation). Let $\mathcal{F}$ be a function class over $\mathbb{R}^d$, $\mathcal{K} : \mathbb{R}^d \times \mathbb{R}^d \to \mathbb{R}$ a PDS kernel, where $\mathbb{H}$ is the corresponding RKHS and $\psi$ the feature map for $\mathcal{K}$. We say that $\mathcal{F}$ can be $(\epsilon, B)$-approximately represented within radius $R$ with respect to $\mathcal{K}$ if for any $f \in \mathcal{F}$, there is $\boldsymbol{v} \in \mathbb{H}$ with $\langle \boldsymbol{v}, \boldsymbol{v} \rangle \leq B$ such that $|f(\boldsymbol{x}) - \langle \boldsymbol{v}, \psi(\boldsymbol{x}) \rangle| \leq \epsilon$, for all $\boldsymbol{x} \in \mathbb{R}^d : \|\boldsymbol{x}\|_2 \leq R$.

For the purposes of TDS learning, we will also require the training marginal to have be hypercontractive with respect to the kernel at hand. This is important to ensure that our test will accept whenever there is no distribution shift. More formally, we require the following.

**Definition 3.4** (Hypercontractivity). Let $\mathcal{D}_{\boldsymbol{x}}$ be some distribution over $\mathbb{R}^d$, let $\mathbb{H}$ be a Hilbert space and let $\psi : \mathbb{R}^d \to \mathbb{H}$. We say that $\mathcal{D}_{\boldsymbol{x}}$ is $(\psi, C, \ell)$-hypercontractive if for any $t \in \mathbb{N}$ and $\boldsymbol{v} \in \mathbb{H}$:

$$\mathbb{E}_{\boldsymbol{x} \sim \mathcal{D}_{\boldsymbol{x}}}[\langle \boldsymbol{v}, \psi(\boldsymbol{x}) \rangle^{2t}] \leq (Ct)^{2\ell t}(\mathbb{E}_{\boldsymbol{x} \sim \mathcal{D}_{\boldsymbol{x}}}[\langle \boldsymbol{v}, \psi(\boldsymbol{x})^2 \rangle])^t$$

If $\mathcal{K}$ is the PDS kernel corresponding to $\psi$, we also say that $\mathcal{D}_{\boldsymbol{x}}$ is $(\mathcal{K}, C, \ell)$-hypercontractive.

### 3.1 TDS Regression via the Kernel Method

We now give a general theorem on TDS regression for bounded distributions, under the following assumptions. Note that, although we assume that the training and test labels are bounded, this assumption can be relaxed in a black-box manner and bounding some constant-degree moment of the distribution of the labels suffices, as we show in Corollary D.2.

**Assumption 3.5.** *For a function class $\mathcal{F} \subseteq \{\mathbb{R}^d \to \mathbb{R}\}$, and training and test distributions $\mathcal{D}, \mathcal{D}'$ over $\mathbb{R}^d \times \mathbb{R}$, we assume the following.*

1. *$\mathcal{F}$ is $(\epsilon, B)$-approximately represented within radius $R$ w.r.t. a PDS kernel $\mathcal{K} : \mathbb{R}^d \times \mathbb{R}^d \to \mathbb{R}$, for some $\epsilon \in (0, 1)$ and $B, R \geq 1$ and let $A = \sup_{\boldsymbol{x}:\|\boldsymbol{x}\|_2 \leq R} \mathcal{K}(\boldsymbol{x}, \boldsymbol{x})$.*

2. *The training marginal $\mathcal{D}_{\boldsymbol{x}}$ (1) is bounded within $\{\boldsymbol{x} : \|\boldsymbol{x}\|_2 \leq R\}$ and (2) is $(\mathcal{K}, C, \ell)$-hypercontractive for some $C, \ell \geq 1$.*

3. *The training and test labels are both bounded in $[-M, M]$ for some $M \geq 1$.*

Consider the function class $\mathcal{F}$, the kernel $\mathcal{K}$ and the parameters $\epsilon, A, B, C, M, \ell$ as defined in the assumption above and let $\delta \in (0, 1)$. Then, we obtain the following theorem.

**Theorem 3.6** (TDS Learning via the Kernel Method). *Under Assumption 3.5, Algorithm 1 learns the class $\mathcal{F}$ in the TDS regression setting up to excess error $5\epsilon$ and probability of failure $\delta$. The time complexity is $O(T) \cdot \mathrm{poly}(d, \frac{1}{\epsilon}, (\log(1/\delta))^\ell, A, B, C^\ell, 2^\ell, M)$, where $T$ is the evaluation time of $\mathcal{K}$.*

The main ideas of the proof are the following.

**Obtaining a concise reference feature map.** The algorithm first draws reference sets $S_{\mathrm{ref}}, S'_{\mathrm{ref}}$ from both the training and the test distributions. The representer theorem, combined with the approximate representation assumption (Definition 3.3) ensure that the reference examples define a new feature map $\phi : \mathbb{R}^d \to \mathbb{R}^{2m}$ with $\phi(\boldsymbol{x}) = (\mathcal{K}(\boldsymbol{x}, \boldsymbol{z}))_{\boldsymbol{z} \in S_{\mathrm{ref}} \cup S'_{\mathrm{ref}}}$ such that the ground truth $f^* = \arg\min_{f \in \mathcal{F}}[\mathcal{L}_{\mathcal{D}}(f) + \mathcal{L}_{\mathcal{D}'}(f)]$ can be approximately represented as a linear combination of the features in $\phi$ with respect to both $S_{\mathrm{ref}}$ and $S'_{\mathrm{ref}}$, i.e., $\|f^* - (\boldsymbol{a}^*)^\top \phi\|_{S_{\mathrm{ref}}}$ and $\|f^* - (\boldsymbol{a}^*)^\top \phi\|_{S'_{\mathrm{ref}}}$ are both small for some $\boldsymbol{a}^* \in \mathbb{R}^{2m}$. In particular, we have the following.

**Proposition 3.7** (Representer Theorem, modification of Theorem 6.11 in Mohri et al. (2018)). *Suppose that a function $f : \mathbb{R}^d \to \mathbb{R}$ can be $(\epsilon, B)$-approximately represented within radius $R$ w.r.t. some PDS kernel $\mathcal{K}$ (as per Definition 3.3). Then, for any set of examples $S$ in $\{\boldsymbol{x} \in \mathbb{R}^d : \|\boldsymbol{x}\|_2 \leq R\}$, there is $\boldsymbol{a} = (a_{\boldsymbol{x}})_{\boldsymbol{x} \in S} \in \mathbb{R}^{|S|}$ such that for $\tilde{p}(\boldsymbol{x}) = \sum_{\boldsymbol{z} \in S} a_{\boldsymbol{z}} \mathcal{K}(\boldsymbol{z}, \boldsymbol{x})$ we have:*

$$\|f - \tilde{p}\|_S \leq \epsilon \text{ and } \sum_{\boldsymbol{x}, \boldsymbol{z} \in S} a_{\boldsymbol{x}} a_{\boldsymbol{z}} \mathcal{K}(\boldsymbol{z}, \boldsymbol{x}) \leq B$$

*Proof.* We first observe that there is some $\boldsymbol{v} \in \mathbb{H}$ such that $\langle \boldsymbol{v}, \boldsymbol{v} \rangle \leq B$ and for $p(\boldsymbol{x}) = \langle \boldsymbol{v}, \psi(\boldsymbol{x}) \rangle$ we have $\|f - p\|_S \leq \epsilon$, because by Definition 3.3, there is a pointwise approximator for $f$ with respect to $\mathcal{K}$. By Theorem 6.11 in Mohri et al. (2018), this implies the existence of $\tilde{p}$ as desired. $\square$

Note that since the evaluation of $\phi(\boldsymbol{x})$ only involves Kernel evaluations, we never need to compute the initial feature expansion $\psi(\boldsymbol{x})$ which could be overly expensive.

**Forming a candidate output hypothesis.** We know that the reference feature map approximately represents the ground truth. However, having no access to test labels, we cannot directly hope to find the corresponding coefficient $\boldsymbol{a}^* \in \mathbb{R}^{2m}$. Instead, we use only the training reference examples to find a candidate hypothesis $\hat{p}$ with close-to-optimal performance on the training distribution which can be also expressed in terms of the reference feature map $\phi$, as $\hat{p} = \hat{\boldsymbol{a}}^\top \phi$. It then suffices to test the quality of $\phi$ on the test distribution.

---

**Algorithm 1:** TDS Regression via the Kernel Method

---

**Input:** Parameters $M, R, B, A, C, \ell \geq 1$, $\epsilon, \delta \in (0, 1)$ and sample access to $\mathcal{D}, \mathcal{D}'_{\boldsymbol{x}}$

Set $m = c \frac{(ABM)^4}{\epsilon^4} \log(\frac{1}{\delta})$, $N = cm^2 \frac{ABC}{\epsilon^4} (4C \log(\frac{4}{\delta}))^{4\ell+1}$, $c$ large enough constant

Draw $m$ i.i.d. labeled examples $\bar{S}_{\text{ref}}$ from $\mathcal{D}$ and $m$ i.i.d. unlabeled examples $S'_{\text{ref}}$ from $\mathcal{D}'_{\boldsymbol{x}}$;
**if** *for some* $\boldsymbol{x} \in S'_{\text{ref}}$ *we have* $\|\boldsymbol{x}\|_2 > R$ **then**
    | **Reject** and terminate;

Let $\hat{\boldsymbol{a}} = (\hat{a}_{\boldsymbol{z}})_{\boldsymbol{z} \in S_{\text{ref}}}$ be the optimal solution to the following convex program

$$\min_{\boldsymbol{a} \in \mathbb{R}^m} \sum_{(\boldsymbol{x}, y) \in \bar{S}_{\text{ref}}} \left( y - \sum_{\boldsymbol{z} \in S_{\text{ref}}} a_{\boldsymbol{z}} \mathcal{K}(\boldsymbol{z}, \boldsymbol{x}) \right)^2$$

$$\text{s.t.} \sum_{\boldsymbol{z}, \boldsymbol{w} \in S_{\text{ref}}} a_{\boldsymbol{z}} a_{\boldsymbol{w}} \mathcal{K}(\boldsymbol{z}, \boldsymbol{w}) \leq B, \text{ where } \boldsymbol{a} = (a_{\boldsymbol{z}})_{\boldsymbol{z} \in S_{\text{ref}}}$$

Draw $N$ i.i.d. unlabeled examples $S_{\text{ver}}$ from $\mathcal{D}_{\boldsymbol{x}}$ and $N$ unlabeled examples $S'_{\text{ver}}$ from $\mathcal{D}'_{\boldsymbol{x}}$;
**if** *for some* $\boldsymbol{x} \in S'_{\text{ver}}$ *we have* $\|\boldsymbol{x}\|_2 > R$ **then**
    | **Reject** and terminate;

Compute the matrix $\hat{\Phi} = (\hat{\Phi}_{\boldsymbol{z}, \boldsymbol{w}})_{\boldsymbol{z}, \boldsymbol{w} \in S_{\text{ref}} \cup S'_{\text{ref}}}$ with $\hat{\Phi}_{\boldsymbol{z}, \boldsymbol{w}} = \frac{1}{N} \sum_{\boldsymbol{x} \in S_{\text{ver}}} \mathcal{K}(\boldsymbol{x}, \boldsymbol{z}) \mathcal{K}(\boldsymbol{x}, \boldsymbol{w})$;
Compute the matrix $\hat{\Phi}' = (\hat{\Phi}'_{\boldsymbol{z}, \boldsymbol{w}})_{\boldsymbol{z}, \boldsymbol{w} \in S_{\text{ref}} \cup S'_{\text{ref}}}$ with $\hat{\Phi}'_{\boldsymbol{z}, \boldsymbol{w}} = \frac{1}{N} \sum_{\boldsymbol{x} \in S'_{\text{ver}}} \mathcal{K}(\boldsymbol{x}, \boldsymbol{z}) \mathcal{K}(\boldsymbol{x}, \boldsymbol{w})$;
Let $\rho$ be the value of the following eigenvalue problem

$$\max_{\boldsymbol{a} \in \mathbb{R}^{2m}} \boldsymbol{a}^\top \hat{\Phi}' \boldsymbol{a} \quad \text{s.t.} \quad \boldsymbol{a}^\top \hat{\Phi} \boldsymbol{a} \leq 1$$

**if** $\rho > 1 + \frac{\epsilon^2}{50AB}$ **then**
    | **Reject** and terminate;
Otherwise, **accept** and output $h : \boldsymbol{x} \mapsto h(\boldsymbol{x}) = \text{cl}_M(\hat{p}(\boldsymbol{x}))$, where $\hat{p}(\boldsymbol{x}) = \sum_{\boldsymbol{z} \in S_{\text{ref}}} \hat{a}_{\boldsymbol{z}} \mathcal{K}(\boldsymbol{z}, \boldsymbol{x})$;

---

**Testing the quality of reference feature map on the test distribution.** We know that the function $\tilde{p}^* = (\boldsymbol{a}^*)^\top \phi$ performs well on the test distribution (since it is close to $f^*$ on a reference test set). We also know that the candidate output $\hat{\boldsymbol{a}}^\top \phi$ performs well on the training distribution. Therefore, in order to ensure that $\hat{p}$ performs well on the test distribution, it suffices to show that the distance between $\hat{p}$ and $\tilde{p}^*$ under the test distribution, i.e., $\|\hat{\boldsymbol{a}}^\top \phi - (\boldsymbol{a}^*)^\top \phi\|_{\mathcal{D}'_{\boldsymbol{x}}}$, is small. In fact, it suffices to bound this distance by the corresponding one under the training distribution, because $\hat{p}$ fits the training data well and $\|\hat{\boldsymbol{a}}^\top \phi - (\boldsymbol{a}^*)^\top \phi\|_{\mathcal{D}_{\boldsymbol{x}}}$ is indeed small. Since we do not know $\boldsymbol{a}^*$, we need to run a test on $\phi$ that certifies the desired bound for any possible $\boldsymbol{a}^*$.

**Using the spectral tester.** We observe that $\|\hat{\boldsymbol{a}}^\top \phi - (\boldsymbol{a}^*)^\top \phi\|_{\mathcal{D}_{\boldsymbol{x}}}^2 = (\hat{\boldsymbol{a}} - \boldsymbol{a}^*)^\top \Phi (\hat{\boldsymbol{a}} - \boldsymbol{a}^*)$, where $\Phi = \mathbb{E}_{\boldsymbol{x} \sim \mathcal{D}_{\boldsymbol{x}}}[\phi(\boldsymbol{x}) \phi(\boldsymbol{x})^\top]$ and similarly $\|\hat{\boldsymbol{a}}^\top \phi - (\boldsymbol{a}^*)^\top \phi\|_{\mathcal{D}'_{\boldsymbol{x}}}^2 = (\hat{\boldsymbol{a}} - \boldsymbol{a}^*)^\top \Phi' (\hat{\boldsymbol{a}} - \boldsymbol{a}^*)$. Since we want to obtain a bound for all $\boldsymbol{a}^*$, we essentially want to ensure that for any $\boldsymbol{a} \in \mathbb{R}^{2m}$ we have $\boldsymbol{a}^\top \Phi' \boldsymbol{a} \leq (1 + \rho) \boldsymbol{a}^\top \Phi \boldsymbol{a}$ for some small $\rho$. Having a multiplicative bound is important because we do not have any bound on the norm of $\|\hat{\boldsymbol{a}} - \boldsymbol{a}^*\|_2$.

To implement the test, and since we cannot test $\Phi$ and $\Phi'$ directly, we draw fresh verification examples $S_{\text{ver}}, S'_{\text{ver}}$ from $\mathcal{D}_{\boldsymbol{x}}$ and $\mathcal{D}'_{\boldsymbol{x}}$ and run a spectral test on the corresponding empirical versions $\hat{\Phi}, \hat{\Phi}'$ of the matrices $\Phi, \Phi'$. To ensure that the test will accept when there is no distribution shift, we use the following lemma (originally from Goel et al. (2024)) on multiplicative spectral concentration for $\hat{\Phi}$, where the hypercontractivity assumption (Definition 3.4) is important.

**Lemma 3.8** (Multiplicative Spectral Concentration, Lemma B.1 in Goel et al. (2024), modified)**.** *Let $\mathcal{D}_{\boldsymbol{x}}$ be a distribution over $\mathbb{R}^d$ and $\phi : \mathbb{R}^d \to \mathbb{R}^m$ such that $\mathcal{D}_{\boldsymbol{x}}$ is $(\phi, C, \ell)$-hypercontractive for some $C, \ell \geq 1$. Suppose that $S$ consists of $N$ i.i.d. examples from $\mathcal{D}_{\boldsymbol{x}}$ and let $\Phi = \mathbb{E}_{\boldsymbol{x} \sim \mathcal{D}_{\boldsymbol{x}}}[\phi(\boldsymbol{x}) \phi(\boldsymbol{x})^\top]$, and $\hat{\Phi} = \frac{1}{N} \sum_{\boldsymbol{x} \in S} \phi(\boldsymbol{x}) \phi(\boldsymbol{x})^\top$. For any $\epsilon, \delta \in (0, 1)$, if $N \geq \frac{64Cm^2}{\epsilon^2} (4C \log_2(\frac{4}{\delta}))^{4\ell+1}$, then with probability at least $1 - \delta$, we have that*

$$\text{For any } \boldsymbol{a} \in \mathbb{R}^m : \boldsymbol{a}^\top \hat{\Phi} \boldsymbol{a} \in [(1-\epsilon) \boldsymbol{a}^\top \Phi \boldsymbol{a}, (1+\epsilon) \boldsymbol{a}^\top \Phi \boldsymbol{a}]$$

Note that the multiplicative spectral concentration lemma requires access to independent samples. However, the reference feature map $\phi$ depends on the reference examples $S_{\text{ref}}, S'_{\text{ref}}$. This is the reason why we do not reuse $S_{\text{ref}}, S'_{\text{ref}}$, but rather draw fresh verification examples.

For the full formal proof of Theorem 3.6 as well as a proof of Lemma 3.8, see Appendix B. The full proof involves appropriate uniform convergence bounds for kernel hypotheses, which are important in order to shift from the reference to the verification examples and back.

## 3.2 APPLICATIONS

Having obtained a general theorem for TDS learning under Assumption 3.5, we will now instantiate it to obtain TDS learning algorithms for learning neural networks with Lipschitz activations. In particular, we recover all of the bounds of Goel et al. (2017), using the additional assumption that the training distribution is hypercontractive in the following standard sense.

**Definition 3.9** (Hypercontractivity). We say that $\mathcal{D}$ is $C$-hypercontractive if for all polynomials of degree $\ell$ and $t \in \mathbb{N}$, we have that

$$\mathbb{E}_{\boldsymbol{x} \sim \mathcal{D}} \left[ p(\boldsymbol{x})^{2t} \right] \leq (Ct)^{2\ell t} \left( \mathbb{E}_{\boldsymbol{x} \sim \mathcal{D}} \left[ p(\boldsymbol{x})^2 \right] \right)^t .$$

Note that many common distributions like log-concave or the uniform over the hypercube are known to be hypercontractive for some constant $C$ (see Carbery & Wright (2001) and O'Donnell (2014)). We provide the following lemma, whose proof can be found in the appendix (see Theorems A.19 and A.21 and Lemma A.16).

**Lemma 3.10.** *The following bounds on the parameters in Assumption 3.5 hold for specific instantiations of the function classes.*

| Function Class | Degree ($\ell$) | Representation Bound ($B$) | Kernel Bound ($A$) |
|---|---|---|---|
| Sigmoid Nets | $O\left(RW^{t-2}(t\log(\frac{W}{\epsilon}))^{t-1}\log R\right)$ | $2^\ell \cdot W^{\tilde{O}(Wt\log(\frac{1}{\epsilon}))^{t-2}}$ | $(2R)^{2^t\ell}$ |
| $L$-Lipschitz Nets | $O\left((WL)^{t-1}Rk\sqrt{k}/\epsilon\right)$ | $(k+\ell)^{O(\ell)}$ | $R^{O(\ell)}$ |

Table 2: We instantiate the parameters relevant to Assumption 3.5 for Sigmoid and Lipschitz Nets. We have: (1) $t$ denotes a bound on the depth of the network, (2) $W$ is a bound on the sum of network weights in all layers other than the first, (3) $(\epsilon, B)$ and radius $R$ are the approximate representation parameters, (4) $k$ is the number of hidden units in the first layer. The kernel function can be evaluated in time $\text{poly}(d, \ell)$. For each of the classes, we assume that the maximum two norm of any row of the matrix corresponding to the weights of the first layer is bounded by 1. The kernel we use is the composed multinomial kernel $\mathsf{MK}_{\boldsymbol{\ell}}^{(t)}$ with appropriately chosen degree vector $\boldsymbol{\ell}$. Here, $\ell$ equals the product of the entries of $\boldsymbol{\ell}$. Any $C$-hypercontractive distribution is also $(\mathsf{MK}_{\boldsymbol{\ell}}^{(t)}, C, \ell)$ hypercontractive for $\ell$ as specified in the table. For the case of $k = 1$, the bound $B$ in the second row can be improved to $2^{O(\ell)}$.

Combining Lemma 3.10 with Theorem 3.6, we obtain the results of the middle column of Table 1.

## 4 UNBOUNDED DISTRIBUTIONS

We showed that the kernel method provides runtime improvements for TDS learning, because it can be used to obtain a concise reference feature map, whose spectral properties on the test distribution are all we need to check to certify low test error. A similar approach would not provide any runtime improvements for the case of unbounded distributions, because the dimension of the reference feature space would not be significantly smaller than the dimension of the multinomial feature expansion. Therefore, we can follow the standard moment-matching testing approach commonly used in TDS learning (Klivans et al., 2024a) and testable agnostic learning (Rubinfeld & Vasilyan, 2023; Gollakota et al., 2023). We require the following assumptions.

**Assumption 4.1.** *For a function class $\mathcal{F} \subseteq \{\mathbb{R}^d \to \mathbb{R}\}$, and training and test distributions $\mathcal{D}, \mathcal{D}'$ over $\mathbb{R}^d \times \mathbb{R}$, we assume the following.*

1. *For any $f \in \mathcal{F}$, there is $W \in \mathbb{R}^{k \times d}$ with $\|W\|_2 = 1$ and $WW^\top = I_k$ and a function $g : \mathbb{R}^k \to \mathbb{R}$ such that $f(\boldsymbol{x}) = g(W\boldsymbol{x})$ for all $\boldsymbol{x} \in \mathbb{R}^d$. Moreover, $f(0) = O(1)$.*

2. *For any $f \in \mathcal{F}$, with $f(\boldsymbol{x}) = g(W\boldsymbol{x})$, there is polynomial $q$ over $\mathbb{R}^k$ of degree at most $\ell$ s.t. for any $\boldsymbol{x} \in \mathbb{R}^d$ with $\|\boldsymbol{x}\|_2 \le R$ we have $|q(W\boldsymbol{x}) - g(W\boldsymbol{x})| \le \epsilon$, where $R \ge 1$, $\epsilon \in (0, 1)$. We also require that $\ell \le \tilde{O}_{\mathcal{F},\epsilon}(R)$, where $\tilde{O}_{\mathcal{F},\epsilon}$ is hiding factors that are at most logarithmic in $R$, but can also depend on $\epsilon, \mathcal{F}$.*

3. *The training marginal $\mathcal{D}_{\boldsymbol{x}}$ is $\gamma$-strictly subexponential for $\gamma \in (0, 1]$.*

4. *The training and test labels are both bounded in $[-M, M]$ for some $M \ge 1$.*

Consider the function class $\mathcal{F}$, and the parameters $\epsilon, \gamma, M, k, \ell$ as defined in the assumption above and let $\delta \in (0, 1)$. Then, we obtain the following theorem.

**Theorem 4.2** (TDS Learning via Uniform Approximation). *Under Assumption 4.1, Algorithm 2 learns the class $\mathcal{F}$ in the TDS regression setting up to excess error $5\epsilon$ and probability of failure $\delta$. The time complexity is $\mathrm{poly}(d^s, 1/\epsilon, \log(1/\delta)^\ell)$ where $s = (\ell \log(M/\epsilon))^{O(1/\gamma)}$.*

Note that Assumption 4.1 involves a low-degree uniform approximation assumption, which only holds within some bounded-radius ball. Since we work under unbounded distributions, we also need to handle the errors outside the ball. To this end, we use the following lemma, which follows from results in Ben-David et al. (2018).

**Lemma 4.3.** *Suppose $f = f_W$ and $q$ satisfy parts 1 and 2 of Assumption 4.1. Then*

$$|p(\boldsymbol{x})| \le (k\ell)^{O(\ell)} \|W\boldsymbol{x}\|_2^\ell, \text{ for all } \|W\boldsymbol{x}\|_2 \ge R.$$

The lemma above gives a bound on the values of a low-degree uniform approximator outside the interval of approximation. Therefore, we can hope to control the error of approximation outside the interval by taking advantage of the tails of our target distribution as well as picking $R$ sufficiently large. In order for the strictly subexponential tails to suffice, the quantitative dependence of $\ell$ on $R$ is important. This is why we assume (see Assumption 4.1) that $\ell = \tilde{O}(R)$. In particular, in order to bound the quantity $\mathbb{E}_{\boldsymbol{x} \sim \mathcal{D}_{\boldsymbol{x}}}[p^2(\boldsymbol{x})\mathbb{1}\{\|W\boldsymbol{x}\|_2 \ge R\}]$, we use Lemma 4.3 the Cauchy-Schwarz inequality and the bounds $\mathbb{E}_{\boldsymbol{x} \sim \mathcal{D}_{\boldsymbol{x}}}[\|W\boldsymbol{x}\|_2^{4\ell}] \le (k\ell)^{O(\ell)}$ and $\mathbf{Pr}_{\boldsymbol{x} \sim \mathcal{D}_{\boldsymbol{x}}}[\|W\boldsymbol{x}\|_2 \ge R] \le \exp(-\Omega(R/k)^{1+\gamma})$. Substituting for $\ell = \tilde{O}(R)$, we observe that the overall bound on the quantity $\mathbb{E}_{\boldsymbol{x} \sim \mathcal{D}_{\boldsymbol{x}}}[p^2(\boldsymbol{x})\mathbb{1}\{\|W\boldsymbol{x}\|_2 \ge R\}]$ decays with $R$, whenever $\gamma$ is strictly positive. Therefore, the overall bound can be made arbitrarily small with an appropriate choice of $R$ (and therefore $\ell$). For more details on the proof, see Appendix C. Apart from the careful manipulations described above, the proof follows the lines of the corresponding results for TDS learning through sandwiching polynomials (Klivans et al., 2024a).

In order to obtain end-to-end results for classes of neural networks (see the rightmost column of Table 1), we need to prove the existence of uniform polynomial approximators whose degree scales almost linearly with respect to the radius of approximation for the reasons described above. For arbitrary Lipschitz nets (see Theorem A.19), we use a general tool from polynomial approximation theory, the multivariate Jackson's theorem (Theorem A.9). This gives us a polynomial with degree scaling linearly in $R$ and polynomially on $\frac{1}{\epsilon}$ and the number of hidden units ($k$) in the first layer.

For sigmoid nets, a more careful derivation yields improved bounds (see Theorem A.21) which have a poly-logarithmic dependence on $\frac{1}{\epsilon}$. Our construction involves composing approximators for the activations at each layer. Naively, the degree of this composition would be super linear in $R$. To get around this, we use the key property that the size of the output of a sigmoid network at any layer is memoryless (i.e., has no $R$ dependence). This follows from the fact that the sigmoid is bounded in $[0, 1]$. Using this, we obtain an approximator with almost-linear dependence on $R$. For more details see Appendix A.5.

ACKNOWLEDGMENTS

We thank the anonymous reviewers of ICLR 2025 for their feedback. Gautam Chandrasekaran was supported by the NSF AI Institute for Foundations of Machine Learning (IFML). Adam R. Klivans was supported by NSF award AF-1909204 and the NSF AI Institute for Foundations of Machine Learning (IFML). Lin Lin Lee was supported by the NSF AI Institute for Foundations of Machine Learning (IFML). Konstantinos Stavropoulos was supported by the NSF AI Institute for Foundations of Machine Learning (IFML) and by scholarships from Bodossaki Foundation and Leventis Foundation.

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

# A  POLYNOMIAL APPROXIMATIONS OF NEURAL NETWORKS

## A.1  RESULTS FROM APPROXIMATION THEORY

We first introduce some definitions that we will use throughout the appendix.

**Definition A.1** (($\epsilon, R$)-Uniform Approximation)**.** For $\epsilon > 0, R \geq 1$, and $g : \mathbb{R}^k \to \mathbb{R}$, we say that $q : \mathbb{R}^k \to \mathbb{R}$ is an ($\epsilon, R$)-uniform approximation polynomial for $g$ if

$$|q(\boldsymbol{x}) - g(\boldsymbol{x})| \leq \epsilon \quad \forall \|x\|_2 \leq R.$$

**Definition A.2.** Let $\mathcal{F} \subseteq \{\mathbb{R}^d \to \mathbb{R}\}$ be a function class over $\mathbb{R}^d$. For $\ell, B > 0$, we say the ($\epsilon, R$)-uniform approximation degree of $\mathcal{F}$ is at most $\ell$ with coefficient bound $B$ if for any $f \in \mathcal{F}$, there is an ($\epsilon, R$)-uniform approximation polynomial $p(\boldsymbol{x})$ for $f$ such that $\deg(p) \leq \ell$ and each of the coefficients of $p$ are bounded in absolute value by $B$.

The following are useful facts about the coefficients of approximating polynomials.

**Fact A.3** (Lemma 23 from Goel et al. (2017))**.** *Let $p$ be a polynomial of degree $\ell$ such that $|p(x)| \leq b$ for $|x| \leq 1$. Then, the sum of squares of all its coefficients is at most $b^2 \cdot 2^{O(\ell)}$.*

**Lemma A.4.** *Let $p$ be a polynomial of degree $\ell$ such that $|p(\boldsymbol{x})| \leq b$ for $|x| \leq R$. Then, the sum of squares of all its coefficients is at most $b^2 \cdot 2^{O(\ell)}$ when $R \geq 1$.*

*Proof.* Consider $q(x) = p(Rx)$. Clearly, $|q(x)| \leq b$ for all $|x| \leq 1$. Thus, the sum of squares of its coefficients is at most $b^2 \cdot 2^{O(\ell)}$ from Fact A.3. Now, $p(x) = q(x/R)$ has coefficients bounded by $b^2 \cdot 2^{O(\ell)}$ when $R \geq 1$. $\qquad\square$

**Fact A.5** (Ben-David et al. (2018))**.** *Let $q$ be a polynomial with real coefficients on $k$ variables with degree $\ell$ such that for all $\boldsymbol{x} \in [0,1]^k$, $|q(\boldsymbol{x})| \leq 1$. Then the magnitude of any coefficient of $q$ is at most $(2k\ell(k+\ell))^\ell$ and the sum of the magnitudes of all coefficients of $q$ is at most $(2(k+\ell))^{3\ell}$.*

**Lemma A.6.** *Let $q$ be a polynomial with real coefficients on $k$ variables with degree $\ell$ such that for all $\boldsymbol{x} \in \mathbb{R}^k$ with $\|\boldsymbol{x}\|_2 \leq R$, $|q(\boldsymbol{x})| \leq b$. Then the sum of the magnitudes of all coefficients of $q$ is at most $b(2(k+\ell))^{3\ell}k^{\ell/2}$ for $R \geq 1$.*

*Proof.* Consider the polynomial $h(\boldsymbol{x}) = 1/b \cdot q(R\boldsymbol{x}/\sqrt{k})$. Then $|h(\boldsymbol{x})| = 1/b \cdot |q(R\boldsymbol{x}/\sqrt{k})| \leq 1$ for $\|\boldsymbol{x}R/\sqrt{k}\|_2 \leq R$, or equivalently for all $\|x\|_2 \leq \sqrt{k}$. In particular, since the unit cube $[0,1]^k$ is contained in the $\sqrt{k}$ radius ball, then $|h(\boldsymbol{x})| \leq 1$ for $\boldsymbol{x} \in [0,1]^k$. By Fact A.5, the sum of the magnitudes of the coefficients of $h$ is at most $(2(k+\ell))^{3\ell}$. Since $q(\boldsymbol{x}) = b \cdot h(\boldsymbol{x}\sqrt{k}/R)$, then the sum of the magnitudes of the coefficients of $q$ is at most $b(2(k+\ell))^{3\ell}k^{\ell/2}$. $\qquad\square$

**Lemma A.7.** *Let $p(\boldsymbol{x})$ be a degree $\ell$ polynomial in $\boldsymbol{x} \in \mathbb{R}^d$ such that each coefficient is bounded in absolute value by $b$. Then the sum of the magnitudes of the coefficients of $p(\boldsymbol{x})^t$ is at most $b^t d^{t\ell}$.*

In the following lemma, we bound the magnitude of approximating polynomials for subspace juntas outside the radius of approximation.

**Lemma A.8.** *Let $\epsilon > 0, R \geq 1$, and $f : \mathbb{R}^d \to \mathbb{R}$ be a $k$-subspace junta, and consider the corresponding function $g(W\boldsymbol{x})$. Let $q : \mathbb{R}^k \to \mathbb{R}$ be an ($\epsilon, R$)-uniform approximation polynomial for $g$, and define $p : \mathbb{R}^d \to \mathbb{R}$ as $p(\boldsymbol{x}) := q(W\boldsymbol{x})$. Let $r := \sup_{\|W\boldsymbol{x}\|_2 \leq R} |g(W\boldsymbol{x})|$. Then*

$$|p(\boldsymbol{x})| \leq (r + \epsilon)(2(k+\ell))^{3\ell}k^{\ell/2} \left\| \frac{W\boldsymbol{x}}{R} \right\|_2^\ell \quad \forall \|W\boldsymbol{x}\|_2 \geq R.$$

*Proof.* Since $q(\boldsymbol{x})$ is an ($\epsilon, R$)-uniform approximation for $g$, then $|q(\boldsymbol{x}) - g(\boldsymbol{x})| \leq \epsilon$ for $\|\boldsymbol{x}\|_2 \leq R$. Let $h(\boldsymbol{x}) = q(R\boldsymbol{x})$. Then $|h(\boldsymbol{x}/R) - g(\boldsymbol{x})| \leq \epsilon$ for $\|\boldsymbol{x}\|_2 \leq R$, and so $|h(\boldsymbol{x}/R)| \leq r + \epsilon$ for $\|\boldsymbol{x}\|_2 \leq R$, or equivalently $|h(\boldsymbol{x})| \leq r + \epsilon$ for $\|\boldsymbol{x}\|_2 \leq 1$. Write $h(\boldsymbol{x}) = \sum_{\|\alpha\|_1 \leq \ell} h_\alpha x_1^{\alpha_1} \ldots x_k^{\alpha_k}$.

By Lemma A.6, $\sum_{\|\alpha\|_1 \leq \ell} |h_\alpha| \leq (r + \epsilon)(2(k + \ell))^{3\ell} \cdot k^{\ell/2}$. Then for $\|x\|_2 \geq 1$,

$$
\begin{aligned}
|h(\boldsymbol{x})| &\leq \sum_{\|\alpha\|_1 \leq \ell} |h_\alpha||x_1^{\alpha_1} \ldots x_k^{\alpha_k}| \\
&\leq \sum_{\|\alpha\|_1 \leq \ell} |h_\alpha| \, \|\boldsymbol{x}\|_2^{\|\alpha\|_1} \\
&\leq \|\boldsymbol{x}\|_2^\ell \cdot \sum_{\|\alpha\|_1 \leq \ell} |h_\alpha|,
\end{aligned}
$$

where the second inequality holds because $|x_i| \leq \|\boldsymbol{x}\|_2$ for all $i$, and the last inequality holds because $\|\boldsymbol{x}\|_2^\ell \geq \|\boldsymbol{x}\|_2^{\|\alpha\|_1}$ for $\|\alpha\|_1 \leq \ell$ when $\|\boldsymbol{x}\|_2 \geq 1$. Then since $p(\boldsymbol{x}) = q(W\boldsymbol{x}) = h(W\boldsymbol{x}/R)$, we have $|p(\boldsymbol{x})| \leq \left\|\frac{W\boldsymbol{x}}{R}\right\|_2^\ell (r + \epsilon)(2(k + \ell))^{3\ell} k^{\ell/2}$ for $\|W\boldsymbol{x}\|_2 \geq R$. $\qquad \square$

*Proof.* Note that $p(\boldsymbol{x})$ has at most $d^\ell$ terms. Expanding $p(\boldsymbol{x})^t$ gives at most $d^{t\ell}$ terms, where any monomial is formed from a product of $t$ terms in $p(\boldsymbol{x})$. Then the coefficients of $p(\boldsymbol{x})^t$ are bounded in absolute value by $B^t$. Summing over all monomials gives the bound. $\qquad \square$

The following is an important theorem that we use later to obtain uniform approximators for Lipschitz Neural networks.

**Theorem A.9** (Newman & Shapiro (1964))**.** *Let $f : \mathbb{R}^k \to \mathbb{R}$ be a function continuous on the unit sphere $S_{k-1}$. Let $\omega_f$ be the function defined as $\omega_f(t) := \sup_{\substack{\|\boldsymbol{x}\|_2, \|\boldsymbol{y}\|_2 \leq 1 \\ \|\boldsymbol{x} - \boldsymbol{y}\|_2 \leq t}} |f(\boldsymbol{x}) - f(\boldsymbol{y})|$ for any $t \geq 0$. Then, we have that there exists a polynomial of degree $\ell$ such that $\sup_{\|x\|_2 \leq 1} |f(\boldsymbol{x}) - p(\boldsymbol{x})| \leq C \cdot \omega_f(k/\ell)$ where $C$ is a universal constant.*

This implies the following corollary.

**Corollary A.10.** *Let $f : \mathbb{R}^k \to \mathbb{R}$ be an $L$-Lipschitz function for $L \geq 0$ and let $R \geq 0$. Then, for any $\epsilon \geq 0$, there exists a polynomial $p$ of degree $O(LRk/\epsilon)$ such that $p$ is an $(\epsilon, R)$-uniform approximation polynomial for $f$.*

*Proof.* Consider the function $g(\boldsymbol{x}) := f(R\boldsymbol{x})$. Then, we have that $g$ is $RL$-Lipschitz. From statement of Theorem A.9, we have that $\omega_g(t) \leq RLt$. Thus, from Theorem A.9, there exists a polynomial $q$ of degree $O(LRk/\epsilon)$ such that $\sup_{\|\boldsymbol{y}\|_2 \leq 1} |g(\boldsymbol{y}) - q(\boldsymbol{y})| \leq \epsilon$. Thus, we have that $\sup_{\|\boldsymbol{x}\|_2 \leq R} |f(\boldsymbol{x}) - q(\boldsymbol{x}/R)| = \sup_{\|\boldsymbol{x}\|_2 \leq R} |g(\boldsymbol{x}/R) - q(\boldsymbol{x}/R)| = \sup_{\|\boldsymbol{y}\|_2 \leq 1} |g(\boldsymbol{y}) - q(\boldsymbol{y})| \leq \epsilon$. $p(\boldsymbol{x}) := q(\boldsymbol{x}/R)$ is the required polynomial of degree $O(LRk/\epsilon)$. $\qquad \square$

## A.2 USEFUL NOTATION AND FACTS

Given a univariate function $g$ on $\mathbb{R}$ and a vector $\boldsymbol{x} = (x_1, \ldots, x_d) \in \mathbb{R}^d$, the vector $g(\boldsymbol{x}) \in \mathbb{R}^d$ is defined as the vector with $i^{th}$ co-ordinate equal to $g(x_i)$. For a matrix $A \in \mathbb{R}^{m \times n}$, we use the following notation:

- $\|A\|_2 := \sup_{\|x\|_2 = 1} \|Ax\|_2$,

- $\|A\|_2^\infty := \sqrt{\max_{i \in [m]} \sum_{j=1}^n (A_{ij})^2}$,

- $\|A\|_1 := \sum_{(i,j) \in [n] \times [m]} |A_{ij}|$.

**Fact A.11.** *Given a matrix $W \in \mathbb{R}^{m \times n}$, we have that*

1. $\|A\|_2 \leq \|A\|_1$,

2. $\|A\|_2 \leq \sqrt{m} \cdot \|A\|_2^\infty$.

*Proof.* We first prove (1). We have that for an $\boldsymbol{x} \in \mathbb{R}^n$ with $\|\boldsymbol{x}\|_2 = 1$,

$$\|A\boldsymbol{x}\|_2 \leq \sqrt{\sum_{i=1}^{m}(A_i \cdot \boldsymbol{x})^2} \leq \sqrt{\sum_{i=1}^{m}\sum_{j=1}^{n}(A_{ij})^2} \leq \|A\|_1$$

where the second inequality follows from Cauchy Schwartz and the last inequality follows from the fact that for any vector $\boldsymbol{v}$, $\|\boldsymbol{v}\|_2 \leq \|\boldsymbol{v}\|_1$. We now prove (2). We have that

$$\|A\boldsymbol{x}\|_2 \leq \sqrt{\sum_{i=1}^{m}(A_i \cdot \boldsymbol{x})^2} \leq \sqrt{m \max_{i \in [m]}\sum_{j=1}^{n}(A_{ij})^2} \leq \sqrt{m}\|A\|_2^\infty$$

where the second inequality follows from Cauchy Schwartz and the last inequality is the definition. $\square$

Recall the definition of a neural network.

**Definition A.12** (Neural Network). Let $\sigma : \mathbb{R} \to \mathbb{R}$ be an activation function with $\sigma(0) \leq 1$. Let $\mathbf{W} = \left(W^{(1)}, \ldots W^{(t)}\right)$ with $W^{(i)} \in \mathbb{R}^{s_i \times s_{i-1}}$ be the tuple of weight matrices. Here, $s_0 = d$ is the input dimension and $s_t = 1$. Define recursively the function $f_i : \mathbb{R}^d \to \mathbb{R}^{s_i}$ as $f_i(\boldsymbol{x}) = W^{(i)} \cdot \sigma\left(f_{i-1}(\boldsymbol{x})\right)$ with $f_1(\boldsymbol{x}) = W^{(1)} \cdot \boldsymbol{x}$. The function $f : \mathbb{R}^d \to \mathbb{R}$ computed by the neural network $(\mathbf{W}, \sigma)$ is defined as $f(\boldsymbol{x}) := f_t(\boldsymbol{x})$. We denote $\|\mathbf{W}\|_1 = \sum_{i=2}^{t}\|W^{(i)}\|_1$. The depth of this network is $t$.

### A.3 KERNEL REPRESENTATIONS

We now state and prove facts about Kernel Representations that we require. First, we recall the multinomial kernel from Goel et al. (2017).

**Definition A.13.** Consider the mapping $\psi_\ell : \mathbb{R}^n \to \mathbb{R}^{N_\ell}$, where $N_d = \sum_{i=1}^{\ell} d^\ell$ indexed by tuples $(i_1, i_2, \ldots, i_j) \in [d]^j$ for $j \in [\ell]$ such that value of $\psi_\ell(\boldsymbol{x})$ at index $(i_1, i_2, \ldots, i_j)$ is equal to $\prod_{t=1}^{j} \boldsymbol{x}_{i_t}$. The kernel $\mathsf{MK}_\ell$ is defined as

$$\mathsf{MK}_\ell(\boldsymbol{x}, \boldsymbol{y}) = \langle \psi_\ell(\boldsymbol{x}), \psi_\ell(\boldsymbol{y}) \rangle = \sum_{i=1}^{d}(\boldsymbol{x} \cdot \boldsymbol{y})^i.$$

We denote the corresponding RKHS as $\mathbb{H}_{\mathsf{MK}_\ell}$.

We now prove that polynomial approximators of subspace juntas can be represented as elements of $\mathbb{H}_{\mathsf{MK}_\ell}$.

**Lemma A.14.** *Let $k \in \mathbb{N}$ and $\epsilon, R \geq 0$. Let $f : \mathbb{R}^d \to \mathbb{R}$ be a $k$-subspace junta such that $f(\boldsymbol{x}) = g(W\boldsymbol{x})$ where $g$ is a function on $\mathbb{R}^k$ and $W$ is a projection matrix from $\mathbb{R}^{k \times d}$. Suppose, there exists a polynomial $q$ of degree $\ell$ such that $\sup_{\|\boldsymbol{y}\|_2 \leq R}|g(\boldsymbol{y}) - q(\boldsymbol{y})| \leq \epsilon$ and the sum of squares of coefficients of $q$ is bounded above by $B^2$. Then, $f$ is $(\epsilon, B^2 \cdot (k+1)^\ell)$-approximately represented within radius $R$ with respect to $\mathbb{H}_{\mathsf{MK}_\ell}$.*

*Proof.* We argue that there exists a vector $\boldsymbol{v} \in \mathbb{H}_{\mathsf{MK}_\ell}$ such that $\langle \boldsymbol{v}, \boldsymbol{v} \rangle \leq B^2$ and $|f(\boldsymbol{x}) - \langle \boldsymbol{v}, \sigma_\ell(\boldsymbol{x}) \rangle| \leq \epsilon$ for all $\|\boldsymbol{x}\|_2 \leq R$. Consider the polynomial $p$ of degree $\ell$ such that $p(\boldsymbol{x}) = q(W\boldsymbol{x})$. We argue that $p(\boldsymbol{x}) = \langle \boldsymbol{v}, \sigma_\ell(\boldsymbol{x}) \rangle$ for some $\boldsymbol{v}$ and that $\langle \boldsymbol{v}, \boldsymbol{v} \rangle \leq B^2$. Let $q(\boldsymbol{y}) = \sum_{S \in \mathbb{N}^k, |S| \leq \ell} q_S \prod_{j=1}^{k} \boldsymbol{y}^{|S_j|}$. From our assumption on $q$, we have that $\sum_{S \in \mathbb{N}^k, |S| \leq \ell} |q_S| \leq B$. For $i \in \ell$, we use define $B_i$ as $B_i = \sum_{S \in \mathbb{N}^k, |S| = \ell} |q_S|$. Given multi-index $S$, for any $i \in [d]$, we define $S(i)$ as the number $t$ such that $\sum_{i=1}^{j-1}|S_i| \leq j < \sum_{i=1}^{j}|S_i|$. We now compute the entry of $\boldsymbol{v}$ indexed by $(i_1, i_2, \ldots, i_t)$. By expanding the expression for $p(\boldsymbol{x})$, we obtain that

$$v_{i_1, \ldots, i_t} = \sum_{|S|=t} q_S \prod_{j=1}^{t} W_{S(j), i_j}.$$

We are now ready to bound $\langle \boldsymbol{v}, \boldsymbol{v} \rangle$. We have that

$$
\begin{aligned}
\langle \boldsymbol{v}, \boldsymbol{v} \rangle &= \sum_{t=0}^{\ell} \sum_{(i_1,\ldots,i_t) \in [d]^k} (v_{i_1,\ldots,i_t})^2 = \sum_{t=0}^{\ell} \sum_{(i_1,\ldots,i_t) \in [d]^k} \left( \sum_{|S|=t} q_S \prod_{j=1}^{t} W_{S(j),i_j} \right)^2 \\
&\leq \sum_{t=0}^{\ell} \sum_{(i_1,\ldots,i_t) \in [d]^k} \left( \sum_{|S|=t} q_S^2 \right) \left( \sum_{|S|=t} \prod_{j=1}^{t} W_{S(j),i_j}^2 \right) \\
&\leq \sum_{t=0}^{\ell} \left( \sum_{|S|=t} q_S^2 \right) \left( \sum_{|S|=t} \prod_{j=1}^{t} \left( \sum_{i=1}^{d} W_{S(j),i}^2 \right) \right) \leq \sum_{t=0}^{\ell} \left( \sum_{|S|=t} q_S^2 \right) \cdot (k+1)^t \\
&\leq \left( \sum_{|S| \leq \ell} q_S^2 \right) \cdot (k+1)^\ell \leq B^2 \cdot (k+1)^\ell.
\end{aligned}
$$

Here, the first inequality follows from Cauchy-Schwartz, the second follows by rearranging terms. The third inequality follows from the fact that the number of multi-indices of size $t$ from a set of $k$ elements is at most $(k+1)^t$. The final inequality follows from the fact that the sum of the squares of the coefficients of $q$ is at most $B^2$. $\qquad\square$

We introduce an extension of the multinomial kernel that will be useful for our application to sigmoid-nets.

**Definition A.15** (Composed multinomial kernel). Let $\boldsymbol{\ell} = (\ell_1, \ldots, \ell_t)$ be a tuple in $\mathbb{N}^t$. We denote a sequence of mappings $\psi_{\boldsymbol{\ell}}^{(0)}, \psi_{\boldsymbol{\ell}}^{(1)}, \ldots, \psi_{\boldsymbol{\ell}}^{(t)}$ on $\mathbb{R}^d$ inductively as follows:

1. $\psi_{\boldsymbol{\ell}}^{(0)}(\boldsymbol{x}) = \boldsymbol{x}$

2. $\psi_{\boldsymbol{\ell}}^{(i)}(\boldsymbol{x}) = \psi_{\ell_i}\left( \psi_{\boldsymbol{\ell}}^{(i-1)}(\boldsymbol{x}) \right).$

Let $N_{\boldsymbol{\ell}}^{(i)}$ denote the number of coordinates in $\psi_{\boldsymbol{\ell}}^{(i)}$. This induces a sequence of kernels $\mathsf{MK}_{\boldsymbol{\ell}}^{(0)}, \mathsf{MK}_{\boldsymbol{\ell}}^{(1)}, \ldots, \mathsf{MK}_{\boldsymbol{\ell}}^{(t)}$ defined as

$$
\mathsf{MK}_{\boldsymbol{\ell}}^{(i)}(\boldsymbol{x}, \boldsymbol{y}) = \langle \psi_{\boldsymbol{\ell}}^{(i)}(\boldsymbol{x}), \psi_{\boldsymbol{\ell}}^{(i)}(\boldsymbol{y}) \rangle = \sum_{j=0}^{\ell_i} \left( \langle \psi_{\boldsymbol{\ell}}^{(i-1)}(\boldsymbol{x}), \psi_{\boldsymbol{\ell}}^{(i-1)}(\boldsymbol{y}) \rangle^j \right)
$$

and a corresponding sequence of RKHS denoted by $\mathcal{H}_{\mathsf{MK}_{\boldsymbol{\ell}}^{(0)}}, \mathcal{H}_{\mathsf{MK}_{\boldsymbol{\ell}}^{(1)}}, \ldots \mathcal{H}_{\mathsf{MK}_{\boldsymbol{\ell}}^{(t)}}$.

Observe that the the multinomial Kernel $\mathsf{MK}_{\ell} = \mathsf{MK}_{(\ell)}^{(1)}$ is an instantiation of the composed multinomial kernel.

We now state some properties of the composed multinomial kernel.

**Lemma A.16.** Let $\boldsymbol{\ell} = (\ell_1, \ldots, \ell_t)$ be a tuple in $\mathbb{N}^t$ and $R \geq 0$. Then, the following hold:

1. $\sup_{\|\boldsymbol{x}\|_2 \leq R} \mathsf{MK}_{\boldsymbol{\ell}}^{(t)}(\boldsymbol{x}, \boldsymbol{x}) \leq \max\{1, (2R)^{2^t \prod_{i=1}^{t} \ell_i}\}$,

2. For any $\boldsymbol{x}, \boldsymbol{y} \in \mathbb{R}^d$, $\mathsf{MK}_{\boldsymbol{\ell}}^{(t)}(\boldsymbol{x}, \boldsymbol{y})$ can be computed in time $\text{poly}\left( d, (\sum_{i=1}^{t} \ell_i) \right)$,

3. For any $\boldsymbol{v} \in \mathcal{H}_{\mathsf{MK}_{\boldsymbol{\ell}}^{(t)}}$ and $\boldsymbol{x} \in \mathbb{R}^d$, we have $\langle \boldsymbol{v}, \psi_{\boldsymbol{\ell}}^{(t)}(\boldsymbol{x}) \rangle$ is a polynomial in $\boldsymbol{x}$ of degree $\prod_{i=1}^{t} \ell_i$.

*Proof.* We assume without loss of generality that $R \geq 1$ as the kernel function is increasing in norm. To prove (1), observe that for any $\boldsymbol{x}$, we have that

$$
\mathsf{MK}_{\boldsymbol{\ell}}^{(i)}(\boldsymbol{x}, \boldsymbol{x}) = \sum_{j=0}^{\ell_i} \left( \mathsf{MK}_{\boldsymbol{\ell}}^{(i-1)}(\boldsymbol{x}, \boldsymbol{x}) \right)^j \leq \left( 2\mathsf{MK}_{\boldsymbol{\ell}}^{(i-1)}(\boldsymbol{x}, \boldsymbol{x}) \right)^{\ell_i + 1}.
$$

We also have that $\sup_{\|x\|_2 \le R} \mathsf{MK}_{\boldsymbol{\ell}}^{(0)}(\boldsymbol{x}, \boldsymbol{x}) = \boldsymbol{x} \cdot \boldsymbol{x} = R$. Thus, unrolling the recurrence gives us $\mathsf{MK}_{\boldsymbol{\ell}}^{(t)}(\boldsymbol{x}, \boldsymbol{x}) \le \max\{1, (2R)^{\prod_{i=1}^{t}(\ell_i+1)}\} \le \max\{1, (2R)^{2^t \prod_{i=1}^t \ell_i}\}$.

The run time follows from the fact that $\mathsf{MK}_{\boldsymbol{\ell}}^{(i)}(\boldsymbol{x}, \boldsymbol{x}) = \sum_{j=0}^{\ell_i} \left( \mathsf{MK}_{\boldsymbol{\ell}}^{(i-1)}(\boldsymbol{x}, \boldsymbol{x})^j \right)$ and thus can be computed from $\mathsf{MK}_{\boldsymbol{\ell}}^{(i-1)}$ with $\ell_i$ additions and exponentiation operations. Recursing gives the final runtime.

The fact that $\langle \boldsymbol{v}, \psi_{\boldsymbol{\ell}}^{(i)}(\boldsymbol{x}) \rangle$ follows immediately from the fact the fact the entries of $\psi_{\boldsymbol{\ell}}^{(i)}(\boldsymbol{x})$ arise from the multinomial kernel and hence are polynomials in $\boldsymbol{x}$. The degree is at most $\prod_{i=1}^t \ell_i$. $\qquad\square$

We now argue that a distribution that is hypercontractive with respect to polynomials is hypercontractive with respect to the multinomial kernel.

**Lemma A.17.** *Let $\mathcal{D}$ be a distribution on $\mathbb{R}^d$ that is $C$-hypercontractive for some constant $C$. Then, $\mathcal{D}$ is also $(\mathsf{MK}_{\boldsymbol{\ell}}^{(t)}, C, \prod_{i=1}^t \ell_i)$-hypercontractive.*

*Proof.* The proof immediately follows from Definition 3.4 and Lemma A.16(3). $\qquad\square$

### A.4 Nets with Lipschitz activations

We are now ready to prove our theorem about uniform approximators for neural networks with Lipschitz activations. First, we prove that such networks describe a Lipschitz function.

**Lemma A.18.** *Let $f : \mathbb{R}^d \to \mathbb{R}$ be the function computed by an $t$-layer neural network with $L$-Lipschitz activation function $\sigma$ and weight matrices $\mathbf{W}$. Say, $\|\mathbf{W}\|_1 \le W$ for $W \ge 0$ and the first hidden layer has $k$ neurons. Then we have that $f$ is $\sqrt{k}\|W^{(1)}\|_2^\infty (WL)^{t-1}$-Lipschitz.*

*Proof.* First, observe from Fact A.11 that for all $1 < i \le T$, $\|W^{(i)}\|_2 \le W$ (since $\|\mathbf{W}\|_1 \le W$) and $\|W^{(1)}\|_2 \le \sqrt{k}\|W^{(1)}\|_2^\infty$. Recall from Definition A.12, we have the functions $f_1, \ldots, f_t$ where $f_i(\boldsymbol{x}) = W^{(i)} \cdot \sigma\big(f_{i-1}(\boldsymbol{x})\big)$ and $f_1(\boldsymbol{x}) = W^{(1)} \cdot \boldsymbol{x}$. We prove by induction on $i$ that $\|f_i(\boldsymbol{x}) - f_i(\boldsymbol{x} + \boldsymbol{u})\|_2 \le \sqrt{k}\|W^{(1)}\|_2^\infty (WL)^{i-1}\|\boldsymbol{u}\|_2$. For the base case, observe that

$$\|f_1(\boldsymbol{x} + \boldsymbol{u}) - f_1(\boldsymbol{x})\|_2 \le \sqrt{\sum_{i=1}^{d_1} \Big( \big( \langle W_i^{(1)}, \boldsymbol{x}\rangle - \langle W_i^{(1)}, \boldsymbol{x} + \boldsymbol{u}\rangle \big)^2 \Big)} \le \sqrt{\sum_{i=1}^{d_1} \Big( \langle W_i^{(1)}, \boldsymbol{u}\rangle \Big)^2}$$

$$\le \|W_i^{(1)}\boldsymbol{u}\|_2 \le \sqrt{k}\|W^{(1)}\|_2^\infty \|\boldsymbol{u}\|_2$$

where the second inequality follows from the Lipschitzness of $\sigma$ and the final inequality follows from the definition of operator norm. We now proceed to the inductive step. Assume by induction that $\|f_i(\boldsymbol{x}) - f_i(\boldsymbol{x} + \boldsymbol{u})\|_2$ is at most $\sqrt{k}\|W^{(1)}\|_2^\infty (WL)^{i-1}\|\boldsymbol{u}\|_2$. Thus, we have

$$\|f_{i+1}(\boldsymbol{x} + \boldsymbol{u}) - f_{i+1}(\boldsymbol{x})\|_2 = \sqrt{\sum_{j=1}^{d_1} \Big( \langle W_j^{(i+1)}, \sigma\left(f_i(\boldsymbol{x})\right)\rangle - \langle W_j^{(i+1)}, \sigma\left(f_i(\boldsymbol{x} + \boldsymbol{u})\right)\rangle \Big)^2}$$

$$\le \|W^{(i+1)}\|_2 \|\sigma(f_i(\boldsymbol{x})) - \sigma(f_i(\boldsymbol{x} + \boldsymbol{u}))\|_2$$

$$\le (WL)\sqrt{k}\|W^{(1)}\|_2^\infty (WL)^{i-1}\|\boldsymbol{u}\|_2 \le \sqrt{k}\|W^{(1)}\|_2^\infty (LW)^i\|\boldsymbol{u}\|_2$$

where the third inequality follows from the Lipschitzness of $\sigma$ and the inductive hypothesis. Thus, we get that $|f(\boldsymbol{x} + \boldsymbol{u}) - f(\boldsymbol{x})| \le \|f_t(\boldsymbol{x} + \boldsymbol{u}) - f_t(\boldsymbol{x})\|_2 \le \sqrt{k}\|W^{(1)}\|_2^\infty (WL)^{t-1} \cdot \|\boldsymbol{u}\|_2$. $\qquad\square$

We now state are theorem regarding the uniform approximation of Lipschitz nets. We also prove that the approximators can be represented by low norm vectors in $\mathcal{R}_{\mathsf{MK}_{\boldsymbol{\ell}}}$ for appropriately chosen degree $\ell$.

**Theorem A.19.** *Let $\epsilon, R \ge 0$. Let $f$ on $\mathbb{R}^d$ be a neural network with an $L$-Lipschitz activation function $\sigma$, depth $t$ and weight matrices $\mathbf{W} = (W^{(1)}, \ldots, W^{(t)})$ where $W^{(i)} \in \mathbb{R}^{s_i \times s_{i-1}}$. Let*

*k be the number of neurons in the first hidden layer. Then, there exists of a polynomial $p$ of degree $\ell = O\left(\|W^{(1)}\|_2^\infty (WL)^{t-1} Rk\sqrt{k}/\epsilon\right)$ that is an $(\epsilon, R)$-uniform approximation polynomial for $f$. Furthermore, $f$ is $(\epsilon, (k+\ell)^{O(\ell)})$-approximately represented within radius $R$ with respect to $\mathbb{H}_{\mathsf{MK}_\ell} = \mathbb{H}_{\mathsf{MK}_{(\ell)}^{(1)}}$. In fact, when $k = 1$, it holds that $f$ is $(\epsilon, 2^{O(\ell)})$-approximately represented within $R$ with respect to $\mathbb{H}_{\mathsf{MK}_\ell^{(1)}}$.*

*Proof.* We can express $f$ as $f(\boldsymbol{x}) = g(P\boldsymbol{x})$ where $P$ is a projection matrix and $g$ is a neural network with input size $k$. We observe that the Lipschitz constant of $g$ is the same as the Lipschitz constant of $f$ since $P$ is a projection matrix. From Lemma A.18, we have that $g$ is $\|\sqrt{k}W^{(1)}\|_2^\infty (WL)^{t-1}$-Lipshitz. From Corollary A.10, we have that there exists a polynomial $q$ of degree $\ell = O\left(\|W^{(1)}\|_2^\infty (WL)^{t-1} Rk\sqrt{k}/\epsilon\right)$ that is an $(\epsilon, R)$-uniform approximation for $g$. From Lemma A.6, we have that the sum of squares of magnitudes of coefficients of $q$ is bounded by $\left(\|\sqrt{k}W^{(1)}\|_2^\infty (WL)^{t-1}R\right)(k+\ell)^{O(\ell)} \le (k+\ell)^{O(\ell)}$. Now, applying Lemma A.14 yields the result. When $k = 1$, we apply Lemma A.4 to obtain that the sum of squares of magnitudes of coefficients of $q$ is bounded by $\|W^{(1)}\|_2^\infty (WL)^{t-1} \cdot 2^{O(\ell)} \le 2^{O(\ell)}$. $\square$

## A.5 SIGMOIDS AND SIGMOID-NETS

We now give a custom proof for the case of neural networks with sigmoid activation. We do this as we can hope to get $O(\log(1/\epsilon))$ degree for our polynomial approximation. We largely follow the proof technique of Goel et al. (2017) and Zhang et al. (2016a). The modifications we make are to handle the case where the radius of approximation is a variable $R$ instead of a constant. We require(for our applications to strictly-subexponential distributions) that the degree of approximation must scale linear in $R$, a property that does not follow directly from the analysis given in Goel et al. (2017). We modify their analysis to achieve this linear dependence.

We first state a result regarding polynomial approximations for a single sigmoid activation.

**Theorem A.20** (Livni et al. (2014))**.** *Let $\sigma : \mathbb{R} \to \mathbb{R}$ denote the function $\sigma(x) = \frac{1}{1+e^{-x}}$. Let $R, \epsilon \ge 0$. Then, there exists a polynomial $p$ of degree $\ell = O(R\log(R/\epsilon))$ such that $\sup_{|x| \le R} |\sigma(x) - p(x)| \le \epsilon$. Also, the sum of the squares of the coefficients of $p$ is bounded above by $2^{O(\ell)}$.*

We now present a construction of a uniform approximation for neural networks with sigmoid activations. The construction is similar to the one in Goel et al. (2017) but the analysis deviates as linear dependence on radius of approximation is important to us.

**Theorem A.21.** *Let $\epsilon, R \ge 0$. Let $f$ on $\mathbb{R}^d$ be a neural network with sigmoid activations, depth $t$ and weight matrices $\mathbf{W} = (W^{(1)}, \dots, W^{(t)})$ where $W^{(i)} \in \mathbb{R}^{s_i \times s_{i-1}}$. Also, let $\|\mathbf{W}\|_1 \le W$. Then, there exists of a polynomial $p$ of degree $\ell = O\left((R\log R) \cdot (\|W^{(1)}\|_2^\infty W^{t-2}) \cdot (t\log(W/\epsilon))^{t-1}\right)$ that is an $(\epsilon, R)$-uniform approximation polynomial for $f$. Furthermore, $f$ is $(\epsilon, B)$-approximately represented within radius $R$ with respect to $H_{\mathsf{MK}_\ell^{(t)}}$ where $\boldsymbol{\ell} = (\ell_1, \dots, \ell_{t-1})$ is a tuple of degrees whose product is bounded by $\ell$. Here, $B \le (2\|W^{(1)}\|_2^\infty)^\ell \cdot W^{O\left(W^{t-2}(t\log(W/\epsilon))^{t-2}\right)}$.*

*Proof.* First, let $q_1$ be the polynomial guaranteed by Theorem A.20 that $(\epsilon/(2W)^t)$-approximates the sigmoid in an interval of radius $R\|W^{(1)}\|_2^\infty$. Denote the degree of $q_1$ as $\ell_1 = O\left(Rt\|W^{(1)}\|_2^\infty \log(RW/\epsilon)\right)$. For all $1 < i < t$, let $q_i$ be the polynomial that $(\epsilon/(2W)^t)$-approximates the sigmoid upto radius $2W$. These have degree equal to $O\left(Wt\log(W/\epsilon)\right)$. Let $\boldsymbol{\ell} = (\ell_1, \dots \ell_{t-1})$. For all $i \in [t-1]$, let $q_i(x) = \sum_{j=0}^{\ell_i} \beta_j^{(i)} x^j$. We know that $\sum_{i=0}^{\ell_i} (\beta_j^{(i)})^2 \le 2^{O(\ell_i)}$.

We now construct the polynomial $p$ that approximates $f$. For $i \in [t]$, define $p_i(\boldsymbol{x}) = W^{(i)} \cdot q_{i-1}(p_{i-1}(\boldsymbol{x}))$ with $p_1(\boldsymbol{x}) = W^{(1)} \cdot \boldsymbol{x}$. Define $p(\boldsymbol{x}) = p_t(\boldsymbol{x})$. Recall that $p_i(\boldsymbol{x})$ is a vector of $s_i$ polynomials. We prove the following by induction: for every $i \in [t]$,

1. $\|p_i(\boldsymbol{x}) - f_i(\boldsymbol{x})\|_\infty \le \epsilon/(2W)^{t-i}$,

2. For each $j \in [s_i]$, we have that $(p_i)_j(\boldsymbol{x}) = \langle \boldsymbol{v}, \psi_{\boldsymbol{\ell}}^{(i)}(\boldsymbol{x}) \rangle$ with $\langle \boldsymbol{v}, \boldsymbol{v} \rangle \leq (2\|W^{(1)}\|_2^\infty)^{O(\prod_{n=1}^{i-1} \ell_n)} \cdot W^{O(\prod_{n=2}^{i-1} \ell_n)}$.

where the function $f_i$ is as defined in [Definition A.12](#).

The above holds trivially for $i = 1$ and $f_1(\boldsymbol{x}) = p_1(\boldsymbol{x}) = W^{(1)} \cdot (\boldsymbol{x})$ is an exact approximator. Also, $(p_1)_i(\boldsymbol{x}) = \langle W_i^{(1)}, \boldsymbol{x} \rangle = \langle W_i^{(1)}, \psi_{\boldsymbol{\ell}}^{(1)}(x) \rangle$ from the definition of $\psi_{\boldsymbol{\ell}}^{(1)}$. Clearly, $\langle W_i^{(1)}, W_i^{(1)} \rangle \leq \left( \|W^{(1)}\|_2^\infty \right)^2$. We now prove that the above holds for $i + 1 \in [t]$ assuming it holds for $i$.

We first prove (1). For $j \in [s_{i+1}]$, we have that

$$|(p_{i+1})_j(\boldsymbol{x}) - (f_{i+1})_j(\boldsymbol{x})| = |W_j^{(i+1)}(q_i(p_i(\boldsymbol{x})) - \sigma(f_i(\boldsymbol{x})))|$$
$$\leq |W_j^{(i+1)}(q_i(p_i(\boldsymbol{x})) - \sigma(p_i(\boldsymbol{x})))| + |W_j^{(i+1)}(\sigma(p_i(\boldsymbol{x})) - \sigma(f_i(\boldsymbol{x})))|$$
$$\leq W \cdot (\epsilon/(2W)^t) + W \cdot \epsilon/(2W)^{t-i} \leq \epsilon/(2W)^{t-(i+1)}.$$

For the second inequality, we analyse the cases $i = 1$ and $i > 1$ separately. When $i = 1$, we have that $(p_1)_j(\boldsymbol{x}) = (f_1)_j(\boldsymbol{x}) \leq R\|W_1\|_2^\infty$ and $\sigma(x) - q_1(x) \leq (\epsilon/(2W)^t)$ when $|x| \leq R\|W_1\|_2^\infty$. For $i > 1$, from the inductive hypothesis, we have that $|W^{(i+1)}p_i(\boldsymbol{x})| \leq |W^{(i+1)}f_i(\boldsymbol{x})| + \|W^{(i+1)}\|_1 \cdot (\epsilon/(2W)^{t-i}) \leq 2W$. The second term in the second inequality is bounded since $\sigma$ is 1-Lipschitz.

We are now ready to prove that $(p_{i+1})_j$ is representable by small norm vectors in $\mathcal{H}_{\mathsf{MK}_{\boldsymbol{\ell}}^{(i+1)}}$ for all $j \in [s_{j+1}]$. We have that

$$(p_{i+1})_j(\boldsymbol{x}) = \sum_{k=1}^{s_i} W_{jk}^{(i+1)} \cdot q_i \left( (p_i)_k(\boldsymbol{x}) \right).$$

From the inductive hypothesis, we have that $(p_i)_k = \langle \boldsymbol{v}^{(k)}, \psi_{\boldsymbol{\ell}}^{(i)} \rangle$. Thus, we have that

$$(p_{i+1})_j(\boldsymbol{x}) = \sum_{k=1}^{s_i} W_{jk}^{(i+1)} \cdot q_i \left( \langle \boldsymbol{v}^{(k)}, \psi_{\boldsymbol{\ell}}^{(i)} \rangle \right).$$

We expand each term in the above sum. We obtain,

$$q_i \left( \langle \boldsymbol{v}^{(k)}, \psi_{\boldsymbol{\ell}}^{(i)} \rangle \right) = \sum_{n=0}^{\ell_i} \beta_n^{(i)} \left( \langle \boldsymbol{v}^{(k)}, \psi_{\boldsymbol{\ell}}^{(i)} \rangle \right)^n$$
$$= \sum_{n=0}^{\ell_i} \beta_n^{(i)} \sum_{(m_1,\ldots,m_n) \in [N_{\boldsymbol{\ell}}^{(i)}]^n} v_{m_1}^{(k)} \ldots v_{m_n}^{(k)} \left( \psi_{\boldsymbol{\ell}}^{(i)}(\boldsymbol{x}) \right)_{m_1} \ldots \left( \psi_{\boldsymbol{\ell}}^{(i)}(\boldsymbol{x}) \right)_{m_n}$$
$$= \langle \boldsymbol{u}^{(k)}, \psi_{\ell_i}((\psi_{\boldsymbol{\ell}}^{(i)}(\boldsymbol{x}))) \rangle = \langle \boldsymbol{u}^{(k)}, \psi_{\boldsymbol{\ell}}^{(i+1)}(\boldsymbol{x}) \rangle.$$

The second inequality follows from expanding the equation. $\boldsymbol{u}^{(k)}$ indexed by $(m_1, \ldots, m_n) \in [N_{\boldsymbol{\ell}}^{(i)}]^n$ for $n \leq \ell_i$ has entries given by $u_{(m_1,\ldots,m_n)}^{(k)} = \beta_n^{(i)} v_{m_1}^{(k)} \ldots v_{m_n}^{(k)}$. Putting things together, we obtain that

$$(p_{i+1})_j(\boldsymbol{x}) = \sum_{k=1}^{s_i} W_{jk}^{(i+1)} \cdot \langle \boldsymbol{u}^{(k)}, \psi_{\boldsymbol{\ell}}^{(i+1)}(\boldsymbol{x}) \rangle$$
$$= \langle \sum_{k=1}^{s_i} W_{jk}^{(i+1)} \boldsymbol{u}^{(k)}, \psi_{\boldsymbol{\ell}}^{(i+1)}(\boldsymbol{x}) \rangle.$$

Thus, we have proved that $(p_{i+1})_j$ is representable in $\mathcal{H}_{\mathsf{MK}_{\boldsymbol{\ell}}^{(i+1)}}$. We now prove that the norm of the representation is small. We have that

$$\| \sum_{k=1}^{s_i} W_{jk}^{(i+1)} \boldsymbol{u}^{(k)} \|_2 \leq \|W^{(i+1)}\|_1 \max_{k \in [s_i]} \|\boldsymbol{u}^{(k)}\|_2 \leq W \cdot \max_{k \in [s_i]} \|\boldsymbol{u}^{(k)}\|_2.$$

We bound $\max_{k \in [s_i]} \|\boldsymbol{u}^{(k)}\|_2$. For any $k$, from the definition of $\boldsymbol{u}^{(k)}$ and the inductive hypothesis, we have that

$$\|\boldsymbol{u}^{(k)}\|_2^2 = \sum_{n=0}^{\ell_i} \left(\beta_n^{(i)}\right)^2 \cdot \sum_{(m_1,\ldots,m_n) \in [N_{\boldsymbol{\ell}}^{(i)}]^n} \prod_{j=1}^n \left(\boldsymbol{u}_{m_j}^{(k)}\right)^2$$

$$= \sum_{n=0}^{\ell_i} \left(\beta_n^{(i)}\right)^2 \|\boldsymbol{v}^{(k)}\|_2^{2n} \leq 2^{O(\ell_i)} \cdot \|\boldsymbol{v}^{(k)}\|_2^{2\ell_i}$$

We analyse the case $i = 1$ and $i > 1$ separately. When $i = 1$, we have that $2^{O(\ell_1)}\|\boldsymbol{v}^{(k)}\|_2^{2\ell_1} \leq (2\|W^{(1)}\|_2^\infty)^{O(\ell_1)}$ from the bound on the base case. When $i > 1$, we have

$$\|\sum_{k=1}^{s_i} W_{jk}^{(i+1)} \boldsymbol{u}^{(k)}\|_2^2 \leq W^2 2^{O(\ell_i)} \|\boldsymbol{v}^{(k)}\|_2^{2\ell_i}$$

$$\leq W^2 2^{O(\ell_i)} \left((2\|W^{(1)}\|_2^\infty)^{O(\prod_{n=1}^{i-1}\ell_n)} \cdot W^{O(\prod_{n=2}^{i-1}\ell_n)}\right)^{2\ell_i}$$

$$\leq (2\|W^{(1)}\|_2^\infty)^{O(\prod_{n=1}^i \ell_n)} \cdot W^{O(\prod_{n=2}^i \ell_n)}$$

which completes the induction. We are ready to calculate the bound on the degree.

We have $\ell_1 = O(Rt\|W^{(1)}\|_2^\infty \log(RW/\epsilon))$. Also, for $i > 1$, we have $\ell_i = O(Wt\log(W/\epsilon))$. Thus, the total degree is $\ell \leq \prod_{i=1}^{t-1} \ell_i = O\left((R\log R) \cdot (\|W^{(1)}\|_2^\infty W^{t-2}) \cdot (t\log(W/\epsilon))^{t-1}\right)$. The square of the norm of the kernel representation is bounded by $B$ where

$$B \leq (2\|W^{(1)}\|_2^\infty)^\ell \cdot W^{O\left(W^{t-2}(t\log(W/\epsilon)^{t-2})\right)}.$$

This concludes the proof. $\qquad\qquad\qquad\qquad\qquad\qquad\qquad\qquad\qquad\qquad\qquad\qquad\quad\square$

# B   TDS LEARNING AND KERNEL METHODS

## B.1   GENERAL THEOREM

We provide here the full proof of Theorem 3.6. First, we restate and prove the multiplicative spectral concentration lemma (Lemma 3.8).

**Lemma B.1** (Multiplicative Spectral Concentration, Lemma B.1 in Goel et al. (2024), modified). *Let $\mathcal{D}_{\boldsymbol{x}}$ be a distribution over $\mathbb{R}^d$ and $\phi : \mathbb{R}^d \to \mathbb{R}^m$ such that $\mathcal{D}_{\boldsymbol{x}}$ is $(\phi, C, \ell)$-hypercontractive for some $C, \ell \geq 1$. Suppose that $S$ consists of $N$ i.i.d. examples from $\mathcal{D}_{\boldsymbol{x}}$ and let $\Phi = \mathbb{E}_{\boldsymbol{x} \sim \mathcal{D}_{\boldsymbol{x}}}[\phi(\boldsymbol{x})\phi(\boldsymbol{x})^\top]$, and $\hat{\Phi} = \frac{1}{N}\sum_{\boldsymbol{x} \in S} \phi(\boldsymbol{x})\phi(\boldsymbol{x})^\top$. For any $\epsilon, \delta \in (0,1)$, if $N \geq \frac{64Cm^2}{\epsilon^2}(4C\log_2(\frac{4}{\delta}))^{4\ell+1}$, then with probability at least $1 - \delta$, we have that*

$$\text{For any } \boldsymbol{a} \in \mathbb{R}^m : \boldsymbol{a}^\top \hat{\Phi} \boldsymbol{a} \in [(1-\epsilon)\boldsymbol{a}^\top \Phi \boldsymbol{a}, (1+\epsilon)\boldsymbol{a}^\top \Phi \boldsymbol{a}]$$

*Proof of Lemma 3.8.* Let $\Phi = UDU^\top$ be the compact SVD of $\Phi$ (i.e., $D$ is square with dimension equal to the rank of $\Phi$ and $U$ is not necessarily square). Note that such a decomposition exists (where the row and column spaces are both spanned by the same basis $U$), because $\Phi = \Phi^\top$, by definition. Moreover, note that $UU^T$ is an orthogonal projection matrix that projects points in $\mathbb{R}^m$ on the span of the rows of $\Phi$. We also have that, $U^\top U = I$.

Consider $\Phi^\dagger = UD^{-1}U^\top$ and $\Phi^{\frac{1}{2}} = UD^{-\frac{1}{2}}U^\top$. Our proof consists of two parts. We first show that it is sufficient to prove that $\|\Phi^{\frac{1}{2}}\Phi\Phi^{\frac{1}{2}} - \Phi^{\frac{1}{2}}\hat{\Phi}\Phi^{\frac{1}{2}}\|_2 \leq \epsilon$ with probability at least $1 - \delta$ and then we give a bound on the probability of this event.

**Claim.** *Suppose that for $\boldsymbol{A} = \Phi^{\frac{1}{2}}\Phi\Phi^{\frac{1}{2}} - \Phi^{\frac{1}{2}}\hat{\Phi}\Phi^{\frac{1}{2}}$ we have $\|\boldsymbol{A}\|_2 \leq \epsilon$. Then, for any $\boldsymbol{a} \in \mathbb{R}^m$:*

$$\boldsymbol{a}^\top \hat{\Phi} \boldsymbol{a} \in [(1-\epsilon)\boldsymbol{a}^\top \Phi \boldsymbol{a}, (1+\epsilon)\boldsymbol{a}^\top \Phi \boldsymbol{a}]$$

*Proof.* Let $\boldsymbol{a} \in \mathbb{R}^m$, $\boldsymbol{a}_+ = UU^\top \boldsymbol{a}$, and $\boldsymbol{a}_0 = (I - UU^\top)\boldsymbol{a}$ (i.e., $\boldsymbol{a} = \boldsymbol{a}_0 + \boldsymbol{a}_+$, where $\boldsymbol{a}_0$ is the component of $\boldsymbol{a}$ lying in the nullspace of $\Phi$). We have that $\boldsymbol{a}^\top \Phi \boldsymbol{a} = \boldsymbol{a}_+^\top \Phi \boldsymbol{a}_+$.

Moreover, for $\boldsymbol{a}_0$, we have that $0 = \boldsymbol{a}_0^\top \Phi \boldsymbol{a}_0 = \mathbb{E}_{\boldsymbol{x} \sim \mathcal{D}_{\boldsymbol{x}}}[(\phi(\boldsymbol{x})^\top \boldsymbol{a}_0)^2]$ and, hence, $\phi(\boldsymbol{x})^\top \boldsymbol{a}_0 = 0$ almost surely over $\mathcal{D}_{\boldsymbol{x}}$. Therefore, we also have $\boldsymbol{a}_0^\top \hat{\Phi} \boldsymbol{a}_0 = \frac{1}{N} \sum_{\boldsymbol{x} \in S}(\phi(\boldsymbol{x})^\top \boldsymbol{a}_0)^2 = 0$, with probability 1. Therefore, $\boldsymbol{a}^\top \hat{\Phi} \boldsymbol{a} = \boldsymbol{a}_+^\top \hat{\Phi} \boldsymbol{a}_+$.

Observe, now, that $\Phi^{\frac{1}{2}} \Phi^{\frac{\dagger}{2}} = U D^{\frac{1}{2}} U^\top U D^{-\frac{1}{2}} U^\top = UU^\top$ and, hence, $\Phi^{\frac{1}{2}} \Phi^{\frac{\dagger}{2}} \boldsymbol{a}_+ = (UU^\top)^2 \boldsymbol{a} = UU^\top \boldsymbol{a} = \boldsymbol{a}_+$, because $UU^\top$ is a projection matrix. Overall, we obtain the following

$$\begin{aligned}
\boldsymbol{a}^\top \hat{\Phi} \boldsymbol{a} &= \boldsymbol{a}^\top \Phi \boldsymbol{a} + \boldsymbol{a}_+^\top (\hat{\Phi} - \Phi) \boldsymbol{a}_+ \\
&= \boldsymbol{a}^\top \Phi \boldsymbol{a} + \boldsymbol{a}_+^\top \Phi^{\frac{1}{2}} (\Phi^{\frac{\dagger}{2}} \hat{\Phi} \Phi^{\frac{\dagger}{2}} - \Phi^{\frac{\dagger}{2}} \Phi \Phi^{\frac{\dagger}{2}}) \Phi^{\frac{1}{2}} \boldsymbol{a}_+ \\
&= \boldsymbol{a}^\top \Phi \boldsymbol{a} + \boldsymbol{a}_+^\top \Phi^{\frac{1}{2}} A \Phi^{\frac{1}{2}} \boldsymbol{a}_+
\end{aligned}$$

Since $\|A\|_2 \leq \epsilon$ and $\Phi^{\frac{1}{2}} \Phi^{\frac{1}{2}} = \Phi$, we have that $|\boldsymbol{a}_+^\top \Phi^{\frac{1}{2}} A \Phi^{\frac{1}{2}} \boldsymbol{a}_+| \leq \epsilon |\boldsymbol{a}_+^\top \Phi \boldsymbol{a}_+| = \epsilon |\boldsymbol{a}^\top \Phi \boldsymbol{a}|$, which concludes the proof of the claim. $\qquad\square$

It remains to show that for the matrix $A$ defined in the previous claim, we have $\|A\|_2 \leq \epsilon$ with probability at least $1 - \delta$. The randomness of $A$ depends on the random choice of $S$ from $\mathcal{D}_{\boldsymbol{x}}^{\otimes m}$. In the rest of the proof, therefore, consider all probabilities and expectations to be over $S \sim \mathcal{D}_{\boldsymbol{x}}^{\otimes m}$. We have the following for $t = \log_2(4/\delta)$.

$$\mathbf{Pr}[\|A\|_2 > \epsilon] \leq \mathbf{Pr}[\|A\|_F > \epsilon] \leq \frac{\mathbb{E}[\|A\|_F^{2t}]}{\epsilon^{2t}}$$

We will now bound the expectation of $\mathbb{E}[\|A\|_F^{2t}]$. To this end, we define $\boldsymbol{a}_i = \Phi^{\frac{\dagger}{2}} \boldsymbol{e}_i \in \mathbb{R}^m$ for $i \in [m]$. We have the following, by using Jensen's inequality appropriately.

$$\begin{aligned}
\mathbb{E}[\|A\|_F^{2t}] &= \mathbb{E}\Big[\Big(\sum_{i,j \in [m]} (\boldsymbol{a}_i^\top \Phi \boldsymbol{a}_j - \boldsymbol{a}_i^\top \hat{\Phi} \boldsymbol{a}_j)^2\Big)^t\Big] \\
&\leq m^{2(t-1)} \sum_{i,j \in [m]} \mathbb{E}[(\boldsymbol{a}_i^\top \Phi \boldsymbol{a}_j - \boldsymbol{a}_i^\top \hat{\Phi} \boldsymbol{a}_j)^{2t}] \\
&\leq m^{2t} \max_{i,j \in [m]} \mathbb{E}[(\boldsymbol{a}_i^\top \Phi \boldsymbol{a}_j - \boldsymbol{a}_i^\top \hat{\Phi} \boldsymbol{a}_j)^{2t}]
\end{aligned}$$

In order to bound the term above, we may use Marcinkiewicz-Zygmund inequality (see Ferger (2014)) to exploit the independence of the samples in $S$ and obtain the following.

$$\begin{aligned}
\mathbb{E}[(\boldsymbol{a}_i^\top \Phi \boldsymbol{a}_j - \boldsymbol{a}_i^\top \hat{\Phi} \boldsymbol{a}_j)^{2t}] &\leq \frac{2(4t)^t}{N^t} \mathbb{E}_{\boldsymbol{x} \sim \mathcal{D}_{\boldsymbol{x}}}[(\boldsymbol{a}_i^\top \Phi \boldsymbol{a}_j - \boldsymbol{a}_i^\top \phi(\boldsymbol{x})\phi(\boldsymbol{x})^\top \boldsymbol{a}_j)^{2t}] \\
&\leq \frac{2(4t)^t}{N^t} \big(2^{2t}(\boldsymbol{a}_i^\top \Phi \boldsymbol{a}_j)^{2t} + 2^{2t} \mathbb{E}_{\boldsymbol{x} \sim \mathcal{D}_{\boldsymbol{x}}}[(\boldsymbol{a}_i^\top \phi(\boldsymbol{x})\phi(\boldsymbol{x})^\top \boldsymbol{a}_j)^{2t}]\big)
\end{aligned}$$

We now observe that $\mathbb{E}_{\boldsymbol{x} \sim \mathcal{D}_{\boldsymbol{x}}}[\boldsymbol{a}_i^\top \phi(\boldsymbol{x})\phi(\boldsymbol{x})^\top \boldsymbol{a}_j] = \boldsymbol{a}_i^\top \Phi \boldsymbol{a}_j = \boldsymbol{e}_i^\top \Phi^{\frac{\dagger}{2}} \Phi \Phi^{\frac{\dagger}{2}} \boldsymbol{e}_j = \boldsymbol{e}_i^\top UU^T \boldsymbol{e}_j$, which is at most equal to 1. Therefore, we have $\mathbb{E}_{\boldsymbol{x} \sim \mathcal{D}_{\boldsymbol{x}}}[(\boldsymbol{a}_i^\top \phi(\boldsymbol{x}))^2] \leq 1$ and, by the hypercontractivity property (which we assume to be with respect to the standard inner product in $\mathbb{R}^m$), we have $\mathbb{E}_{\boldsymbol{x} \sim \mathcal{D}_{\boldsymbol{x}}}[(\boldsymbol{a}_i^\top \phi(\boldsymbol{x}))^{4t}] \leq (4Ct)^{4\ell t}$. We can bound $\mathbb{E}_{\boldsymbol{x} \sim \mathcal{D}_{\boldsymbol{x}}}[(\boldsymbol{a}_i^\top \phi(\boldsymbol{x})\phi(\boldsymbol{x})^\top \boldsymbol{a}_j)^{2t}]$ by applying the Cauchy-Schwarz inequality and using the bound for $\mathbb{E}_{\boldsymbol{x} \sim \mathcal{D}_{\boldsymbol{x}}}[(\boldsymbol{a}_i^\top \phi(\boldsymbol{x}))^{4t}]$. In total, we have the following bound.

$$\mathbf{Pr}[\|A\|_2 > \epsilon] \leq 4\Big(\frac{16m^2 t (4Ct)^{4\ell}}{N\epsilon^2}\Big)^t$$

We choose $N$ such that $\frac{16m^2 t (4Ct)^{4\ell}}{N\epsilon^2} \leq \frac{1}{2}$ and $t = \log_2(4/\delta)$ so that the bound is at most $\delta$. $\qquad\square$

We are now ready to prove the main theorem, which we restate here for convenience.

**Theorem B.2** (TDS Learning via the Kernel Method). *Suppose that $\mathcal{F} \subseteq \{\mathbb{R}^d \to \mathbb{R}\}$, the training and test distributions $\mathcal{D}$, $\mathcal{D}'$ over $\mathbb{R}^d \times \mathbb{R}$, are such that the following are true for $A, B, C, M, \ell \geq 1$ and $\epsilon \in (0, 1)$.*

1. $\mathcal{F}$ is $(\epsilon, B)$-approximately represented within radius $R$ w.r.t. a PDS kernel $\mathcal{K} : \mathbb{R}^d \times \mathbb{R}^d \to \mathbb{R}$, for some $\epsilon \in (0,1)$ and $B, R \geq 1$ and let $A = \sup_{\boldsymbol{x} : \|\boldsymbol{x}\|_2 \leq R} \mathcal{K}(\boldsymbol{x}, \boldsymbol{x})$.

2. The training marginal $\mathcal{D}_{\boldsymbol{x}}$ (1) is bounded within $\{\boldsymbol{x} : \|\boldsymbol{x}\|_2 \leq R\}$ and (2) is $(\mathcal{K}, C, \ell)$-hypercontractive for some $C, \ell \geq 1$.

3. The training and test labels are both bounded in $[-M, M]$ for some $M \geq 1$.

*Then, Algorithm 1 learns the class $\mathcal{F}$ in the TDS regression setting up to excess error $5\epsilon$ and probability of failure $\delta$. The time complexity is $O(T) \cdot \mathrm{poly}(d, \frac{1}{\epsilon}, (\log(1/\delta))^{\ell}, A, B, C^{\ell}, 2^{\ell}, M)$, where $T$ is the evaluation time of $\mathcal{K}$.*

*Proof of Theorem 3.6.* Consider the reference feature map $\phi : \mathbb{R}^d \to \mathbb{R}^{2m}$ with $\phi(\boldsymbol{x}) = (\mathcal{K}(\boldsymbol{x}, \boldsymbol{z}))_{\boldsymbol{z} \in S_{\mathrm{ref}} \cup S'_{\mathrm{ref}}}$. Let $f^* = \arg\min_{f \in \mathcal{F}} [\mathcal{L}_{\mathcal{D}}(f) + \mathcal{L}_{\mathcal{D}'}(f)]$ and $f_{\mathrm{opt}} = \arg\min_{f \in \mathcal{F}} [\mathcal{L}_{\mathcal{D}}(f)]$. By Assumption 3.5, we know that there are functions $p^*, p_{\mathrm{opt}} : \mathbb{R}^d \to \mathbb{R}$ with $p^*(\boldsymbol{x}) = \langle \boldsymbol{v}^*, \psi(\boldsymbol{x}) \rangle$ and $p_{\mathrm{opt}} = \langle \boldsymbol{v}_{\mathrm{opt}}, \psi(\boldsymbol{x}) \rangle$, that uniformly approximate $f^*$ and $f_{\mathrm{opt}}$ within the ball of radius $R$, i.e., $\sup_{\boldsymbol{x} : \|\boldsymbol{x}\|_2 \leq R} |f^*(\boldsymbol{x}) - p^*(\boldsymbol{x})| \leq \epsilon$ and $\sup_{\boldsymbol{x} : \|\boldsymbol{x}\|_2 \leq R} |f_{\mathrm{opt}}(\boldsymbol{x}) - p_{\mathrm{opt}}(\boldsymbol{x})| \leq \epsilon$. Moreover, $\langle \boldsymbol{v}^*, \boldsymbol{v}^* \rangle, \langle \boldsymbol{v}_{\mathrm{opt}}, \boldsymbol{v}_{\mathrm{opt}} \rangle \leq B$.

By Proposition 3.7, there is $\boldsymbol{a}^* \in \mathbb{R}^{2m}$ such that for $\tilde{p}^* : \mathbb{R}^d \to \mathbb{R}$ with $\tilde{p}^*(\boldsymbol{x}) = (\boldsymbol{a}^*)^{\top} \phi(\boldsymbol{x})$ we have $\|f^* - \tilde{p}^*\|_{S_{\mathrm{ref}}} \leq 3\epsilon/2$ and $\|f^* - \tilde{p}^*\|_{S'_{\mathrm{ref}}} \leq 3\epsilon/2$. Let $\boldsymbol{K}$ be a matrix in $\mathbb{R}^{2m \times 2m}$ such that $\boldsymbol{K}_{\boldsymbol{z}, \boldsymbol{w}} = \mathcal{K}(\boldsymbol{z}, \boldsymbol{w})$ for $\boldsymbol{z}, \boldsymbol{w} \in S_{\mathrm{ref}} \cup S'_{\mathrm{ref}}$. We additionally have that $(\boldsymbol{a}^*)^{\top} \boldsymbol{K} \boldsymbol{a}^* \leq B$. Therefore, for any $\boldsymbol{x} \in \mathbb{R}^d$ we have

$$\begin{aligned}
(\tilde{p}^*(\boldsymbol{x}))^2 &= \Big( \Big\langle \sum_{\boldsymbol{z} \in S_{\mathrm{ref}} \cup S'_{\mathrm{ref}}} a_{\boldsymbol{z}}^* \psi(\boldsymbol{z}), \psi(\boldsymbol{x}) \Big\rangle \Big)^2 \\
&\leq \Big\langle \sum_{\boldsymbol{z} \in S_{\mathrm{ref}} \cup S'_{\mathrm{ref}}} a_{\boldsymbol{z}}^* \psi(\boldsymbol{z}), \sum_{\boldsymbol{z} \in S_{\mathrm{ref}} \cup S'_{\mathrm{ref}}} a_{\boldsymbol{z}}^* \psi(\boldsymbol{z}) \Big\rangle \cdot \langle \psi(\boldsymbol{x}), \psi(\boldsymbol{x}) \rangle \\
&= (\boldsymbol{a}^*)^{\top} \boldsymbol{K} \boldsymbol{a}^* \cdot \mathcal{K}(\boldsymbol{x}, \boldsymbol{x}) \leq B \cdot \mathcal{K}(\boldsymbol{x}, \boldsymbol{x}),
\end{aligned}$$

where we used the Cauchy-Schwarz inequality. For $\boldsymbol{x}$ with $\|\boldsymbol{x}\|_2 \leq R$, we, hence, have $(\tilde{p}^*(\boldsymbol{x}))^2 \leq AB$ (recall that $A = \max_{\|\boldsymbol{x}\|_2 \leq R} \mathcal{K}(\boldsymbol{x}, \boldsymbol{x})$).

Similarly, by applying the representer theorem (Theorem 6.11 in Mohri et al. (2018)) for $p_{\mathrm{opt}}$, we have that there exists $\boldsymbol{a}^{\mathrm{opt}} = (a_{\boldsymbol{z}}^{\mathrm{opt}})_{\boldsymbol{z} \in S_{\mathrm{ref}}} \in \mathbb{R}^m$ such that for $\tilde{p}_{\mathrm{opt}} : \mathbb{R}^d \to \mathbb{R}$ with $\tilde{p}_{\mathrm{opt}}(\boldsymbol{x}) = \sum_{\boldsymbol{z} \in S_{\mathrm{ref}}} a_{\boldsymbol{z}}^{\mathrm{opt}} \mathcal{K}(\boldsymbol{z}, \boldsymbol{x})$ we have $\mathcal{L}_{\bar{S}_{\mathrm{ref}}}(\tilde{p}_{\mathrm{opt}}) \leq \mathcal{L}_{\bar{S}_{\mathrm{ref}}}(p_{\mathrm{opt}})$ and $\sum_{\boldsymbol{z}, \boldsymbol{w} \in S_{\mathrm{ref}}} a_{\boldsymbol{z}}^{\mathrm{opt}} a_{\boldsymbol{w}}^{\mathrm{opt}} \mathcal{K}(\boldsymbol{z}, \boldsymbol{w}) \leq B$. Since $\hat{p}$ in Algorithm 1 is formed by solving a convex program whose search space includes $\tilde{p}_{\mathrm{opt}}$, we have

$$\mathcal{L}_{\bar{S}_{\mathrm{ref}}}(\hat{p}) \leq \mathcal{L}_{\bar{S}_{\mathrm{ref}}}(\tilde{p}_{\mathrm{opt}}) \leq \mathcal{L}_{\bar{S}_{\mathrm{ref}}}(p_{\mathrm{opt}}) \tag{1}$$

In the following, we abuse the notation and consider $\hat{\boldsymbol{a}}$ to be a vector in $\mathbb{R}^{2m}$, by appending $m$ zeroes, one for each of the elements of $S'_{\mathrm{ref}}$. Note that we then have $\hat{\boldsymbol{a}}^{\top} \boldsymbol{K} \hat{\boldsymbol{a}} \leq B$, and, also, $(\hat{p}(\boldsymbol{x}))^2 \leq A \cdot B$ for all $\boldsymbol{x}$ with $\|\boldsymbol{x}\|_2 \leq R$.

**Soundness.** Suppose first that the algorithm has accepted. In what follows, we will use the triangle inequality of the norms to bound for functions $h_1, h_2, h_3$ the quantity $\|h_1 - h_2\|_{\mathcal{D}}$ by $\|h_1 - h_3\|_{\mathcal{D}} + \|h_2 - h_3\|_{\mathcal{D}}$. We also use the inequality $\mathcal{L}_{\mathcal{D}}(h_1) \leq \mathcal{L}_{\mathcal{D}}(h_2) + \|h_1 - h_2\|_{\mathcal{D}}$, as well as the fact that $\|\mathrm{cl}_M \circ h_1 - \mathrm{cl}_M \circ h_2\|_{\mathcal{D}} \leq \|\mathrm{cl}_M \circ h_1 - h_2\|_{\mathcal{D}} \leq \|h_1 - h_2\|_{\mathcal{D}}$. We bound the test error of the output hypothesis $h : \mathbb{R}^d \to [-M, M]$ of Algorithm 1 as follows.

$$\mathcal{L}_{\mathcal{D}'}(h) \leq \|h - \mathrm{cl}_M \circ f^*\|_{\mathcal{D}'_{\boldsymbol{x}}} + \mathcal{L}'_{\mathcal{D}}(f^*)$$

Since $(h(\boldsymbol{x}) - \mathrm{cl}_M(f^*(\boldsymbol{x})))^2 \leq 4M^2$ for all $\boldsymbol{x}$ and the hypothesis $h$ does not depend on the set $S'_{\mathrm{ref}}$, by a Hoeffding bound and the fact that $m$ is large enough, we obtain that $\|h - \mathrm{cl}_M \circ f^*\|_{\mathcal{D}'_{\boldsymbol{x}}} \leq \|h - \mathrm{cl}_M \circ f^*\|_{S'_{\mathrm{ref}}} + \epsilon/10$, with probability at least $1 - \delta/10$. Moreover, we have $\|h - \mathrm{cl}_M \circ f^*\|_{S'_{\mathrm{ref}}} \leq \|h - \mathrm{cl}_M \circ \tilde{p}^*\|_{S'_{\mathrm{ref}}} + \|\tilde{p}^* - f^*\|_{S'_{\mathrm{ref}}}$. We have already argued that $\|\tilde{p}^* - f^*\|_{S'_{\mathrm{ref}}} \leq 3\epsilon/2$.

In order to bound the quantity $\|h - \mathrm{cl}_M \circ \tilde{p}^*\|_{S'_{\mathrm{ref}}}$, we observe that while the function $h$ does not depend on $S'_{\mathrm{ref}}$, the function $\tilde{p}^*$ does depend on $S'_{\mathrm{ref}}$ and, therefore, standard concentration

arguments fail to bound the $\|h - \mathrm{cl}_M \circ \tilde{p}^*\|_{S'_{\mathrm{ref}}}$ in terms of $\|h - \mathrm{cl}_M \circ \tilde{p}^*\|_{\mathcal{D}'_{\boldsymbol{x}}}$. However, since we have clipped $\tilde{p}^*$, and $\tilde{p}^*$ is of the form $\langle \boldsymbol{v}^*, \psi \rangle$, we may obtain a bound using standard results from generalization theory (i.e., bounds on the Rademacher complexity of kernel-based hypotheses like Theorem 6.12 in Mohri et al. (2018) and uniform convergence bounds for classes with bounded Rademacher complexity under Lipschitz and bounded losses like Theorem 11.3 in Mohri et al. (2018)). In particular, we have that with probability at least $1 - \delta/10$

$$\|h - \mathrm{cl}_M \circ \tilde{p}^*\|_{S'_{\mathrm{ref}}} \leq \|h - \mathrm{cl}_M \circ \tilde{p}^*\|_{\mathcal{D}'_{\boldsymbol{x}}} + \epsilon/10$$

The corresponding requirement for $m = |S'_{\mathrm{ref}}|$ is determined by the bounds on the Lipschitz constant of the loss function $(y, t) \mapsto (y - \mathrm{cl}_M(t))^2$, with $y \in [-M, M]$ and $t \in \mathbb{R}$, which is at most $5M$, the overall bound on this loss function, which is at most $4M^2$, as well as the bounds $A = \max_{\boldsymbol{x}:\|\boldsymbol{x}\|_2 \leq R} \mathcal{K}(\boldsymbol{x}, \boldsymbol{x})$ and $(\boldsymbol{a}^*)^\top \boldsymbol{K} \boldsymbol{a} \leq B$ (which give bounds on the Rademacher complexity).

By applying the Hoeffding bound, we are able to further bound the quantity $\|h - \mathrm{cl}_M \circ \tilde{p}^*\|_{\mathcal{D}'_{\boldsymbol{x}}}$ by $\|h - \mathrm{cl}_M \circ \tilde{p}^*\|_{S'_{\mathrm{ver}}} + \epsilon/10$, with probability at least $1 - \delta$. We have effectively managed to bound the quantity $\|h - \mathrm{cl}_M \circ \tilde{p}^*\|_{S'_{\mathrm{ref}}}$ by $\|h - \mathrm{cl}_M \circ \tilde{p}^*\|_{S'_{\mathrm{ver}}} + \epsilon/5$. This is important, because the set $S'_{\mathrm{ver}}$ is a fresh set of examples and, therefore, independent from $\tilde{p}$. Our goal is now to use the fact that our spectral tester has accepted. We have the following for the matrix $\hat{\Phi}' = (\hat{\Phi}'_{\boldsymbol{z},\boldsymbol{w}})_{\boldsymbol{z},\boldsymbol{w} \in S_{\mathrm{ref}} \cup S'_{\mathrm{ref}}}$ with $\hat{\Phi}'_{\boldsymbol{z},\boldsymbol{w}} = \frac{1}{N} \sum_{\boldsymbol{x} \in S'_{\mathrm{ver}}} \mathcal{K}(\boldsymbol{x}, \boldsymbol{z}) \mathcal{K}(\boldsymbol{x}, \boldsymbol{w})$.

$$\begin{aligned}
\|h - \mathrm{cl}_M \circ \tilde{p}^*\|^2_{S'_{\mathrm{ver}}} &\leq \|\hat{p} - \tilde{p}^*\|^2_{S'_{\mathrm{ver}}} \\
&= (\hat{\boldsymbol{a}} - \boldsymbol{a}^*)^\top \hat{\Phi}' (\hat{\boldsymbol{a}} - \boldsymbol{a}^*)
\end{aligned}$$

Since our test has accepted, we know that $(\hat{\boldsymbol{a}} - \boldsymbol{a}^*)^\top \hat{\Phi}' (\hat{\boldsymbol{a}} - \boldsymbol{a}^*) \leq (1 + \rho)(\hat{\boldsymbol{a}} - \boldsymbol{a}^*)^\top \hat{\Phi} (\hat{\boldsymbol{a}} - \boldsymbol{a}^*)$, for the matrix $\hat{\Phi} = (\hat{\Phi}_{\boldsymbol{z},\boldsymbol{w}})_{\boldsymbol{z},\boldsymbol{w} \in S_{\mathrm{ref}} \cup S_{\mathrm{ref}}}$ with $\hat{\Phi}_{\boldsymbol{z},\boldsymbol{w}} = \frac{1}{N} \sum_{\boldsymbol{x} \in S_{\mathrm{ver}}} \mathcal{K}(\boldsymbol{x}, \boldsymbol{z}) \mathcal{K}(\boldsymbol{x}, \boldsymbol{w})$. We note here that having a multiplicative bound of this form is important, because we do not have any upper bound on the norms of $\hat{\boldsymbol{a}}$ and $\boldsymbol{a}^*$. Instead, we only have bounds on distorted versions of these vectors, e.g., on $\hat{\boldsymbol{a}}^\top \boldsymbol{K} \hat{\boldsymbol{a}}$, which does not imply any bound on the norm of $\hat{\boldsymbol{a}}$, because $\boldsymbol{K}$ could have very small singular values.

Overall, we have that $\|\hat{p} - \tilde{p}^*\|_{S'_{\mathrm{ver}}} - \|\hat{p} - \tilde{p}^*\|_{S_{\mathrm{ver}}} \leq \sqrt{\rho(2\|\hat{p}\|^2_{S_{\mathrm{ver}}} + 2\|\tilde{p}^*\|^2_{S_{\mathrm{ver}}})} \leq \sqrt{4AB\rho} \leq \frac{3\epsilon}{10}$.

By using results from generalization theory once more, we obtain that with probability at least $1 - \delta/5$ we have $\|\hat{p} - \tilde{p}^*\|_{S_{\mathrm{ver}}} \leq \|\hat{p} - \tilde{p}^*\|_{S_{\mathrm{ref}}} + \epsilon/5$. This step is important, because the only fact we know about the quality of $\hat{p}$ is that it outperforms every polynomial on the sample $S_{\mathrm{ref}}$ (not necessarily over the entire training distribution). We once more may use bounds on the values of $\hat{p}$ and $\tilde{p}^*$, this time without requiring clipping, since we know that the training marginal is bounded and, hence, the values of $\hat{p}$ and $\tilde{p}^*$ are bounded as well. This was not true for the test distribution, since we did not make any assumptions about it.

In order to bound $\|\hat{p} - \tilde{p}^*\|_{S_{\mathrm{ref}}}$, we have the following.

$$\begin{aligned}
\|\hat{p} - \tilde{p}^*\|_{S_{\mathrm{ref}}} &\leq \mathcal{L}_{\bar{S}_{\mathrm{ref}}}(\hat{p}) + \mathcal{L}_{\bar{S}_{\mathrm{ref}}}(\mathrm{cl} \circ f^*) + \|f^* - \tilde{p}^*\|_{S_{\mathrm{ref}}} \\
&\leq \mathcal{L}_{\bar{S}_{\mathrm{ref}}}(\tilde{p}_{\mathrm{opt}}) + \mathcal{L}_{\bar{S}_{\mathrm{ref}}}(\mathrm{cl} \circ f^*) + \|f^* - \tilde{p}^*\|_{S_{\mathrm{ref}}} && \text{(By equation 1)} \\
&\leq \mathcal{L}_{\bar{S}_{\mathrm{ref}}}(p_{\mathrm{opt}}) + \mathcal{L}_{\bar{S}_{\mathrm{ref}}}(\mathrm{cl} \circ f^*) + \|f^* - \tilde{p}^*\|_{S_{\mathrm{ref}}}
\end{aligned}$$

The first term above is bounded as $\mathcal{L}_{\bar{S}_{\mathrm{ref}}}(p_{\mathrm{opt}}) \leq \mathcal{L}_{\bar{S}_{\mathrm{ref}}}(\mathrm{cl}_M \circ f_{\mathrm{opt}}) + \|p_{\mathrm{opt}} - f_{\mathrm{opt}}\|_{S_{\mathrm{ref}}}$, where the second term is at most $\epsilon$ (by the definition of $p_{\mathrm{opt}}$) and the first term can be bounded by $\mathcal{L}_{\mathcal{D}}(f_{\mathrm{opt}}) + \epsilon/10 = \mathrm{opt} + \epsilon/10$, with probability at least $1 - \delta/10$, due to an application of the Hoeffding bound.

For the term $\mathcal{L}_{\bar{S}_{\mathrm{ref}}}(\mathrm{cl} \circ f^*)$ we can similarly use the Hoeffding bound to obtain, with probability at least $1 - \delta/10$ that $\mathcal{L}_{\bar{S}_{\mathrm{ref}}}(\mathrm{cl} \circ f^*) \leq \mathcal{L}_{\mathcal{D}}(f^*) + \epsilon/10$.

Finally, for the term $\|f^* - \tilde{p}^*\|_{S_{\mathrm{ref}}}$, we have that $\|f^* - \tilde{p}^*\|_{S_{\mathrm{ref}}} \leq 3\epsilon/2$, as argued above.

Overall, we obtain a bound of the form $\mathcal{L}'_{\mathcal{D}}(h) \leq \mathcal{L}_{\mathcal{D}}(f^*) = \mathcal{L}_{\mathcal{D}'}(f^*) + \mathcal{L}_{\mathcal{D}}(f_{\mathrm{opt}}) + 5\epsilon$, with probability at least $1 - \delta$, as desired.

**Completeness.** For the completeness criterion, we assume that the test marginal is equal to the training marginal. Then, by Lemma 3.8 (where we observe that any $(\psi, C, \ell)$-hypercontractive

distribution is also $(\phi, C, \ell)$-hypercontractive), with probability at least $1 - \delta$, we have that for all $\boldsymbol{a} \in \mathbb{R}^{2m}$, $\boldsymbol{a}^\top \hat{\Phi}' \boldsymbol{a} \le \frac{1+(\rho/4)}{1-(\rho/4)} \boldsymbol{a}^\top \hat{\Phi} \boldsymbol{a} \le (1 + \rho)\boldsymbol{a}^\top \hat{\Phi} \boldsymbol{a}$, because $\mathbb{E}[\hat{\Phi}] = \mathbb{E}[\hat{\Phi}']$ and the matrices are sums of independent samples of $\phi(\boldsymbol{x})\phi(\boldsymbol{x})^\top$, where $\boldsymbol{x} \sim \mathcal{D}_{\boldsymbol{x}}$. It is crucial here that $\phi$ (which recall is formed by using $S_{\text{ref}}, S'_{\text{ref}}$) does not depend on the verification samples $S_{\text{ver}}$ and $S'_{\text{ver}}$, which is why we chose them to be fresh. Therefore, the test will accept with probability at least $1 - \delta$.

**Efficient Implementation.** To compute $\hat{\boldsymbol{a}}$, we may run a least squares program, in time polynomial in $m$. For the spectral tester, we first compute the SVD of $\hat{\Phi}$ and check that any vector in the kernel of $\hat{\Phi}$ is also in the kernel of $\hat{\Phi}'$ (this can be checked without computing the SVD of $\hat{\Phi}'$). Otherwise, reject. Then, let $\hat{\Phi}^{\frac{\dagger}{2}}$ be the root of the Moore-Penrose pseudoinverse of $\hat{\Phi}$ and find the maximum singular value of the matrix $\hat{\Phi}^{\frac{\dagger}{2}} \hat{\Phi}' \hat{\Phi}^{\frac{\dagger}{2}}$. If the value is higher than $1 + \rho$, reject. Note that this is equivalent to solving the eigenvalue problem described in Algorithm 1. $\square$

## B.2 APPLICATIONS

We first state and prove our end to end results on TDS learning Sigmoid and Lipschitz nets over bounded marginals that are $C$-hypercontractive for some constant $C$.

**Theorem B.3** (TDS Learning for Nets with Sigmoid Activation). *Let $\mathcal{F}$ on $\mathbb{R}^d$ be the class of neural network with sigmoid activations, depth $t$ and weight matrices $\mathbf{W} = (W^{(1)}, \ldots, W^{(t)})$ such that $\|\mathbf{W}\|_1 \le W$. Let $\epsilon \in (0, 1)$. Suppose the training and test distributions $\mathcal{D}, \mathcal{D}'$ over $\mathbb{R}^d \times \mathbb{R}$ are such that the following are true:*

1. *$\mathcal{D}_{\boldsymbol{x}}$ is bounded within $\{\boldsymbol{x} : \|\boldsymbol{x}\|_2 \le R\}$ and is $C$-hypercontractive for $R, C \ge 1$,*

2. *The training and test labels are bounded in $[-M, M]$ for some $M \ge 1$.*

*Then, Algorithm 1 learns the class $\mathcal{F}$ in the TDS regression up to excess error $\epsilon$ and probability of failure $\delta$. The time and sample complexity is $\text{poly}\left(d, \frac{1}{\epsilon}, C^\ell, M, \log(1/\delta)^\ell, (2R)^{2^t \cdot \ell}, (2\|W^{(1)}\|_2^\infty)^\ell \cdot W^{O\left((Wt\log(W/\epsilon))^{t-2}\right)}\right)$ where $\ell = O\left((R\log R) \cdot (\|W^{(1)}\|_2^\infty W^{t-2}) \cdot (t\log(W/\epsilon))^{t-1}\right)$.*

*Proof.* From Theorem A.21, we have that $\mathcal{F}$ is $\left(\epsilon, (2\|W^{(1)}\|_2^\infty)^\ell W^{O\left(W^{t-2}(t\log(W/\epsilon)^{t-2})\right)}\right)$-approximately represented within radius $R$ w.r.t $\mathsf{MK}_{\boldsymbol{\ell}}^{(t)}$ where $\ell$ is a degree vector whose product is equal to $\ell = O\left((R\log R) \cdot (\|W^{(1)}\|_2^\infty W^{t-2}) \cdot (t\log(W/\epsilon))^{t-1}\right)$. Also, from Lemma A.16, we have that $A := \sup_{\|\boldsymbol{x}\|_2 \le R} \mathsf{MK}_{\boldsymbol{\ell}}^{(t)}(\boldsymbol{x}, \boldsymbol{x}) \le (2R)^{2^t \ell}$. From Lemma A.16, the entries of the kernel can be computed in $\text{poly}(d, \ell)$ time and from Lemma A.17, we have that $\mathcal{D}_{\boldsymbol{x}}$ is $\left(\mathsf{MK}_{\boldsymbol{\ell}}^{(t)}, C, \ell\right)$ hypercontractive. Now, we obtain the result by applying Theorem B.2. $\square$

The following corollary on TDS learning two layer sigmoid networks in polynomial time readily follows.

**Corollary B.4.** *Let $\mathcal{F}$ on $\mathbb{R}^d$ be the class of two-layer neural networks with weight matrices $\mathbf{W} = (W^{(1)}, W^{(2)})$ and sigmoid activations. Let $\|W^{(1)}\|_2^\infty \le O(1)$ and $\|\mathbf{W}\|_1 \le W$. Suppose the training and test distributions satisfy the assumptions from Theorem B.3 with $R = O(1)$. Then, Algorithm 1 learns the class $\mathcal{F}$ in the TDS regression setting up to excess error $\epsilon$ and probability of failure $0.1$ in time and sample complexity $\text{poly}(d, 1/\epsilon, W, M)$.*

*Proof.* The proof immediately follows from Theorem B.3 by setting $t = 2$ and the other parameters to the appropriate constants. $\square$

**Theorem B.5** (TDS Learning for Nets with Lipschitz Activation). *Let $\mathcal{F}$ on $\mathbb{R}^d$ be the class of neural network with $L$-Lipschitz activations, depth $t$ and weight matrices $\mathbf{W} = (W^{(1)}, \ldots, W^{(t)})$ such that $\|\mathbf{W}\|_1 \le W$. Let $\epsilon \in (0, 1)$. Suppose the training and test distributions $\mathcal{D}, \mathcal{D}'$ over $\mathbb{R}^d \times \mathbb{R}$ are such that the following are true:*

1. $\mathcal{D}_{\boldsymbol{x}}$ is bounded within $\{\boldsymbol{x} : \|\boldsymbol{x}\|_2 \le R\}$ and is $C$-hypercontractive for $R, C \ge 1$,

2. The training and test labels are bounded in $[-M, M]$ for some $M \ge 1$.

Then, Algorithm 1 learns the class $\mathcal{F}$ in the TDS regression up to excess error $\epsilon$ and probability of failure $\delta$. The time and sample complexity is $\mathrm{poly}\left(d, \frac{1}{\epsilon}, C^{\ell}, M, \log(1/\delta)^{\ell}, (2R(k+\ell))^{O(\ell)}\right)$ where $\ell = O\left(\|W^{(1)}\|_2^{\infty}(WL)^{t-1}Rk\sqrt{k}/\epsilon\right)$. When $k = 1$, we have that the time and sample complexity is $\mathrm{poly}(d, \frac{1}{\epsilon}, C^{\ell}, M, \log(1/\delta)^{\ell}, (2R)^{O(\ell)})$ where $\ell = O\left(\|W^{(1)}\|_2^{\infty}(WL)^{t-1}R/\epsilon\right)$.

*Proof.* From Theorem A.19, for $k > 1$ we have that $\mathcal{F}$ is $(\epsilon, (k+\ell)^{O(\ell)})$-approximately represented within radius $R$ w.r.t $\mathsf{MK}_{\boldsymbol{\ell}}^{(1)}$ where $\boldsymbol{\ell}$ is a degree vector whose product is equal to $\ell = O\left(\|W^{(1)}\|_2^{\infty}(WL)^{t-1}Rk\sqrt{k}/\epsilon\right)$. For $k = 1$, we have that we have that $\mathcal{F}$ is $(\epsilon, 2^{O(\ell)})$-approximately represented within radius $R$ w.r.t $\mathsf{MK}_{\boldsymbol{\ell}}^{(1)}$ where $\boldsymbol{\ell}$ is a degree vector whose product is equal to $\ell = O\left(\|W^{(1)}\|_2^{\infty}(WL)^{t-1}R/\epsilon\right)$. Also, from Lemma A.16, we have that $A := \sup_{\|\boldsymbol{x}\|_2 \le R} \mathsf{MK}_{\boldsymbol{\ell}}^{(t)}(\boldsymbol{x}, \boldsymbol{x}) \le (2R)^{O(\ell)}$. From Lemma A.16, the entries of the kernel can be computed in $\mathrm{poly}(d, \ell)$ time and from Lemma A.17, we have that $\mathcal{D}_{\boldsymbol{x}}$ is $\left(\mathsf{MK}_{\boldsymbol{\ell}}^{(1)}, C, \ell\right)$ hypercontractive. Now, we obtain the result by applying Theorem B.2. $\qquad\square$

The above theorem implies the following corollary about TDS learning the class of ReLUs.

**Corollary B.6.** *Let $\mathcal{F} = \{\boldsymbol{x} \to \max(0, \boldsymbol{w} \cdot \boldsymbol{x}) : \|\boldsymbol{w}\|_2 = 1\}$ on $\mathbb{R}^d$ be the class of ReLU functions with unit weight vectors. Suppose the training and test distributions satisfy the assumptions from Theorem B.5 with $R = O(1)$. Then, Algorithm 1 learns the class $\mathcal{F}$ in the TDS regression setting up to excess error $\epsilon$ and probability of failure $0.1$ in time and sample complexity $\mathrm{poly}(d, 2^{O(1/\epsilon)}, M)$.*

*Proof.* The proof immediately follows from Theorem B.5 by setting $t = 2, \mathbf{W} = (\boldsymbol{w})$ and the activation to be the ReLU function. $\qquad\square$

In particular, this implies that the class of ReLUs is TDS learnable in polynomial time when $\epsilon < O(1/\log d)$.

## C  TDS LEARNING AND UNIFORM APPROXIMATION

### C.1  PRELIMINARIES

We first define the notion of a subspace junta which will be useful in this section. Intuitively, we want to consider the neural network as a function of $W\boldsymbol{x}$ after the first layer of weights has been applied, which allows us to project from the higher $d$-dimensional input space to a $k$-dimensional subspace.

**Definition C.1** (Subspace Junta). *A function $f : \mathbb{R}^d \to \mathbb{R}$ is a $k$-subspace junta (where $k \le d$) if there exists $W \in \mathbb{R}^{k \times d}$ with $\|W\|_2 = 1$ and $WW^{\top} = I_k$ and a function $g : \mathbb{R}^k \to \mathbb{R}$ such that*

$$f(\boldsymbol{x}) = f_W(\boldsymbol{x}) = g(Wx) \quad \forall x \in \mathbb{R}^d.$$

Note that by taking $k = d$, letting $W = I_d$ covers all functions $f : \mathbb{R}^d \to \mathbb{R}$.

We obtain the following corollary which gives the analogous bound on the $(\epsilon, R)$-uniform approximation to a $k$-subspace junta, given the $(\epsilon, R)$-uniform approximation to the corresponding function $g$.

**Corollary C.2.** *Let $\epsilon > 0, R \ge 1$, and $f : \mathbb{R}^d \to \mathbb{R}$ be a $k$-subspace junta, and consider the corresponding function $g(Wx)$. Let $q : \mathbb{R}^k \to \mathbb{R}$ be an $(\epsilon, R)$-uniform approximation polynomial for $g$, and define $p : \mathbb{R}^d \to \mathbb{R}$ as $p(\boldsymbol{x}) := q(Wx)$. Then $|p(\boldsymbol{x}) - f(\boldsymbol{x})| \le \epsilon$ for all $\|Wx\|_2 \le R$.*

In this section, we obtain TDS learning algorithms with respect to a training marginal which is a strictly sub-exponential distribution, which we now define.

**Definition C.3** (Strictly Sub-exponential Distribution). A distribution $\mathcal{D}$ on $\mathbb{R}^d$ is $\gamma$-strictly subexponential if there exist constants $C, \gamma \in (0, 1]$ such that for all $\boldsymbol{w} \in \mathbb{R}^d, \|\boldsymbol{w}\| = 1, t \geq 0$,

$$\mathbf{Pr}_{x \sim \mathcal{D}}[|\langle \boldsymbol{w}, \boldsymbol{x} \rangle| > t] \leq e^{-Ct^{1+\gamma}}.$$

These distributions have the following bounds on their moments.

**Fact C.4** (see Vershynin (2018)). *Let $\mathcal{D}$ on $\mathbb{R}^d$ be a $\gamma$-strictly subexponential distribution. Then for all $\boldsymbol{w} \in \mathbb{R}^d, \|\boldsymbol{w}\| = 1, t \geq 0, p \geq 1$, there exists a constant $C'$ such that*

$$\mathbb{E}_{x \sim \mathcal{D}}[|\langle \boldsymbol{w}, \boldsymbol{x} \rangle|^p] \leq (C'p)^{\frac{p}{1+\gamma}}.$$

*In fact, the two conditions are equivalent.*

We will use the following bounds on the concentration of subexponential moments in the analysis of our algorithm. This will be useful in showing the sample complexity $N$ required in order for the empirical moments of the sample $S$ concentrate around the moments of the training marginal $\mathcal{D}_{\boldsymbol{x}}$.

**Lemma C.5** (Moment Concentration of Subexponential Distributions). *Let $\mathcal{D}_{\boldsymbol{x}}$ be a distribution over $\mathbb{R}^d$ such that for any $\boldsymbol{w} \in \mathbb{R}^d$ with $\|\boldsymbol{w}\|_2 = 1$ and any $t \in \mathbb{N}$ we have $\mathbb{E}_{\boldsymbol{x} \sim \mathcal{D}_{\boldsymbol{x}}}[|\boldsymbol{w} \cdot \boldsymbol{x}|^t] \leq (Ct)^t$ for some $C \geq 1$. For $\alpha = (\alpha_i)_{i \in [d]} \in \mathbb{N}^d$, we denote with $\boldsymbol{x}^\alpha$ the quantity $\boldsymbol{x}^\alpha = \prod_{i=1}^d x_i^{\alpha_i}$, where $\boldsymbol{x} = (x_i)_{i \in [d]}$. Then, for any $\Delta, \delta \in (0, 1)$, if $S$ is a set of at least $N = \frac{1}{\Delta^2}(Cc)^{4\ell}\ell^{8\ell+1}(\log(20d/\delta))^{4\ell+1}$ i.i.d. examples from $\mathcal{D}_{\boldsymbol{x}}$ for some sufficiently large universal constant $c \geq 2$, we have that with probability at least $1 - \delta$, the following is true.*

*For any $\alpha \in \mathbb{N}^d$ with $\|\alpha\|_1 \leq 2\ell$ we have $|\mathbb{E}_{\boldsymbol{x} \sim S}[\boldsymbol{x}^\alpha] - \mathbb{E}_{\boldsymbol{x} \sim \mathcal{D}_{\boldsymbol{x}}}[\boldsymbol{x}^\alpha]| \leq \Delta.$*

*Proof.* Let $\alpha = (\alpha_i)_{i \in [d]} \in \mathbb{N}^d$ with $\|\alpha\|_1 \leq 2\ell$. Consider the random variable $X = \frac{1}{m_{\text{train}}} \sum_{\boldsymbol{x} \in S} \boldsymbol{x}^\alpha = \frac{1}{m_{\text{train}}} \sum_{\boldsymbol{x} \in S} \prod_{i \in [d]} x_i^{\alpha_i}$. We have that $\mathbb{E}[X] = \mathbb{E}_{\boldsymbol{x} \sim \mathcal{D}_{\boldsymbol{x}}}[\boldsymbol{x}^\alpha]$ and also the following.

$$\mathbf{Pr}[|X - \mathbb{E}[X]| > \Delta] \leq \frac{\mathbb{E}[(X - \mathbb{E}[X])^{2t}]}{\Delta^{2t}}$$
$$\leq \frac{2(4t)^t}{(N\Delta^2)^t} \mathbb{E}[(\boldsymbol{x}^\alpha - \mathbb{E}[\boldsymbol{x}^\alpha])^{2t}]$$

where the last inequality follows from the Marcinkiewicz–Zygmund inequality (see Ferger (2014)). We have that $\mathbb{E}[(\boldsymbol{x}^\alpha - \mathbb{E}[\boldsymbol{x}^\alpha])^{2t}] \leq 4^t \mathbb{E}[(\boldsymbol{x}^\alpha)^{2t}]$. Since $\|\alpha\|_1 \leq 2\ell$, we have that $\mathbb{E}[(\boldsymbol{x}^\alpha)^{2t}] \leq \sup_{\|\boldsymbol{w}\|_2=1}[\mathbb{E}[(\boldsymbol{w} \cdot \boldsymbol{x})^{4t\ell}]] \leq (4Ct\ell)^{4t\ell}$, which yields the desired result, due to the choice of $N$ and after a union bound over all the possible choices of $\alpha$ (at most $d^{2\ell}$). $\square$

## C.2 GENERAL THEOREM

We now present the assumptions that are required by our TDS learner under strictly sub-exponential distributions.

**Assumption C.6.** *For a function class $\mathcal{F} \subseteq \{\mathbb{R}^d \to \mathbb{R}\}$ consisting of $k$-subspaces juntas, and training and test distributions $\mathcal{D}, \mathcal{D}'$ over $\mathbb{R}^d \times \mathbb{R}$, we assuming the following.*

1. *For $f \in \mathcal{F}$, there exists an $(\epsilon, R)$-uniform approximation polynomial for $f$ with degree at most $\ell = R \log R \cdot g_{\mathcal{F}}(\epsilon)$, where $g_{\mathcal{F}}(\epsilon)$ is a function depending only on the class $\mathcal{F}$ and $\epsilon$.*

2. *For $f \in \mathcal{F}$, the value $r_f := \sup_{\|W\boldsymbol{x}\|_2 \leq R} |f(x)|$ is bounded by a constant $r > 0$.*

3. *The training marginal $\mathcal{D}_{\boldsymbol{x}}$ is a $\gamma$-strictly subexponential distribution.*

4. *The training and test labels are both bounded in $[-M, M]$ for some $M \geq 1$.*

Given this assumption, we now give the statement of the TDS learning algorithm.

**Theorem C.7** (TDS Learning via Uniform Approximation). *Assume Assumption C.6 holds. Let $\epsilon, \delta \in (0, 1)$. Then, algorithm (Algorithm 2) learns $\mathcal{F}$ in the TDS regression setting up to excess error $4\epsilon$ and has probability of failure $\delta$. The time complexity is $\text{poly}(d^s, \ln(1/\delta)^\ell, 1/\epsilon)$ where $s = \text{poly}\left((kg_{\mathcal{F}}(\epsilon)\log(r)\log(M/\epsilon))^{1+1/\gamma}\right)$ and TDS learns $\mathcal{F}$ with respect to $\mathcal{D}_{\boldsymbol{x}}$ up to excess error $4\epsilon$ and with failure probability $\delta$.*

The following lemma allows us to relate the squared loss of the difference of polynomials under a set $S$ and under $\mathcal{D}$, as long as we have a bound on the coefficients of the polynomials. This will be convenient in the analysis of the algorithm.

**Lemma C.8** (Transfer Lemma for Square Loss, see Klivans et al. (2024a)). *Let $\mathcal{D}$ be a distribution over $\mathbb{R}^d$ and $S$ be a set of points in $\mathbb{R}^d$. If $|\mathbb{E}_{\boldsymbol{x}\sim S}[\boldsymbol{x}^\alpha] - \mathbb{E}_{x\sim\mathcal{D}}[\boldsymbol{x}^\alpha]| \leq \Delta$ for all $\alpha \in \mathbb{N}^d$ with $\|\alpha\|_1 \leq 2\ell$, then for any degree $\ell$ polynomials $p_1, p_2$ with coefficients absolutely bounded by $B$, it holds that*

$$\left|\mathbb{E}_{\boldsymbol{x}\sim S}[(p_1(\boldsymbol{x}) - p_2(\boldsymbol{x}))^2] - \mathbb{E}_{x\sim\mathcal{D}}[(p_1(\boldsymbol{x}) - p_2(\boldsymbol{x}))^2]\right| \leq 4B^2 d^{2\ell}\Delta$$

---

**Algorithm 2:** TDS Regression via Uniform Approximation

**Input:** Parameters $\epsilon > 0, \delta \in (0,1), R \geq 1, M \geq 1$, and sample access to $\mathcal{D}, \mathcal{D}'_{\boldsymbol{x}}$

Set $\epsilon' = \epsilon/11, \delta' = \delta/4, \ell = R\log R \cdot g_{\mathcal{F}}(\epsilon), t = 2\log\left(\frac{2M}{\epsilon'}\right), B = r(2(k+\ell))^{3\ell}, \Delta = \frac{\epsilon'^2}{4B^2 d^{2\ell t}}$

Set $m_{\text{train}} = m_{\text{test}} = \text{poly}(M, \ln(1/\delta)^\ell, 1/\epsilon, d^\ell, r)$ and draw $m_{\text{train}}$ i.i.d. labeled examples $S$ from $\mathcal{D}$ and $m_{\text{test}}$ i.i.d. unlabeled examples $S'$ from $\mathcal{D}'_{\boldsymbol{x}}$.

For each $\alpha \in \mathbb{N}^d$ with $\|\alpha\|_1 \leq 2\max(\ell, t)$, compute the quantity

$\widehat{M}_\alpha = \mathbb{E}_{\boldsymbol{x}\sim S'}[\boldsymbol{x}^\alpha] = \mathbb{E}_{\boldsymbol{x}\sim S'}\left[\prod_{i\in[d]} x_i^{\alpha_i}\right]$

**Reject** and terminate if $|\widehat{M}_\alpha - \mathbb{E}_{\boldsymbol{x}\sim\mathcal{D}_{\boldsymbol{x}}}[\boldsymbol{x}^\alpha]| > \Delta$ for some $\alpha$ with $\|\alpha\|_1 \leq 2\max(\ell, t)$.

**Otherwise**, solve the following least squares problem on $S$ up to error $\epsilon'$

$$\min_p \quad \mathbb{E}_{(\boldsymbol{x},y)\sim S}\left[(y - p(\boldsymbol{x}))^2\right]$$

$$\text{s.t. } p \text{ is a polynomial with degree at most } \ell$$

$$\text{each coefficient of } p \text{ is absolutely bounded by } B$$

Let $\hat{p}$ be an $\epsilon'^2$-approximate solution to the above optimization problem.

**Accept** and output $\text{cl}_M(\hat{p}(\boldsymbol{x}))$.

---

*Proof.* We will prove soundness and completeness separately.

**Soundness.** Suppose the algorithm accepts and outputs $\text{cl}_M(\hat{p})$. Let $f^* = \arg\min_{f\in\mathcal{F}}[\mathcal{L}_{\mathcal{D}}(f) + \mathcal{L}_{\mathcal{D}'}(f)]$ and $f_{\text{opt}} = \arg\min_{f\in\mathcal{F}}[\mathcal{L}_{\mathcal{D}}(f)]$. By the uniform approximation assumption in Assumption C.6, there are polynomials $p^*, p_{\text{opt}}$ which are $(\epsilon, R)$-uniform approximations for $f^*$ and $f_{\text{opt}}$, respectively. Let $f^*$ and $f_{\text{opt}}$ have the corresponding matrices $W^*, W_{\text{opt}} \in \mathbb{R}^{k\times d}$, respectively. Denote $\lambda_{\text{train}} = \mathcal{L}_{\mathcal{D}}(f^*)$ and $\lambda_{\text{test}} = \mathcal{L}_{\mathcal{D}'}(f^*)$. Note that for any $f, g : \mathbb{R}^d \to \mathbb{R}$, "unclipping" both functions will not increase their squared loss under any distribution, i.e. $\|\text{cl}_M(f) - \text{cl}_M(g)\|_{\mathcal{D}} \leq \|f - g\|_{\mathcal{D}}$, which can be seen through casework on $\boldsymbol{x}$ and when $f(\boldsymbol{x}), g(\boldsymbol{x})$ are in $[-M, M]$ or not. Recalling that the training and test labels are bounded, we can use this fact as we bound the error of the hypothesis on $\mathcal{D}'$.

$$\mathcal{L}_{\mathcal{D}'}(\text{cl}_M(\hat{p})) \leq \mathcal{L}_{\mathcal{D}'}(\text{cl}_M(f^*)) + \|\text{cl}_M(f^*) - \text{cl}_M(\hat{p})\|_{\mathcal{D}'}$$
$$\leq \mathcal{L}_{\mathcal{D}'}(f^*) + \|\text{cl}_M(f^*) - \text{cl}_M(\hat{p})\|_{S'} + \epsilon'.$$

The second inequality follows from unclipping the first term and by applying Hoeffding's inequality, so that for $m_{\text{test}} \geq \frac{8M^4\ln(2/\delta')}{\epsilon'^4}$, the second term is bounded with probability $\geq 1 - \delta'$. Proceeding with more unclipping and using the triangle inequality:

$$\mathcal{L}_{\mathcal{D}'}(\text{cl}_M(\hat{p})) \leq \lambda_{\text{test}} + \|\text{cl}_M(f^*) - \text{cl}_M(p^*)\|_{S'} + \|\text{cl}_M(p^*) - \text{cl}_M(\hat{p})\|_{S'} + \epsilon'$$
$$\leq \lambda_{\text{test}} + \|\text{cl}_M(f^*) - \text{cl}_M(p^*)\|_{S'} + \|p^* - \hat{p}\|_{S'} + \epsilon'.$$

We first bound $\|\text{cl}_M(f^*) - \text{cl}_M(p^*)\|_{S'} = \sqrt{\mathbb{E}_{\boldsymbol{x}\sim S'}[(\text{cl}_M(f^*(\boldsymbol{x})) - \text{cl}_M(p^*(\boldsymbol{x})))^2]}$. Since $p^*(\boldsymbol{x})$ is an $(\epsilon, R)$-uniform approximation to $f^*(\boldsymbol{x})$, we separately consider when we fall in the region of good approximation ($\|W^*\boldsymbol{x}\| \leq R$) or not.

$$\mathbb{E}_{\boldsymbol{x}\sim S'}[(\text{cl}_M(f^*(\boldsymbol{x})) - \text{cl}_M(p^*(\boldsymbol{x})))^2]$$
$$= \mathbb{E}_{\boldsymbol{x}\sim S'}[(\text{cl}_M(f^*(\boldsymbol{x})) - \text{cl}_M(p^*(\boldsymbol{x})))^2 \cdot \mathbb{1}[\|W^*\boldsymbol{x}\| \leq R]$$
$$+ \mathbb{E}_{\boldsymbol{x}\sim S'}[(\text{cl}_M(f^*(\boldsymbol{x})) - \text{cl}_M(p^*(\boldsymbol{x})))^2 \cdot \mathbb{1}[\|W^*\boldsymbol{x}\| > R]]$$
$$\leq \epsilon^2 + \mathbb{E}_{\boldsymbol{x}\sim S'}[2(\text{cl}_M(f^*(\boldsymbol{x}))^2 + \text{cl}_M(p^*(\boldsymbol{x}))^2) \cdot \mathbb{1}[\|W^*\boldsymbol{x}\| > R]]$$

Then by applying Cauchy-Schwarz, (and similarly for $\mathrm{cl}_M(p^*)$):
$$\mathbb{E}_{\boldsymbol{x}\sim S'}[\mathrm{cl}_M(f^*(\boldsymbol{x}))^2 \cdot \mathbb{1}[\|W^*\boldsymbol{x}\| > R]] \leq \sqrt{\mathbb{E}_{\boldsymbol{x}\sim S'}[\mathrm{cl}_M(f^*(\boldsymbol{x}))^4]} \cdot \sqrt{\mathbf{Pr}_{\boldsymbol{x}\sim S'}[\|W^*\boldsymbol{x}\| > R]}.$$
By definition, $\mathrm{cl}_M(p^*)^2, \mathrm{cl}_M(f^*)^2 \leq M^2$. So it suffices to bound $\mathbf{Pr}_{\boldsymbol{x}\sim S'}[\|W^*\boldsymbol{x}\| > R]$, since we now have
$$\mathbb{E}_{\boldsymbol{x}\sim S'}[(\mathrm{cl}_M(f^*(\boldsymbol{x})) - \mathrm{cl}_M(p^*(\boldsymbol{x})))^2] \leq \epsilon^2 + 4M^2\sqrt{\mathbf{Pr}_{\boldsymbol{x}\sim S'}[\|W^*\boldsymbol{x}\| > R]}. \qquad (2)$$
In order to bound this probability of the test samples falling outside the region of good approximation, we use the property that the first $2t$ moments of $S'$ are close to the moments of $\mathcal{D}$ (as tested by the algorithm). Applying Markov's inequality, we have
$$\mathbf{Pr}_{\boldsymbol{x}\sim S'}[\|W^*\boldsymbol{x}\| > R] \leq \frac{\mathbb{E}_{\boldsymbol{x}\sim S'}[\|W^*\boldsymbol{x}\|^{2t}]}{R^{2t}}.$$

Write $\|W^*\boldsymbol{x}\|^{2t} = \left(\sum_{i=1}^k \langle W_i^*, \boldsymbol{x}\rangle^2\right)^t$, where $\sum_{i=1}^k \langle W_i^*, \boldsymbol{x}\rangle^2 = \sum_{i=1}^k \left(\sum_{j=1}^d W_{ij}^* x_j\right)^2$ is a degree 2 polynomial with each coefficient bounded in absolute value by $2k$ (noting that since $WW^\top = 1$, then $|W_{ij}| \leq 1$). Let $a_\alpha$ denote the coefficients of $\|W^*\boldsymbol{x}\|^{2t}$. Applying [Lemma A.7](#), $\sum_{\|\alpha\|_1 \leq 2t} |a_\alpha| \leq (2k)^t d^{2t} \leq d^{O(t)}$. By linearity of expectation, we also have $|\mathbb{E}_{\boldsymbol{x}\sim S'}[\|W^*\boldsymbol{x}\|^{2t} - \mathbb{E}_{x\sim\mathcal{D}}[\|W^*\boldsymbol{x}\|^{2t}]| \leq \sum_{\|\alpha\|_1 \leq 2t} |a_\alpha| \cdot \Delta \leq d^{O(t)} \cdot \Delta \leq \epsilon$, where $\Delta \leq \epsilon' \cdot d^{-\Omega(t)}$. Since $\mathcal{D}$ is $\gamma$-strictly subexponential, then by [Fact C.4](#), $\mathbb{E}_{x\sim\mathcal{D}}[\langle W_i^*, \boldsymbol{x}\rangle^{2t}] \leq (2C't)^{\frac{2t}{1+\gamma}}$. Then, we can bound the numerator $\mathbb{E}_{\boldsymbol{x}\sim S'}[\|W^*\boldsymbol{x}\|^{2t}] \leq \mathbb{E}_{x\sim\mathcal{D}}[\|W^*\boldsymbol{x}\|^{2t}] + \epsilon' \leq (Ckt)^{\frac{2t}{1+\gamma}}$ for some large constant $C$. So we have that
$$\mathbf{Pr}_{\boldsymbol{x}\sim S'}[\|W^*\boldsymbol{x}\| > R] \leq \frac{(Ckt)^{\frac{2t}{1+\gamma}}}{R^{2t}}.$$
Setting $t \geq C'(\log(M/\epsilon))$ and $R \geq C'(kt) \geq C'k\log(M/\epsilon)$ for large enough $C'$ makes the above probability at most $16\epsilon'^4/M^4$ so that $4M^2\sqrt{\mathbf{Pr}_{\boldsymbol{x}\sim S'}[\|W^*\boldsymbol{x}\| > R]} \leq \epsilon'^2$. Thus, from [Equation (2)](#), we have that
$$\|\mathrm{cl}_M(f^*) - \mathrm{cl}_M(p^*)\|_{S'} \leq \epsilon + \epsilon'.$$
We now bound the second term $\|\mathrm{cl}_M(p^*) - \mathrm{cl}_M(\hat{p})\|_{S'}$. By [Lemma C.5](#), the first $2\ell$ moments of $S$ will concentrate around those of $\mathcal{D}_{\boldsymbol{x}}$ whenever $m_{\mathrm{train}} \geq \frac{1}{\Delta^2}(Cc)^{4\ell}\ell^{8\ell+1}(\log(20d/\delta))^{4\ell+1}$, and similarly the first $2\ell$ moments of $S'$ match with $\mathcal{D}_{\boldsymbol{x}}$ because the algorithm accepted. Using the transfer lemma ([Lemma C.8](#)) when considering $p' = (p^* - \hat{p})^2$, along with the triangle inequality, we get:
$$\|p^*(\boldsymbol{x}) - \hat{p}(\boldsymbol{x})\|_{S'} \leq \|p^*(\boldsymbol{x}) - \hat{p}(\boldsymbol{x})\|_{\mathcal{D}} + \sqrt{4B^2 d^{2\ell}\Delta}$$
$$\leq \|p^*(\boldsymbol{x}) - \hat{p}(\boldsymbol{x})\|_S + 2\epsilon'$$
$$\leq \mathcal{L}_S(p^*) + \mathcal{L}_S(\hat{p}) + 2\epsilon',$$
where we note that we can bound the sum of the magnitudes of the coefficients by $r(2(k + \ell))^{3\ell}$ using [Lemma A.6](#). Recall that by definition $\hat{p}$ is an $\epsilon'^2$-approximate solution to the optimization problem in [Algorithm 2](#), so $\mathcal{L}_S(\hat{p}) \leq \mathcal{L}_S(p_{\mathrm{opt}}) + \epsilon'$. Plugging this in, we obtain
$$\|p^*(\boldsymbol{x}) - \hat{p}(\boldsymbol{x})\|_{S'} \leq \mathcal{L}_S(p^*) + \mathcal{L}_S(p_{\mathrm{opt}}) + 3\epsilon'$$
$$\leq \|p^* - \mathrm{cl}_M(f^*)\|_S + \mathcal{L}(\mathrm{cl}_M(f^*))_S$$
$$+ \|p_{\mathrm{opt}}(\boldsymbol{x}) - \mathrm{cl}_M(f_{\mathrm{opt}}(\boldsymbol{x}))\|_S + \mathcal{L}_S(\mathrm{cl}_M(f_{\mathrm{opt}})) + 3\epsilon'.$$
By applying Hoeffding's inequality, we get that $\|\mathrm{cl}_M(f^*) - y\|_S \leq \|\mathrm{cl}_M(f^*) - y\|_{\mathcal{D}} + \epsilon'$ which holds with probability $\geq 1 - \delta'$ when $m_{\mathrm{train}} \geq \frac{8M^4 \ln(2/\delta')}{\epsilon'^4}$. By unclipping $\mathrm{cl}_M(f^*)$, this is at most $\lambda_{\mathrm{train}} + \epsilon'$. Similarly, with probability $\geq 1 - \delta'$, $\|\hat{\mathrm{cl}}_M(f_{\mathrm{opt}}(\boldsymbol{x})) - y\|_S \leq \mathrm{opt} + \epsilon'$. It remains to bound $\|p^*(\boldsymbol{x}) - \mathrm{cl}_M(f^*)\|_S$ and $\|p_{\mathrm{opt}} - \mathrm{cl}_M(f_{\mathrm{opt}}(\boldsymbol{x}))\|_S$. The analysis for both is similar to how we bounded $\|\mathrm{cl}_M(p^*) - \mathrm{cl}_M(f^*)\|_S$, except since we do not clip $p^*$ or $p_{\mathrm{opt}}$ we will instead take advantage of the bound on $p^*(\boldsymbol{x})$ on $\|W^*\boldsymbol{x}\| > R$ (respectively $p_{\mathrm{opt}}(\boldsymbol{x})$ on $\|W_{\mathrm{opt}}\boldsymbol{x}\| > R$). We show how to bound $\|p^*(\boldsymbol{x}) - \mathrm{cl}_M(f^*)\|_S$:
$$\mathbb{E}_{\boldsymbol{x}\sim S}[(\mathrm{cl}_M(f^*(\boldsymbol{x})) - p^*(\boldsymbol{x}))^2] = \mathbb{E}_{\boldsymbol{x}\sim S}[(\mathrm{cl}_M(f^*(\boldsymbol{x})) - p^*(\boldsymbol{x}))^2 \cdot \mathbb{1}[\|W^*\boldsymbol{x}\| \leq R]$$
$$+ \mathbb{E}_{\boldsymbol{x}\sim S}[(\mathrm{cl}_M(f^*(\boldsymbol{x})) - p^*(\boldsymbol{x}))^2 \cdot \mathbb{1}[\|W^*\boldsymbol{x}\| > R]]$$
$$\leq \epsilon^2 + 2\mathbb{E}_{\boldsymbol{x}\sim S}[\mathrm{cl}_M(f^*(\boldsymbol{x}))^2 \cdot \mathbb{1}[\|W^*\boldsymbol{x}\| > R]]$$
$$+ 2\mathbb{E}_{\boldsymbol{x}\sim S}[p^*(\boldsymbol{x})^2 \cdot \mathbb{1}[\|W^*\boldsymbol{x}\| > R]].$$

We can bound the first expectation term with $\epsilon'^2/4$ since the same analysis holds for bounding $\mathbb{E}_{\boldsymbol{x}\sim S'}[\mathrm{cl}_M(f^*(\boldsymbol{x}))^2 \cdot \mathbb{1}[\|W^*\boldsymbol{x}\| > R]]$, except instead of matching the first $2t$ moments of $S'$ with $\mathcal{D}_{\boldsymbol{x}}$, we match the first $2\ell$ moments of $S$ with $\mathcal{D}_{\boldsymbol{x}}$. We use the strictly subexponential tails of $\mathcal{D}_{\boldsymbol{x}}$ to bound the second term. Cauchy-Schwarz gives

$$\mathbb{E}_{\boldsymbol{x}\sim S}[p^*(\boldsymbol{x})^2 \cdot \mathbb{1}[\|W^*\boldsymbol{x}\| > R]] \leq \sqrt{\mathbb{E}_{\boldsymbol{x}\sim S}[p^*(\boldsymbol{x})^4] \cdot \mathbf{Pr}_{\boldsymbol{x}\sim S}[\|W^*\boldsymbol{x}\| > R]]}$$

Note that by definition of $r$ and using that $p^*$ is an $(\epsilon, R)$-uniform approximation of $f^*$, then $p^*(\boldsymbol{x}) \leq (r+\epsilon)$ when $\|W^*\boldsymbol{x}\| \leq R$. By [Lemma A.6], $|p^*(\boldsymbol{x})| \leq (r+\epsilon)\cdot(2k\ell)^{c\ell}\|(W^*x)/R\|^{\ell}$ for sufficiently large constant $c_1 > 0$. Then since $R \geq 1$, $p^*(\boldsymbol{x}) \leq (r+\epsilon)^4 \cdot (2k\ell)^{c\ell}\|W^*\boldsymbol{x}\|^{4\ell}$. Then we have

$$\begin{aligned}
\mathbb{E}_{\boldsymbol{x}\sim S}[p^*(\boldsymbol{x})^4] &\leq (r+\epsilon)^4 \cdot (2k\ell)^{c_1\ell} \cdot \mathbb{E}_{\boldsymbol{x}\sim S}[\|W^*\boldsymbol{x}\|^{4\ell}] \\
&\leq (r+\epsilon)^4 \cdot (2k\ell)^{c_1\ell} \cdot (\mathbb{E}_{x\sim\mathcal{D}_{\boldsymbol{x}}}[\|W^*\boldsymbol{x}\|^{4\ell}] + 1) \\
&\leq (r+\epsilon)^4 \cdot (2k\ell)^{c\ell}
\end{aligned}$$

where using [Fact C.4] we bound on $\mathbb{E}_{x\sim\mathcal{D}_{\boldsymbol{x}}}[\|W^*\boldsymbol{x}\|^{4\ell}] \leq k^{2\ell}(4\ell)^{\frac{4C\ell}{1+\gamma}}$ similar to above, which can be upper bounded with $(2k\ell)^{c_2\ell}$ for $c_2 > 0$ a sufficiently large constant. Take $c = c_1 + c_2$. We bound $\mathbf{Pr}_{\boldsymbol{x}\sim S}[\|W^*\boldsymbol{x}\| > R]$ as follows:

$$\begin{aligned}
\mathbf{Pr}_{\boldsymbol{x}\sim S}[\|W^*\boldsymbol{x}\| > R]] &= \mathbf{Pr}_{\boldsymbol{x}\sim S}\left[\sum_{i=1}^{k}\langle W_i^*, \boldsymbol{x}\rangle^2 > R^2\right] \\
&\leq \sum_{i=1}^{k}\mathbf{Pr}_{\boldsymbol{x}\sim S}[\langle W_i^*, \boldsymbol{x}\rangle^2 > R^2/k] \\
&\leq k\sup_{\|\boldsymbol{w}\|_2=1}\mathbf{Pr}_{\boldsymbol{x}\sim S}[\langle W, \boldsymbol{x}\rangle^2 > R^2/k],
\end{aligned}$$

where the first inequality follows from a union bound. Since $\langle \boldsymbol{w}, \boldsymbol{x}\rangle^2$ is a degree 2 polynomial, we can view $\mathrm{sign}(\langle \boldsymbol{w}, \boldsymbol{x}\rangle^2 - R^2/k)$ as a degree-2 PTF. The class of these functions has VC dimension at most $d^2$ (e.g. by viewing it as the class of halfspaces in $d^2$ dimensions). Using standard VC arguments, whenever $m_{\text{train}} \geq C \cdot \frac{d^2 + \log(1/\delta')}{(\epsilon''/k)^2}$ for some sufficiently large universal constant $C > 0$, with probability $\geq 1 - \delta'$ we have

$$\mathbf{Pr}_{\boldsymbol{x}\sim S}[\langle \boldsymbol{w}, \boldsymbol{x}\rangle^2 > R^2/k] \leq \mathbf{Pr}_{x\sim\mathcal{D}_{\boldsymbol{x}}}[\langle \boldsymbol{w}, \boldsymbol{x}\rangle^2 > R^2/k] + \epsilon''/k.$$

Using the strictly subexponential tails of $\mathcal{D}_{\boldsymbol{x}}$, we have

$$\begin{aligned}
\mathbf{Pr}_{\boldsymbol{x}\sim S}[\|W^*\boldsymbol{x}\| > R]] &\leq k\left(\sup_{\|w\|=1}\mathbf{Pr}_{x\sim\mathcal{D}_{\boldsymbol{x}}}[\langle \boldsymbol{w}, \boldsymbol{x}\rangle^2 > R^2/k] + \epsilon''/k\right) \\
&\leq 2k\cdot\exp\left(-(R/k)^{1+\gamma}\right) + \epsilon''.
\end{aligned}$$

Choose $\epsilon'' = \frac{\epsilon'^4}{(r+\epsilon)^4(2k\ell)^{c\ell}}$. Putting it together:

$$\begin{aligned}
\mathbb{E}_{\boldsymbol{x}\sim S}[p^*(\boldsymbol{x})^4] \cdot \mathbf{Pr}_{\boldsymbol{x}\sim S}[\|W^*\boldsymbol{x}\| > R]] &\leq (r+\epsilon)^4 \cdot (2k\ell)^{c\ell}e^{-(R/k)^{1+\gamma}} + \epsilon'^4 \\
&\leq (r+\epsilon)^4 \cdot \exp\left(c\ell\log(2k\ell) - (R/k)^{1+\gamma}\right) + \epsilon'^4.
\end{aligned}$$

We want to bound the first part with $\epsilon'^4$. Equivalently, we need to show that the exponent is $\leq 4\ln\frac{\epsilon'}{r+\epsilon}$. Substituting $\ell = R\log R \cdot g_{\mathcal{F}}(\epsilon)$, we get that $c\ell\log(2k\ell) \leq cg_{\mathcal{F}}(\epsilon)R(\log R)^2\log(2kg_{\mathcal{F}}(\epsilon))$. Thus, it suffices to show that

$$\left(\frac{R}{k}\right)^{1+\gamma} \geq cg_{\mathcal{F}}(\epsilon)R(\log R)^2(2kg_{\mathcal{F}}(\epsilon)) - 4\ln\frac{\epsilon'}{r+\epsilon}.$$

This is satisfied when $R \geq \mathrm{poly}\left((kg_{\mathcal{F}}(\epsilon)\log(r)\log(M/\epsilon))^{1+\frac{1}{\gamma}}\right)$. Then, we have that

$$\mathbb{E}_{\boldsymbol{x}\sim S}[p^*(\boldsymbol{x})^2 \cdot \mathbb{1}[\|W^*\boldsymbol{x}\| > R]] \leq \epsilon'^2\sqrt{2}.$$

So,

$$\|\mathrm{cl}_M(f^*) - p^*\|_S \leq \sqrt{\epsilon^2 + 2 \cdot \epsilon'^2/4 + 2\epsilon'^2\sqrt{2}} \leq \epsilon + 2\epsilon'.$$

The same argument will also give

$$\|\mathrm{cl}_M(f_{\mathrm{opt}}(\boldsymbol{x})) - p_{\mathrm{opt}}(\boldsymbol{x})\|_S \leq \epsilon + 2\epsilon'.$$

Putting everything together, we have

$$\mathcal{L}_{\mathcal{D}'}(\mathrm{cl}_M(\hat{p})) \leq \lambda + \mathrm{opt} + 3\epsilon + 11\epsilon' \leq \lambda + \mathrm{opt} + 4\epsilon.$$

The result holds with probability at least $1 - 5\delta' = 1 - \delta$ (taking a union bound over 5 bad events).

**Completeness.** For completeness, it is sufficient to ensure that $m_{\mathrm{test}} \geq N$ for $N$ in Lemma C.5. This is because when $\mathcal{D}_{\boldsymbol{x}} = \mathcal{D}'_{\boldsymbol{x}}$, our test samples $S'$ are in fact being drawn from the subexponential distribution $\mathcal{D}_{\boldsymbol{x}}$. Then the moment concentration of subexponential distributions (Lemma C.5) gives that the empirical moments of $S'$ are close to the moments of $\mathcal{D}_{\boldsymbol{x}}$ with probability $\geq 1 - \delta'$. This is the only condition for acceptance, so when $\mathcal{D}_{\boldsymbol{x}} = \mathcal{D}'_{\boldsymbol{x}}$, the probability of acceptance is at least $1 - \delta$, as required.

**Runtime.** The runtime of the algorithm is $\mathrm{poly}(d^\ell, m_{\mathrm{train}}, m_{\mathrm{test}})$, where $\ell = R \log R \cdot g_{\mathcal{F}}(\epsilon)$. As noted above, the two lower bounds on $R$ required in the proof are satisfied by setting $R \geq \left((kg_{\mathcal{F}}(\epsilon) \log(r) \log(M/\epsilon))^{O(\frac{1}{\gamma})}\right)$. Note that the lower bounds we required for $m_{\mathrm{train}}$ in the proof are satisfied whenever $m_{\mathrm{train}} = \mathrm{poly}(M, \ln(1/\delta)^\ell, 1/\epsilon, d^\ell, r)$. For $m_{\mathrm{test}}$ we required that $m_{\mathrm{test}} \geq \frac{8M^4 \ln(2/\delta')}{\epsilon'^4}$ and also $m_{\mathrm{test}} \geq N$ for $N$ in Lemma C.5. This is satisfied by choosing $m_{\mathrm{test}} = m_{\mathrm{train}}$. Putting this altogether, we see that the runtime is $\mathrm{poly}(d^s, \ln(1/\delta)^\ell, 1/\epsilon)$ where $s = \left((kg_{\mathcal{F}}(\epsilon) \log(r) \log(M/\epsilon))^{O(1/\gamma)}\right)$. $\qquad\square$

### C.3 Applications

We are now ready to state our theorem for TDS learning neural networks with sigmoid activations.

**Theorem C.9** (TDS Learning for Nets with Sigmoid Activation and Strictly Subexponential Marginals). *Let $\mathcal{F}$ on $\mathbb{R}^d$ be the class of neural network with sigmoid activations, depth $t$ and weight matrices $\mathbf{W} = (W^{(1)}, \ldots, W^{(t)})$ such that $\|W\|_1 \leq W$. Let $\epsilon \in (0, 1)$. Suppose the training and test distributions $\mathcal{D}, \mathcal{D}'$ over $\mathbb{R}^d \times \mathbb{R}$ are such that the following are true:*

  1. *$\mathcal{D}_{\boldsymbol{x}}$ is $\gamma$-strictly subexponential,*

  2. *The training and test labels are bounded in $[-M, M]$ for some $M \geq 1$.*

*Then, Algorithm 2 learns the class $\mathcal{F}$ in the TDS regression up to excess error $\epsilon$ and probability of failure $\delta$. The time and sample complexity is at most $\mathrm{poly}(d^s, \log(1/\delta)^s)$ where $s = \left(k \log M \cdot (\|W^{(1)}\|_2^\infty W^{t-2}) \cdot (t \log(W/\epsilon))^{t-1}\right)^{O(\frac{1}{\gamma})}$.*

*Proof.* From Theorem A.21, we have that $\mathcal{F}$ there is an $(\epsilon, R)$-uniform approximation polynomial for $f$ with degree $\ell = O\left((R \log R) \cdot (\|W^{(1)}\|_2^\infty W^{t-2}) \cdot (t \log(W/\epsilon))^{t-1}\right)$. Here, let $g_{\mathcal{F}}(\epsilon) := (\|W^{(1)}\|_2^\infty W^{t-2}) \cdot (t \log(W/\epsilon))^{t-1}$. We also have that $r = \sup_{\|\boldsymbol{x}\|_2 \leq R, f \in \mathcal{F}} |f(\boldsymbol{x})| \leq \mathrm{poly}(Rk\|W^{(1)}\|_2^\infty W^{t-2})$ from the Lipschitzness of the sigmoid nets (Lemma A.18) and the fact that the sigmoid evaluated at 0 has value 1. The theorem now directly follows from Theorem C.7. $\qquad\square$

We now state our theorem on TDS learning neural networks with arbitrary Lipschitz activations.

**Theorem C.10** (TDS Learning for Nets with Lipschitz Activation with strictly subexponential marginals). *Let $\mathcal{F}$ on $\mathbb{R}^d$ be the class of neural network with $L$-Lipschitz activations, depth $t$ and weight matrices $\mathbf{W} = (W^{(1)}, \ldots, W^{(t)})$ such that $\|W\|_1 \leq W$. Let $\epsilon \in (0, 1)$. Suppose the training and test distributions $\mathcal{D}, \mathcal{D}'$ over $\mathbb{R}^d \times \mathbb{R}$ are such that the following are true:*

  1. *$\mathcal{D}_{\boldsymbol{x}}$ is $\gamma$-strictly subexponential,*

2. *The training and test labels are bounded in $[-M, M]$ for some $M \geq 1$.*

*Then, Algorithm 2 learns the class $\mathcal{F}$ in the TDS regression up to excess error $\epsilon$ and probability of failure $\delta$. The time and sample complexity is at most $\mathrm{poly}(d^s, \log(1/\delta^s))$ where $s = \left(k \log M \cdot \|W^{(1)}\|_2^\infty (WL)^{t-1}/\epsilon\right)^{O(\frac{1}{\gamma})}$.*

*Proof.* From Theorem A.19, we have that $\mathcal{F}$ there is an $(\epsilon, R)$-uniform approximation polynomial for $f$ with degree $\ell = O\left(Rk\sqrt{k} \cdot \|W^{(1)}\|_2^\infty (WL)^{t-1}/\epsilon\right)$. Here, let $g_{\mathcal{F}}(\epsilon) := k\sqrt{k}\|W^{(1)}\|_2^\infty (WL)^{t-1}/\epsilon$. We also have that $r = \sup_{\|\boldsymbol{x}\|_2 \leq R, f \in \mathcal{F}} |f(\boldsymbol{x})| \leq \mathrm{poly}(Rk\|W^{(1)}\|_2^\infty W^{t-2})$ from the Lipschitz constant (Lemma A.18) and the fact that the each individual activation has value at most $1$ when evaluated at $0$ (see Definition A.12. The theorem now directly follows from Theorem C.7. $\square$

# D ASSUMPTIONS ON THE LABELS

Our main theorems involve assumptions on the labels of both the training and test distributions. Ideally, one would want to avoid any assumptions on the test distribution. However, we demonstrate that this is not possible, even when the training marginal and the training labels are bounded, and the test labels have bounded second moment. On the other hand, we show that obtaining algorithms that work for bounded labels is sufficient even in the unbounded labels case, as long as some moment of the labels (strictly higher than the second moment) is bounded.

We begin with the lower bound, which we state for the class of linear functions, but would also hold for the class of single ReLU neurons, as well as other unbounded classes.

**Proposition D.1** (Label Assumption Necessity). *Let $\mathcal{F}$ be the class of linear functions over $\mathbb{R}^d$, i.e., $\mathcal{F} = \{\boldsymbol{x} \mapsto \boldsymbol{w} \cdot \boldsymbol{x} : \boldsymbol{w} \in \mathbb{R}^d, \|\boldsymbol{w}\|_2 \leq 1\}$. Even if we assume that the training marginal is bounded within $\{\boldsymbol{x} \in \mathbb{R}^d : \|\boldsymbol{x}\|_2 \leq 1\}$, that the training labels are bounded in $[0, 1]$, and that for the test labels we have $\mathbb{E}_{y \sim \mathcal{D}'_y}[y^2] \leq Y$ where $Y > 0$, no TDS regression algorithm with finite sample complexity can achieve excess error less than $Y/4$ and probability of failure less than $1/4$ for $\mathcal{F}$.*

The proof is based on the observation that because we cannot make any assumption on the test marginal, the test distribution could take some very large value with very small probability, while still being consistent with some linear function. The training distribution, on the other hand, gives no information about the ground truth and is information theoretically indistinguishable from the constructed test distribution. Therefore, the tester must accept and its output will have large excess error. The bound on the second moment of the labels does imply a bound on excess error, but this bound cannot be made arbitrarily small by drawing more samples.

*Proof of Proposition D.1.* Suppose, for contradiction that we have a TDS regression algorithm for $\mathcal{F}$ with excess error $\epsilon < Y/4$ and probability of failure $\delta < 1/4$. Let $m \in \mathbb{N}$ be the sample complexity of the algorithm and $p \in (0, 1)$ such that $m \ll 1/p$. We consider three distributions over $\mathbb{R}^d \times \mathbb{R}$. First $\mathcal{D}^{(1)}$ outputs $(0, 0)$ with probability $1$. Second, $\mathcal{D}^{(2)}$ outputs $(0, 0)$ with probability $1 - p$ and $(\frac{\sqrt{Y}}{\sqrt{p}}\boldsymbol{w}, \frac{\sqrt{Y}}{\sqrt{p}})$ with probability $p$, for some $\boldsymbol{w} \in \mathbb{R}^d$ with $\|\boldsymbol{w}\|_2 = 1$. Third, $\mathcal{D}^{(3)}$ outputs $(0, 0)$ with probability $1 - p$ and $(\frac{\sqrt{Y}}{\sqrt{p}}\boldsymbol{w}, 0)$ with probability $p$.

We consider two instances of the TDS regression problem. The first instance corresponds to the case $\mathcal{D} = \mathcal{D}^{(1)}$ and $\mathcal{D}' = \mathcal{D}^{(2)}$. The second corresponds to the case $\mathcal{D} = \mathcal{D}^{(1)}$ and $\mathcal{D}' = \mathcal{D}^{(3)}$. Note that the assumptions we asserted regarding the test distribution and the test labels are true for both instances. For $\mathcal{D}^{(2)}$, in particular, we have $\mathbb{E}_{y \sim \mathcal{D}_y^{(2)}}[y^2] = p \cdot (\sqrt{Y}/\sqrt{p})^2 = Y$. Moreover, in each of the cases, there is a hypothesis in $\mathcal{F}$ that is consistent with all of the examples (either the hypothesis $\boldsymbol{x} \mapsto 0$ or $\boldsymbol{x} \mapsto \boldsymbol{w} \cdot \boldsymbol{x}$), so $\mathrm{opt} := \min_{f \in \mathcal{F}}[\mathcal{L}_{\mathcal{D}}(f)] = 0 = \min_{f' \in \mathcal{F}}[\mathcal{L}_{\mathcal{D}}(f') + \mathcal{L}_{\mathcal{D}'}(f')] =: \lambda$.

Note that the total variation distance between $\mathcal{D}^{(1)}$ and $\mathcal{D}^{(2)}$ is $p$ and similarly between $\mathcal{D}^{(1)}$ and $\mathcal{D}^{(3)}$. Therefore, by the completeness criterion, as well as the fact that sampling only increases total variation distance at a linear rate, i.e., $\mathrm{d}_{\mathrm{tv}}((\mathcal{D})^{\otimes m}, (\mathcal{D}')^{\otimes m}) \leq m \cdot \mathrm{d}_{\mathrm{tv}}(\mathcal{D}, \mathcal{D}') \leq m \cdot p$, we have that

in each of the two instances, the algorithm will accept with probability at least $1 - m \cdot p - \delta$ (due to the definition of total variation distance[1]).

Suppose that the algorithm accepts in both instances (which happens w.p. at least $1 - 2\delta - 2mp$). By the soundness criterion, with overall probability at least $1 - 4\delta - 2mp$, we have the following.

$$p \cdot (h(\boldsymbol{x}) - 0)^2 < Y/4$$
$$p \cdot (h(\boldsymbol{x}) - \sqrt{Y}/\sqrt{p})^2 < Y/4$$

The inequalities above cannot be satisfied simultaneously, so we have arrived to a contradiction. It only remains to argue that $1 - 4\delta - 2mp > 0$, which is true if we choose $p < \frac{1-4\delta}{2m}$. Therefore, such a TDS regression algorithm cannot exist. $\qquad\square$

The lower bound of Proposition D.1 demonstrates that, in the worst case, the best possible excess error scales with the second moment of the distribution of the test labels. In contrast, we show that a bound on any strictly higher moment is sufficient.

**Corollary D.2.** *Suppose that for any $M > 0$, we have an algorithm that learns a class $\mathcal{F}$ in the TDS setting up to excess error $\epsilon \in (0, 1)$, assuming that both the training and test labels are bounded in $[-M, M]$. Let $T(M)$ and $m(M)$ be the corresponding time and sample complexity upper bounds.*

*Then, in the same setting, there is an algorithm that learns $\mathcal{F}$ up to excess error $4\epsilon$ under the relaxed assumption that for both training and test labels we have $\mathbb{E}[y^2 g(|y|)] \leq Y$ for some $Y > 0$ and $g$ some strictly increasing, positive-valued and unbounded function. The corresponding time and sample complexity upper bounds are $T(g^{-1}(Y/\epsilon^2))$ and $m(g^{-1}(Y/\epsilon^2))$.*

The proof is based on the observation that the effect of clipping on the labels, as measured by the squared loss, can be controlled by drawing enough samples, whenever a moment that is strictly higher than the second moment is bounded.

**Lemma D.3.** *Let $Y > 0$ and $g : (0, \infty) \to (0, \infty)$ be strictly increasing and surjective. Let $y$ be a random variable over $\mathbb{R}$ such that $\mathbb{E}[y^2 g(|y|)] \leq Y$. Then, for any $\epsilon \in (0, 1)$, if $M \geq g^{-1}(Y/\epsilon^2)$, we have $\sqrt{\mathbb{E}[(y - \mathrm{cl}_M(y))^2]} \leq \epsilon$.*

*Proof of Lemma D.3.* We have that $\mathbb{E}[(y - \mathrm{cl}_M(y))^2] \leq \mathbb{E}[y^2 \mathbb{1}\{|y| > M\}]$, because $y \geq \mathrm{cl}_M(y)$ and $y, \mathrm{cl}_M(y)$ always have the same sign, so $(y - \mathrm{cl}_M(y))^2 \geq y^2$ and also $(y - \mathrm{cl}_M(y))^2 = 0$ if $|y| \leq M$. Since $g(|y|)$ is non-zero whenever $y > 0$, we have $\mathbb{E}[y^2 \mathbb{1}\{|y| > M\}] = \mathbb{E}[y^2 \cdot \frac{g(|y|)}{g(|y|)} \cdot \mathbb{1}\{|y| > M\}]$. We now use the fact that $g$ is increasing to conclude that $\mathbb{E}[y^2 \mathbb{1}\{|y| > M\}] \leq \frac{\mathbb{E}[y^2 g(|y|)]}{g(M)} \leq \frac{Y}{g(M)}$. By choosing $M \geq g^{-1}(Y/\epsilon^2)$, we obtain the desired bound. $\qquad\square$

We are now ready to prove Corollary D.2, by reducing TDS learning with moment-bounded labels to TDS learning with bounded labels.

*Proof of Corollary D.2.* The idea is to reduce the problem under the relaxed label assumptions to a corresponding bounded-label problem for $M = g^{-1}(Y/\epsilon^2)$. In particular, consider a new training distribution $\mathrm{cl}_M \circ \mathcal{D}$ and a new test distribution $\mathrm{cl}_M \circ \mathcal{D}'$, where the samples are formed by drawing a sample $(\boldsymbol{x}, y)$ from the corresponding original distribution and clipping the label $y$ to $\mathrm{cl}_M(y)$. Note that whenever we have access to i.i.d. examples from $\mathcal{D}$, we also have access to i.i.d. examples from $\mathrm{cl}_M \circ \mathcal{D}$ and similarly for $(\mathcal{D}'_{\boldsymbol{x}}, \mathrm{cl}_M \circ \mathcal{D}'_{\boldsymbol{x}})$. Therefore, we may solve the corresponding TDS problem for $\mathrm{cl}_M \circ \mathcal{D}$ and $\mathrm{cl}_M \circ \mathcal{D}'$, to either reject or obtain some hypothesis $h$ such that

$$\mathcal{L}_{\mathrm{cl}_M \circ \mathcal{D}'}(h) \leq \min_{f \in \mathcal{F}}[\mathcal{L}_{\mathrm{cl}_M \circ \mathcal{D}}(f)] + \min_{f' \in \mathcal{F}}[\mathcal{L}_{\mathrm{cl}_M \circ \mathcal{D}}(f') + \mathcal{L}_{\mathrm{cl}_M \circ \mathcal{D}'}(f')] + \epsilon$$

Our algorithm either rejects when the algorithm for the bounded labels case rejects or accepts and outputs $h$. It suffices to show $\mathcal{L}_{\mathcal{D}'}(h) \leq \min_{f \in \mathcal{F}}[\mathcal{L}_{\mathcal{D}}(f)] + \min_{f' \in \mathcal{F}}[\mathcal{L}_{\mathcal{D}}(f') + \mathcal{L}_{\mathcal{D}'}(f')] + 4\epsilon$, because the marginal distributions do not change and completeness is, therefore, satisfied directly.

---

[1] We know that the algorithm would accept with probability at least $1 - \delta$ if the set of test examples was drawn from $(\mathcal{D}_{\boldsymbol{x}})^{\otimes m}$. Since $(\mathcal{D}'_{\boldsymbol{x}})^{\otimes m}$ is $(mp)$-close to $(\mathcal{D}_{\boldsymbol{x}})^{\otimes m}$, no algorithm can have different behavior if we substitute $(\mathcal{D}_{\boldsymbol{x}})^{\otimes m}$ with $(\mathcal{D}'_{\boldsymbol{x}})^{\otimes m}$ except with probability $m \cdot p$. Hence, any algorithm must accept with probability at least $1 - m \cdot p - \delta$.

It suffices to show that for any distribution $\mathcal{D}$, we have $|\mathcal{L}_{\mathcal{D}}(h) - \mathcal{L}_{\mathrm{cl}_M \circ \mathcal{D}}(h)| \leq \epsilon$. To this end, note that $\mathcal{L}_{\mathrm{cl}_M \circ \mathcal{D}}(h) = \sqrt{\mathbb{E}_{(\boldsymbol{x},y) \sim \mathcal{D}}[(\mathrm{cl}_M(y) - h(\boldsymbol{x}))^2]}$. We have the following.

$$
\begin{aligned}
\mathcal{L}_{\mathrm{cl}_M \circ \mathcal{D}}(h) &= \sqrt{\mathbb{E}_{(\boldsymbol{x},y) \sim \mathcal{D}}[(\mathrm{cl}_M(y) - h(\boldsymbol{x}))^2]} \\
&= \sqrt{\mathbb{E}_{(\boldsymbol{x},y) \sim \mathcal{D}}[(\mathrm{cl}_M(y) - y + y - h(\boldsymbol{x}))^2]} \\
&\leq \sqrt{\mathbb{E}_{(\boldsymbol{x},y) \sim \mathcal{D}}[(\mathrm{cl}_M(y) - y)^2]} + \sqrt{\mathbb{E}_{(\boldsymbol{x},y) \sim \mathcal{D}}[(y - h(\boldsymbol{x}))^2]} \\
&\leq \epsilon + \mathcal{L}_{\mathcal{D}}(h)
\end{aligned}
$$

The first inequality follows from an application of the triangle inequality for the $\mathcal{L}_2$-norm and the second inequality follows from Lemma D.3. The other side follows analogously. $\qquad\square$

