# OpenReview forum: "Learning Neural Networks with Distribution Shift: Efficiently Certifiable Guarantees"
_ICLR.cc/2025/Conference — ICLR 2025 Poster_

### Official Review · Reviewer_iuF7 · 2024-10-30

**Soundness:** 4
**Presentation:** 3
**Contribution:** 3
**Rating:** 8
**Confidence:** 3

**Summary:**

This work considers the Testable Learning with Distribution Shift (TDS) framework.

In TDS, the training and test time distributions can be different, and the learner receives labeled training examples as well as unlabeled test examples. The learner can either declare success—-in which case it must satisfy PAC-like guarantees—or it can declare failure if it detects too large a distribution shift, but this detection must be reliable.

While all previous works studied TDS in the classification setting, this work is concerned with TDS for (non-convex) regression. The authors present a general-purpose polynomial time TDS learning algorithm with theoretical performance guarantees.

The results imply TDS learnability for neural networks, under mild assumptions on the network activations, as well as a hypercontractivity assumption on the training distribution. This setting allows for the use of kernel methods and thereby essentially reduces the learning task to convex agnostic learning over polynomials.

**Strengths:**

- Extending the TDS framework to regression is an important contribution. As such, this paper is valuable for the broader statistical learning theory community.
- The tools employed for the proofs are technically non-trivial and could be valuable in their own right for future research in TDS. For example, the coupling of the training and test distribution is achieved through the use of data-dependent feature maps imported from the kernel framework. Additionally, the full analysis combines tools from polynomial approximation theory, hypercontractive concentration, and spectral concentration.
- The overall writing style is excellent, and the literature review seems to be thorough (though I am admittedly not familiar with the TDS line of work).

**Weaknesses:**

- Poor scaling of computation complexity:
While the runtime, as claimed, is polynomial in most of the parameters, it is exponential in $1/\epsilon$ for ReLU neurons and Lipschitz nets. For deep nets, the scaling is doubly exponential in the depth, and in most cases exponential in the width of the first hidden layer.
Even for the parameters that do scale polynomially, the exponents are very large and most likely quite far from being optimal. For example, for general function classes, even if one naively assumes $A,B,C,l$ all to be constant, the runtime of Algorithm 1 scales at least as $\Omega(1/\epsilon^{16})$ from constructing the matrices $\hat{\Phi}, \hat{\Phi’}$ for the spectral tester, which requires $\sim m\cdot N$ kernel evaluations.

- Very large sample complexity:
Algorithm 1 requires $m+N$ i.i.d. samples from both $D$ and $D’$, which for Lipschitz nets is doubly exponential in the depth, and exponential in $1/\epsilon$ and the width of the first hidden layer. Again, even if $A,B,C,l$ are all assumed to be constant, the sample complexity scales at least as $\Omega(1/\epsilon^{12})$ since the sample complexity is dominated by the size of the verification samples.

In the light of the above, the results have more of a proof-of-concept flavour, rather than providing a practically applicable algorithm. I suspect that Algorithm 1 cannot be ran in practise even for very small dimensions and parameter scales.

I want to make clear that I do not consider the above major weaknesses—theoretical results for complex learning settings are often difficult to obtain, and finding efficient, general purpose algorithms can take many years of research, or may not necessarily be possible in the first place, even with the best available mathematical tools.

I think that the paper would benefit from a short discussion regarding the limitations, especially regarding:
- Sample complexity (see above). Sample complexity is currently not discussed at all. I understand that this work is mostly concerned with runtime, but sample complexity should still not be completely left out of the picture in my opinion. Perhaps you could add a column in Table 1 that contains the sample complexities.
- I would suggest to mention at least once explicitly that, generally, rather large exponents are obtained for both computational and sample complexity. This limitation should be obvious to the reader even from a cursory read. It takes some effort to puzzle together the scaling exponents from Algorithm 1 and Table 2 due to the multitude of parameters involved. I suggest adding a short discussion of these limitations at the end of Section 1.1, or in a new Conclusion/Limitations section at the end of the paper.
- This is optional, but perhaps you could also highlight which steps in the analysis you believe to be particularly loose (or sharp), and what you believe the general limitations of the kernel-based approach to be. Such a discussion would make it easier for the reader to gauge the current state of this line of work, and what the major technical roadblocks are.

Note that in this iteration of ICLR, 10 pages are available, so space should not be an issue.

**Questions:**

- Which steps in the analysis do you believe to be particularly loose (or sharp), and what do you believe are the general limitations of a kernel-based analysis? Perhaps you could provide me with 1-2 key steps you believe to be most ripe for improvement in future work.
- In Def. 1.1, the completeness condition seems rather lax at first glance. I suspect that items 1 and 2 together imply that the algorithm must accept also in cases where $D_x$ is very close to, but not necessarily equal to, $D’_x$. Is this correct?
- In domain adaption, why is it standard to consider the training error part of the performance criterion? Intuitively, one might only care about the test error. Even if one does care about training error, why not consider a weighted combination of training and test error?

---

> ### Author Response · Authors · 2024-11-19
>
> Thank you for your time and feedback!
>
> Regarding the runtime in unbounded case, note that there is evidence that the dependence on the parameter $\epsilon$ must be exponential, **even without distribution shift** and assuming that the training marginal is Gaussian (see lines 198-204 and Diakonikolas et al., 2020b; Goel et al., 2020b). In the bounded case, it is an interesting open direction to achieve better runtime, but this is beyond the scope of this work.
>
> Overall, obtaining provable guarantees for learning often requires algorithms with high time complexity. The value of such results, especially in the TDS setting where we make no assumptions on the test distribution, is that one can be confident in the performance of the algorithms. This is crucial in critical settings where errors might lead to catastrophic results, and heuristic approaches are not appropriate.
>
> Regarding the limitations of our work, we thank the reviewer for their suggestions and we will consider including them in future versions of our paper, to make it more accessible to a general audience. However, we stress that our results are a part of a long line of works on computational learning theory. In this context, obtaining a polynomial-time algorithm (even if the exponent is large) is the gold standard for problems where only exponential time results were previously known. Reducing the exponent in these cases is usually reserved for follow up work.
>
> **Questions:**
> - We believe that an important open question is whether one can remove the hypercontractivity assumption from the results on bounded training distributions or show that this assumption is necessary. In the classical setting (where there is no distribution shift), similar results can be obtained without assuming hypercontractivity. It is not clear whether the need for hypercontractivity can be alleviated with a different analysis, or whether it is an inherent limitation of the Kernel method for TDS learning.
> - Our tests will actually accept whenever the training and test distributions have matching low-degree moments. In order to obtain testers that will accept when the two distributions are close in statistical distance, one might combine our results with prior work on TDS learning (Goel et al., 2024). See point 5 in our response to reviewer ovXB for more details.
> - You are correct that we only care about the test error. However, in order to obtain a bound on the test error and since we do not have access to labels from the test distribution, we use the training error as a proxy. The error of this proxy depends on the joint error $\lambda$ (see line 058) plus some excess error depending on the amount of distribution shift (usually measured in terms of some notion of distance between distributions, like the discrepancy distance, see, e.g., Mansour et al., 2009). Our tests essentially certify that the final excess error term is small.

---

> > ### Comment · Reviewer_iuF7 · 2024-11-20
> >
> > Thank you for the detailed feedback. My questions have been answered satisfactorily. As already mentioned in my initial review, I agree that in the context of similar theoretical works, the large exponents do not constitute a major weakness. The only weaknesses that at present I consider somewhat significant is the hypercontractivity assumption, which is not provably necessary in this setting.
> >
> > I am now convinced that this an important contribution for the learning theory community, both in terms of the presented results (fully poly-time TDS algorithm, applicable to certain classes of neural networks) and techniques (including a--to my best knowledge--novel application of the kernel trick). I am adjusting my score accordingly.

---

### Official Review · Reviewer_MWC1 · 2024-11-03

**Soundness:** 3
**Presentation:** 3
**Contribution:** 3
**Rating:** 6
**Confidence:** 3

**Summary:**

This paper presents two TDS learning algorithms for regression problems. The first algorithm integrates kernel methods into the TDS framework, achieving fully polynomial runtime for bounded and hypercontractive training distributions. For unbounded distributions (strictly subexponential distribution), the second algorithm utilizes the moment-matching testing and the uniform approximation approaches. The results apply to classes of Lipschitz neural networks.

**Strengths:**

The paper is well-written, and the results are clearly presented and demonstrated. It introduces a novel TDS learning algorithm that leverages kernel methods for regression problems, which are different from the classification setting that has been discussed in the literature.

**Weaknesses:**

There is a lack of experimental results supporting the validity of Theorem 3.6 and Theorem 4.2.

**Questions:**

Is it feasible to conduct experiments and create plots to validate the theorems?

---

> ### Author Response · Authors · 2024-11-19
>
> We wish to thank the reviewer for their feedback and for appreciating our work. We would like to stress that this is a theory paper and the main contribution is to a long line of theoretical research on provably efficient learning algorithms. We believe that experimental evaluation of our algorithms is an interesting and promising direction but more suitable for a separate project.

---

### Official Review · Reviewer_pjxs · 2024-11-04

**Soundness:** 4
**Presentation:** 3
**Contribution:** 2
**Rating:** 6
**Confidence:** 3

**Summary:**

This paper considers the Testable Learning with Distribution Shift (TDS) framework for learning neural networks under bounded or subgaussian inputs. In the TDS framework, the learner recieves m samples $(x_i, f^*(x_i) + \text{guassian noise})$ from some known covariate distribution $D_x$, and then $m$ unlableled test samples from an **unknown and arbitrary** covariate distribution $D'_x$. If $D'_x = D_x$, then the learner must output labels for the test samples, and is evaluated on the $\ell_2$ loss. If $D'_x$ is not $D_x$, then the learner can abstain.

The authors main technical result (Theorem 3.6) shows that if
   1. The covariates and the labels are bounded and
   2. the function class $\mathcal{F}$ is well approximated by some kernel K and
   3. The distribution $D_x$ is *hypercontractive* with respect to K,

then learning in the TDS framework is possible to $\epsilon^2$ mean squared error with polynomial in in the input dimension $d$ and $1/\epsilon$. A similar result (under stronger assumptions) is shown for unbounded covariates.

This is then applied to TDS for neural networks with bounded data, yielding an algorithm which runs in time polynomial in $d$ for sigmoid nets, 1-Lipshitz nets, and single ReLU units. Similar results are obtained for unbouned covariates, but the dependence is now exponential int the number of hidden neurons.

The result is proved using standard kernel regression techniques, with some adaptation for the TDS setting: (1) the algorithm rejects if the data is unbounded and (2) The algorithm uses a spectral testor to if the empirical covariance of the kernel matrix is significantly different from the samples from $D'_x$.

The authors prove the applications for neural networks by showing how too approximating $f^*$ (with a uniform bound on the domain) by a function in the the RKHS given by a low-dimensional polynomial kernel.

**Strengths:**

The TDS setting is an exciting area, and this paper extends to the TDS toolbox --- which had previously only been studies in classification settings or linear regression settings --- to the kernel regression setting, for both bounded and unbounded distributions. For unbounded distribution, this requires a novel analysis to control the approximation error on the tails of the distribution. In the bounded distribution setting, the main insight to get the results for NNs seems to be using uniform function approximations by polynomials (which is possible for neural nets with low degree polynomials!), as opposed to "sandwitching" approximations which were priorly used for TDS.

**Weaknesses:**

The application to neural networks does not seem to contribute much to our understanding of NNs, because it seems the main novelty is proving the specific assumptions relating to hypercontractivity, which I do not know how useful they are outside of this TDS setting. Further, the bounded distribution assumption is quite strong.

The paper could be more clear in many place:
- Pointing out where the technical novelty and main challenges are in the algorithm/analysis
  - For example, for bounded distributions, the Algorithm seems quite standard aside from the spectral test for the TDS part, which seems like a straightforward idea. Could the authors clarify what are the main technical challenges overcome here?
 - There is nowhere in the main body of the paper explaining what new approximation theory ideas are needed for the NN applications. Are there any potential uses for the new approximation techniques outside of this paper?
- The authors should explain at the beginning of "our techniques" that they intend to approximate NNs by a low-dimensional polynomial kernel, as per other cited works. This was confusing at first to understand why the text was talking about kernels without the connection to NNs made explicit.
- Where in Algorithm 1 is the part about finding a concise reference feature map?

**Questions:**

See weaknesses above.

---

> ### Author Response · Authors · 2024-11-19
>
> We thank the reviewer for their feedback.
>
> We would first like to clarify that we do not make any structural assumptions on the noise on the labels (the reviewer seems to suggest that the label noise is Gaussian).
>
> **Assumptions on the training marginal.** We stress that our results not only capture bounded distributions, but also arbitrary subgaussian distributions (Section 4), which is a significantly milder assumption.
> For bounded and hypercontractive distributions, we essentially match the best known guarantees for the standard setting (without distribution shift).
> For unbounded distributions, our results give new bounds even for the standard setting, where all of the known results require much stronger distributional assumptions (like Gaussianity).
> Note also that there is strong evidence that there are no dimension-efficient learning algorithms with provable guarantees under no distributional assumptions even for a single ReLU or sigmoid neuron (see, e.g., [DKMR’22]).
>
> In this context, our work provides a positive message: the algorithmic approaches proposed in standard learning theory (like Kernel methods and polynomial regression) can be enhanced with appropriate testers (like Algorithm 1) to provide strong guarantees even in the presence of distribution shift.
>
> [DKMR'22] Diakonikolas, I., Kane, D., Manurangsi, P., & Ren, L. (2022, May). Hardness of learning a single neuron with adversarial label noise. In *International Conference on Artificial Intelligence and Statistics* (pp. 8199-8213). PMLR.
>
> **Clarity questions.** Thank you for your suggestions to improve the presentation of our work. However, we kindly disagree that our paper lacks clarity in general (see also the Strengths section of Reviewer ovXB).
> - In section 1.2 we provide a high-level explanation of the main technical challenges and we provide more details in section 3 (particularly, subsection 3.1) and section 4. Regarding Algorithm 1 (for bounded distributions), as we stress in lines 110-117, the straightforward idea of matching pairwise correlations of the expanded features provides suboptimal runtime. We instead follow a data-dependent testing approach (see, e.g., lines 118-126 and section 3.1), which crucially uses the kernel trick in a non-standard way. In particular, we use test examples to form a data-dependent feature map $\phi$ (see lines 298-318 for the definition of $\phi$) and test the pairwise correlations of the corresponding features.
> - For an explanation on the approximation theory ideas for our applications on neural networks with unbounded training marginals, see lines 474-485.
> - We explain how we use the polynomial kernels in lines 127-130. However, motivated by your comment, we will consider putting this explanation in the beginning of section 1.2 for future versions of our work.
> The definition of the concise feature map $\phi$ is given in lines 298-318. The way it is used in Algorithm 1 is through matrices $\hat\Phi, \hat \Phi’$ (see lines 361-371 for the relationship between $\phi$ and $\hat\Phi, \hat\Phi’$).

---

> > ### Comment · Reviewer_pjxs · 2024-12-03
> >
> > Thank you for your responses. I will retain my score.

---

### Official Review · Reviewer_ovXB · 2024-11-05

**Soundness:** 4
**Presentation:** 4
**Contribution:** 4
**Rating:** 8
**Confidence:** 3

**Summary:**

This paper studies distribution shift for nonconvex regression using the recently introduced Testable Learning with Distribution Shift framework (TDS) (Klivans et al.’24). The setup is as follows. A TDS  learner observe iid labeled training data from train distribution $D$, along with *unlabeled* iid test data from some other test distribution $D’$. Then, the learner can decide to abstain from predicting on the test data. If it chooses to not abstain, it must produce a hypothesis $h \in \mathcal{F}$ generalize on the test data, where generalization is defined with respect to some function class $\mathcal{F}$.

The paper designs efficient TDS algorithms under a bounded label assumption (although this is slightly relaxed in the appendix) and sufficiently nice *training* marginals (bounded and hypercontractive or strictly subexponential), when the function class is a Lipschitz neural network. Notably, the paper makes no assumption about the test marginal. The efficient algorithms are based on kernel methods.

**Strengths:**

1. The paper comes up with a generic TDS framework using kernel methods, and then verifying the assumptions for a number of concrete examples. Previous algorithms in these testable settings relied on moment-based algorithms and various notions of polynomial approximation.
2. The paper discusses the numerous assumptions on the marginals / label distributions quite extensively. There is also a matching lower bound in the appendix that shows that some assumptions on the test labels are needed.
3. The algorithm itself is quite interesting, as it certifies a valid comparison between the training and test marginals via the spectral tester; this latter tester only works under the hypercontractivity assumption.
4. The results in Section 4 also handle the unbounded setting, by using a moment-based testable learner using polynomial regression instead, building on the previous TDS results of Klivans et al.’24.
5. The paper is well written, and gives substantial intuitions about how the proof works.

**Weaknesses:**

1. Although I appreciated that the paper does not make assumptions on the test marginal, it feels like this point is a little misleading, since the whole point of the tester is to certify the niceness of the test marginal. I can definitely agree that it is better to have a provable verifier that ensures your learner works on distribution than not, though.

**Questions:**

1. Can the authors comment a bit on when it would be realistic to have access to unlabeled test examples? I know that it is a standard setup for domain adaptation, but it would be good to have some discussion on it
2. The statement of Lemma A.17 seems unfinished.
3. It seems that one needs to prove uniform convergence for the function classes used for the TDS algorithm to work. Are there any situations where we might expect to be able to surpass this barrier, as uniform convergence is quite a strong assumption
4. Line 330, 333, should $\overline{S}_{\mathrm{ref}}$ have no overline?
5. If I have multiple test time distributions, it seems that a naive extension of the current algorithm would need to retrain my TDS learner for each test distribution? Might there be a way to design an algorithm which can be amortized over multiple different test time distributions?

---

> ### Author Response · Authors · 2024-11-19
>
> We wish to thank the reviewer for appreciating our work and for their thorough feedback!
>
> **Question 1.** One motivating example for the setting we consider is when one is, say, given a model that is trained on data from some hospital A in some part of the world and then is asked whether the predictions of the model can be trusted in order to decide the correct treatments for the patients in some other hospital B elsewhere. In this case, the learner does have unlabeled examples from hospital B: each patient has a file with the results of their exams, as well as any experienced symptoms. The crucial question is whether we can trust our model to label patients from hospital B, and knowing even when to reject is very useful. In general, TDS learning is particularly important for critical tasks, where obtaining even a small number of labels is very expensive.
>
> **Question 2.** Thank you for catching this, we have included the full statement in the revision.
>
>
> **Question 3.** Note that our algorithms only require uniform convergence with respect to the training distribution, which is structured. We do **not** require uniform convergence with respect to the test distribution, which can be arbitrary. This enables, for example, a sample complexity upper bound which is independent from the input dimension $d$ in the case of bounded training marginals (see Algorithm 1). We will highlight this in future revisions of the paper.
>
> Relaxing this requirement (uniform convergence with respect to the training distribution) would be an interesting result, as it would imply an information-theoretic separation between TDS learning and a closely related learning primitive (testable agnostic learning, see Rubinfeld & Vasilyan, 2023). In particular, Rademacher complexity is known to characterize the sample complexity of testable agnostic learning for any function class (see Gollakota et al., 2023). However, proving a lower bound for the sample complexity of TDS learning in terms of the Rademacher complexity or showing that no such bound exists remains open.
>
> **Question 4.** We use the overline notation to refer to a set of labeled examples and distinguish it from its unlabeled counterpart ($\{(x,y)\}$ versus $\{(x)\}$). It seems that the notation is consistent in lines 330, 333 (Algorithm 1).
>
>
> **Question 5.** The reviewer is raising a very interesting question. Our tests are formally guaranteed to accept only when the test distribution is equal to the training distribution. However, there are recent techniques in TDS learning which yield tests that accept much wider classes of test distributions. For example, Goel et al. (2024) showed that, in the case of binary classification, it is possible to accept all distributions that are close to the training distribution in total variation distance. We expect this result to extend to our setting with an appropriate modification of our algorithms, but we did not work it out. Moreover, Chandrasekaran et al. (2024) showed that, for some function classes like halfspaces, there are TDS learners that accept whenever the test marginal is sufficiently structured (i.e., has some mild concentration and anti-concentration properties). Such distributions can even be far from the training marginal in statistical distance. It is interesting to explore whether such results can also be obtained in the regression setting.
>
> Finally, note in our algorithms, the candidate output hypothesis does not depend on the examples from the test distribution. This means that if we wanted to test a number of different test distributions, we would not need to train the model from scratch, but rather only use test examples to decide whether to accept or reject.

---

> > ### Comment · Reviewer_ovXB · 2024-11-24
> >
> > Thank you for addressing my questions so thoroughly. I have left my score unchanged. Good luck!

---

### Meta-Review · Area_Chair_EDxD · 2024-12-18

**Metareview:**

This paper presents the first provably efficient algorithms for learning neural networks under distribution shift, working within the Testable Learning with Distribution Shift (TDS) framework. Unlike prior work focused on classification, the proposed approach addresses nonconvex regression with real-valued networks and arbitrary Lipschitz activations, applying to training distributions with strictly sub-exponential tails. For bounded and hypercontractive training distributions, a polynomial-time algorithm is proposed for TDS learning one hidden-layer networks with sigmoid activations by leveraging classical kernel methods and data-dependent feature maps.

This paper adds interesting and useful tools (both in algorithm and analysis) to the TDS framework, addressing a very challenging problem with novel use of the best available mathematical tools.  The presentation is excellent.  Although its practical significance is left for future investigation, the paper makes a solid contribution to the conference.

**Additional Comments On Reviewer Discussion:**

The rebuttal has been noted by the reviewers and have been taken into account by the AC in the recommendation of acceptance/rejection.

---

### Decision · Program_Chairs · 2025-01-22

Accept (Poster)